# RIEMANNIAN FUZZY K-MEANS
# ON PRODUCT MANIFOLDS

## ABSTRACT

**In this paper, we address an open problem: how to perform fast clustering on product manifolds.** With the increasing interest in non-Euclidean data representations, clustering such data has become an important problem. However, a naive extension of the classic K-Means algorithm to product manifolds requires $\mathcal{O}(\nu\omega)$ time, where $\omega$ is the number of alternating iterations and $\nu$ is the time complexity of each Riemannian optimization. Due to the need for numerous Riemannian optimizations, the naive Riemannian K-Means (NRK) is not suitable for large-scale data. To this end, we propose the Riemannian Fuzzy K-Means (RFK) algorithm for product manifolds, which reduces the time complexity to $\mathcal{O}(\nu)$. Importantly, RFK is not a straightforward extension of K-Means or Fuzzy K-Means to manifolds, it avoids the computation of the Fréchet mean and and achieve a true single-loop optimization. Furthermore, we introduce Radan to accelerate the optimization of RFK. We conduct extensive experiments. RFK and Radan outperform across nearly all metrics in almost every dataset, reaching an impressive level of performance. **RFK and Radan have been integrated into several non-Euclidean machine learning libraries, such as here. (See Appendix G)**

## 1 INTRODUCTION

Non-Euclidean data representations have received widespread attention. Examples include text embeddings in hyperbolic space (Dhingra et al., 2018), tree embeddings in the Poincaré disk (Nickel & Kiela, 2017), and representations of cell-cycle data on the sphere (Bjerregaard et al., 2025). This is because many real-world datasets exhibit non-Euclidean structure, and embedding such data in appropriate non-Euclidean spaces can better preserve those structures (Sinha et al., 2024; Khan et al., 2025). For example, hyperbolic space captures the hierarchical structure (Mandica et al., 2024a; Chlenski et al., 2024), while spheres retain the periodic information (Bonev et al., 2025). Many datasets not only have a single structure, so to preserve as much information as possible (Gu et al., 2018), it is necessary to represent them on product manifolds.

A product manifold is formed by the Cartesian product of multiple manifolds (Wang et al., 2021), i.e., $\mathcal{M} = \mathcal{M}_1 \times \cdots \times \mathcal{M}_Q = \otimes_{p=1}^{Q} \mathcal{M}_p$, where $\mathcal{M}_p$ denotes the $p$-th component manifold of the product manifold, with $p \in \{1, \ldots, Q\}$. The product manifold inherits the characteristics of each component manifold and possesses greater expressive power (Chlenski et al., 2025a). As a result, product manifolds have been widely used for representing data from diverse domains (Sun et al., 2022; McNeela et al., 2023; Xu et al., 2022; Chen et al., 2025), and clustering data represented on such manifolds or their components has become an important problem (Sun et al., 2023a).

A natural approach is to naively extend the K-Means to product manifolds, referred to as Naive Riemannian K-Means (NRK) (Miolane et al., 2020). However, we point out that NRK incurs a time complexity of $\mathcal{O}(\nu\omega)$, where $\omega$ is the number of alternating iterations and $\nu$ is the time complexity of each Riemannian optimization (Yuan et al., 2025b). This results in a double-loop structure to solve the clustering problem, which is unacceptable for large-scale data. Therefore, how to perform clustering efficiently remains an open problem that requires a solution (Tepper et al., 2018).

To address this problem, we propose Riemannian Fuzzy K-Means, abbreviated as RFK. Specifically, we consider the equivalent relaxed version of K-Means, namely Fuzzy K-Means (Dehariya et al., 2010). We identify a special structure of Fuzzy K-Means and leverage a particular technique to

transform the required double loop into a single loop and reduces the time from complexity $\mathcal{O}(\nu\omega)$ to $\mathcal{O}(\nu)$. In other words, the previously required $\omega$ times Riemannian optimizations are reduced to only 1, significantly lowering the computational cost, where $\omega \gg 1$.

To further accelerate RFK, we adapt the well-known Nesterov adaptive optimization algorithm (Adan) (Xie et al., 2024) to product manifolds, resulting in a Riemannian Nesterov acceleration method, termed Radan. We also establish the regret bound (Mukkamala & Hein, 2017) and convergence properties of Radan under certain conditions.

We validate our algorithm on a wide range of datasets. Specifically, we perform clustering using various methods on data represented in hyperbolic space, spherical manifolds, Euclidean space, and their product manifolds. We compare the speed of RFK with that of NRK, the speed of Radan with that of Riemannian Adam (Becigneul & Ganea, 2019), and the clustering performance of RFK with several state-of-the-art clustering algorithms. We conducted extensive experiments, which yielded remarkable results: RFK significantly outperforms NRK in speed, Radan converges faster than Radam, and RFK achieved the best clustering performance on nearly all datasets. In summary, our contributions are following.

- We address the open problem of fast clustering on product manifolds by proposing the RFK algorithm, which reduces the time complexity from $\mathcal{O}(\nu\omega)$ of the naive Riemannian K-Means to $\mathcal{O}(\nu)$.

- We modify the Adan optimizer to make it compatible with product manifolds, resulting in Radan, and provide theoretical guarantees including a regret bound and convergence proof.

- We conduct extensive numerical experiments to demonstrate the effectiveness of our algorithm. RFK is significantly faster than NRK, Radan provides acceleration over the Riemannian Adam (Radam), and RFK substantially outperforms existing algorithms in clustering metrics on manifold-represented data.

In addition, we propose a new insight, pointing out that the reason NRK cannot be accelerated lies in its hard assignment. We recommend RFK instead of NRK for clustering on manifolds.

## 2 PRELIMINARIES

### 2.1 NOTATIONS

Let the dataset be $X = \{x_1, \ldots, x_N\}$, and let $c_j$ denote the $j$-th cluster center. $C$ is the number of clusters. For a product manifold denoted by $\mathcal{M}$, each of its component manifolds is written as $\mathcal{M}_p$, such that $\mathcal{M} = \otimes_{p=1}^{Q}\mathcal{M}_p$. For any $\mathbf{x} \in \mathcal{M}$, $\mathbf{x}$ can be represented as $(x^1, x^2, \ldots, x^Q)$, where $x^p \in \mathcal{M}_p$. For any points $x^p$ and $y^p$ on the component manifold $\mathcal{M}_p$, $d_p(x^p, y^p)$ denotes the geodesic distance between $x^p$ and $y^p$ on $\mathcal{M}_p$, the geodesic distance on $\mathcal{M}$ is denoted by $d(x, y)$.

Let $\mathbb{H}^{h_i, K}$ denote a Lorentz hyperbolic space of dimension $h_i$ with curvature $K$, $\mathbb{S}^{s_i, K}$ denote a spherical manifold of dimension $s_i$ with curvature $K$, $\mathbb{R}^{r_i}$ denote a Euclidean space of dimension $r_i$, and $\mathbb{D}$ denote a two-dimensional Poincaré disk. Especially, when the curvatures of $\mathbb{S}^{s_i, K}$ and $\mathbb{H}^{h_i, K}$ are $(1, -1)$, we denote them simply as $\mathbb{S}^{s_i}$ and $\mathbb{H}^{h_i}$, respectively.

$T_{x^p}\mathcal{M}_p$ denotes the tangent space of the component manifold $\mathcal{M}_p$ at point $x^p$, and $\|\cdot\|$ denotes the norm in Euclidean space. The parallel transport on $\mathcal{M}_p$ from point $x^p$ to $y^p$ is denoted by $\varphi^p_{x^p \to y^p}(u^p)$, where $u^p \in T_{x^p}\mathcal{M}_p$. When there is no ambiguity, it is abbreviated as $\varphi^p(u^p)$. The parallel transport on the product manifold $\mathcal{M}$ is denoted by $\varphi_{x \to y}(u)$. The exponential map on $\mathcal{M}_p$ is denoted by $\mathrm{Exp}^p_{c^p}(u^p)$, and the exponential map on $\mathcal{M}$ is denoted by $\mathrm{Exp}_c(u)$. $Log^p_{c^p}(x^p)$ denotes the logarithmic map on $\mathcal{M}_p$. $Log_c(x)$ denotes the logarithmic map on the product manifold $\mathcal{M}$; $log(\cdot)$ refers to the natural logarithm. All the notations are summarized in Table 4.

### 2.2 CONSTANT-CURVATURE SPACES AND PRODUCT MANIFOLDS

Constant-curvature spaces (Jos et al., 1967) refer to one of the following: spherical spaces (positive curvature), hyperbolic spaces (negative curvature) or Euclidean spaces(Alekseevskij et al., 1993).

For an $s$-dimensional sphere $\mathbb{S}^{s,K}$, it can be represented as $\mathbb{S}^{s,K} = \left\{ x \in \mathbb{R}^{s+1} \mid \|x\| = \frac{1}{K}, \ K > 0 \right\}$. $\forall x, y \in \mathbb{S}^{s,K}$, the geodesic distance between $x$ and $y$ is $d(x,y) = \frac{\cos^{-1}(K^2 \langle x,y \rangle)}{K}$, where $\langle x, y \rangle$ denotes the normal inner product in $\mathbb{R}^{s+1}$ (Whittlesey, 2019).

For an $h$-dimensional hyperbolic space $\mathbb{H}^{h,K}$, it can be represented as: $\mathbb{H}^{h,K} = \left\{ x \in \mathbb{R}^{h+1} \mid \|x\|_h = \langle x, x \rangle_h = -\frac{1}{K^2}, \ K < 0, \ x^0 \geq 0 \right\}$, where any $x \in \mathbb{H}^{h,K}$ is written as $x = (x^0, \dots, x^h)$, with $x^i \in \mathbb{R}^1$ (Iversen, 1992), and the Lorentzian inner product (Tsamparlis, 2024) is defined as $\langle x, y \rangle_h = -x^0 y^0 + \sum_{i=1}^{h} x^i y^i$. For any $x, y \in \mathbb{H}^{h,K}$, the geodesic distance (He et al., 2025) between $x$ and $y$ is given by $d(x,y) = -\frac{\cosh^{-1}(K^2 \langle x,y \rangle_h)}{K}$. where $\mathbb{H}$ is also known as the well-known Lorentz (hyperboloid) model of hyperbolic space.

A product manifold can be represented as $\mathcal{M} = \otimes_{p=1}^{Q} \mathcal{M}_p$. For any $x, y \in \mathcal{M}$, the geodesic distance is generally given by Equation (1), where $x^p \in \mathcal{M}_p$ (Fumero et al., 2021).

$$d(x,y) = \sqrt{\sum_{p=1}^{Q} d_p^2(x^p, y^p)}, \ x,y \in \mathcal{M} = \otimes_{p=1}^{Q} \mathcal{M}_p, \ x^p, y^p \in \mathcal{M}_p \tag{1}$$

When we focus on product manifolds composed of constant-curvature spaces, the structure becomes $\mathcal{M} = \otimes_{i=1}^{n} \mathbb{S}^{s_i, K} \times \otimes_{j=1}^{m} \mathbb{H}^{h_j, K} \times \mathbb{R}^r$. The dimension is $\sum_{i=1}^{n} s_i + \sum_{j=1}^{m} h_j + r$ (Lui, 2012).

### 2.3 K-MEANS AND FUZZY K-MEANS

The K-Means algorithm is a well-known clustering method (Likas et al., 2003; Na et al., 2010), and its optimization problem can be formulated as following(Sinaga & Yang, 2020):

$$\begin{cases} \min\limits_{c_j, u_{ij}} J_{KM} = \sum_{i=1}^{N} \sum_{j=1}^{C} u_{ij} \|x_i - c_j\|^2 \\ \text{s.t.} \quad \sum_{j=1}^{C} u_{ij} = 1, \quad u_{ij} \in \{0,1\}, \quad \forall i = 1, \dots, N, \ \forall j = 1, \dots, C \end{cases} \tag{2}$$

Here, $u_{ij}$ is an indicator variable, where $u_{ij} = 1$ indicates that the $i$-th sample belongs to the $j$-th cluster. This problem is typically solved by alternating updates of $\{u_{ij}\}$ and $\{c_j\}$.

Fuzzy K-Means is a relaxed version of K-Means (Xu et al., 2016; Krasnov et al., 2023), in which the constraint $u_{ij} \in \{0,1\}$ is relaxed to $0 \leq u_{ij} \leq 1$, with the additional requirement that $\sum_{j=1}^{C} u_{ij} = 1$, where $C$ is the number of clusters. Moreover, the loss term of K-Means $u_{ij} \|x_i - c_j\|^2$ is replaced by $u_{ij}^m \|x_i - c_j\|^2$ when using fuzzy K-Means , where $m$ is the fuzziness parameter (Li & Wang, 2023; Suganya & Shanthi, 2012; Bezdek et al., 1984). Other related work can be found in Appendix C.

## 3 OUR PROPOSED METHOD

### 3.1 NAIVE EXTENSION OF K-MEANS

The K-Means is clearly unsuitable for data represented on a manifold $\mathcal{M}$, for two following reasons and shown in Figure 1.

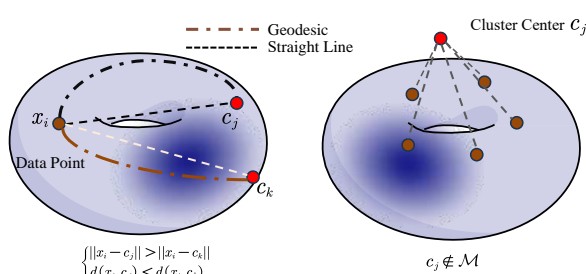

Figure 1: Visualization of the two reasons

- **Incorrect distance comparisons:** When data lie on a manifold, the Euclidean distance may have $\|x_i - c_j\| < \|x_i - c_k\|$, while the actual geodesic distances satisfy $d(x_i, c_j) > d(x_i, c_k)$. This mismatch can lead to incorrect cluster assignments.

- **Invalid cluster centers:** Without appropriate constraints, the computed cluster centers $c_j$ may lie outside the manifold, i.e., $c_j \notin \mathcal{M}$, rendering the cluster centers meaningless in the context of the manifold. Meanwhile, the geodesic distance $d(x_i, c_j)$ is not well-defined.

Therefore, a naive approach is to replace the Euclidean distance with geodesic distance and impose the constraint that the cluster centers lie on the manifold. This leads to the following Equation (3).

$$
\begin{cases}
\min\limits_{c_j, u_{ij}} J_{KM}(u_{ij}, c_j) = \sum_{i=1}^{N} \sum_{j=1}^{C} u_{ij} d^2(x_i, c_j) = \sum_{i=1}^{N} \sum_{j=1}^{C} \sum_{p=1}^{Q} u_{ij} d_p^2(x_i^p, c_j^p) \\
\text{s.t.} \sum_{j=1}^{C} u_{ij} = 1, u_{ij} \in \{0, 1\}, \quad \forall i = 1, \ldots, N, \forall j = 1, \ldots, C \\
\text{s.t.} \quad c_j \in \mathcal{M}, \ \mathcal{M} = \otimes_{p=1}^{Q} \mathcal{M}_p, \quad \forall j = 1, \ldots, C
\end{cases}
\tag{3}
$$

Similar to K-Means in Euclidean space, this problem can be solved by alternating updates of $\{u_{ij}\}$ and $\{c_j\}$. The update of $\{u_{ij}\}$ is identical to that in the Euclidean case: for each $x_i$, one simply identifies the cluster center $c_j$ that minimizes $\sum_{p=1}^{Q} d_p^2(x_i^p, c_j^p)$, and sets the corresponding $u_{ij} = 1$. However, the update of $\{c_j\}$ differs significantly from the Euclidean case.

When updating $\{c_j\}$, the constraint $c_j \in \mathcal{M}$, $\mathcal{M} = \otimes_{p=1}^{Q} \mathcal{M}_p$ leads to the Riemannian optimization problem (4), which can typically be addressed using methods such as Riemannian gradient descent.

$$
\begin{cases}
\min\limits_{c_j} J_{KM}(c_j) = \sum_{i=1}^{N} \sum_{j=1}^{C} u_{ij} d^2(x_i, c_j) = \sum_{i=1}^{N} \sum_{j=1}^{C} \sum_{p=1}^{Q} u_{ij} d_p^2(x_i^p, c_j^p) \\
\text{s.t.} \quad c_j \in \mathcal{M}, \quad \mathcal{M} = \otimes_{p=1}^{Q} \mathcal{M}_p, \quad \forall j = 1, \ldots, C
\end{cases}
\tag{4}
$$

This is the well-known problem of finding Fréchet means (Iao et al., 2025; Wu & Pan, 2025a) on a manifold. In general, closed-form solutions do not exist (Capitaine et al., 2024), it's the fundamental difference from the flat Euclidean spaces. It is also means that the naive extension of Fuzzy K-Means to manifolds also requires computing the Fréchet centers, which entails the same time complexity. This **highlights** that our proposed RFK algorithm is not a naive extension of Fuzzy K-Means.

This approach to performing K-Means clustering on product manifolds is referred to as Naive Riemannian K-Means (NRK). Analyzing this algorithm, it is not difficult to see that if computing the Fréchet mean in each iteration requires Riemannian optimization with time complexity $\mathcal{O}(\nu)$ (Lou et al., 2020), and the clustering process involves $\mathcal{O}(\omega)$ alternating updates of $\{u_{ij}\}$ and $\{c_j\}$, then the total time complexity is $\mathcal{O}(\nu\omega)$. Since both $\nu$ and $\omega$ are typically large, clustering becomes unacceptable for large-scale data. Therefore, reducing the time complexity is of critical importance.

## 3.2 RIEMANNIAN FUZZY K-MEANS

From the above analysis, it is clear that due to the constraint $c_j \in \mathcal{M}$, $\mathcal{M} = \otimes_{p=1}^{Q} \mathcal{M}_p$, Riemannian optimization is unavoidable. Therefore, if we aim to reduce the overall complexity, the only viable approach is to reconsider the treatment of $\{u_{ij}\}$.

If a smooth mapping $u_{ij} = f(c_j)$ can be found, such that $J_{KM}$ becomes a differentiable function of $c_j$, then alternating optimization can be avoided entirely. However, for standard K-Means, this is not possible. The update rule for $u_{ij}$ is inherently non-smooth and discrete:

$$
u_{ij} = \begin{cases}
1, & j = \text{argmin}_{j \in \{1, \ldots, C\}} \sum_{p=1}^{Q} d_p^2(x_i^p, c_j^p), \\
0, & \text{otherwise.}
\end{cases}
\tag{5}
$$

To address this issue, we adopt the relaxed version of K-Means, Fuzzy K-Means, whose optimization objective is given by:

$$
\begin{cases}
\min\limits_{c_j, u_{ij}} J_{FK}(u_{ij}, c_j) = \sum_{i=1}^{N} \sum_{j=1}^{C} u_{ij}^m d^2(x_i, c_j) = \sum_{i=1}^{N} \sum_{j=1}^{C} \sum_{p=1}^{Q} u_{ij}^m d_p^2(x_i^p, c_j^p) \\
\text{s.t.} \sum_{j=1}^{C} u_{ij} = 1, u_{ij} \geq 0, \quad \forall i = 1, \ldots, N, \forall j = 1, \ldots, C \\
\text{s.t.} \quad c_j \in \mathcal{M}, \ \mathcal{M} = \otimes_{p=1}^{Q} \mathcal{M}_p, \quad \forall j = 1, \ldots, C
\end{cases}
\tag{6}
$$

For fixed $\{c_j\}$, the optimal memberships $u_{ij}$ are given in closed form by:

$$u_{ij}(c_j) = \operatorname*{argmin}_{u_{ij} \geq 0, \sum_{j=1}^{C} u_{ij} = 1} \left( \sum_{i=1}^{N} \sum_{j=1}^{C} \sum_{p=1}^{Q} u_{ij}^m d_p^2(x_i^p, c_j^p) \right) = \left( \sum_{k=1}^{C} \left( \frac{\sum_{p=1}^{Q} d_p^2(x_i^p, c_j^p)}{\sum_{p=1}^{Q} d_p^2(x_i^p, c_k^p)} \right)^{\frac{1}{m-1}} \right)^{-1}, \quad (7)$$

By substituting $u_{ij}(c_j)$ into $J_{FK}$, the objective function $J_{FK}$ can be expressed as an optimization problem depending solely on $\{c_j\}$, specifically:

$$J_{FK}(u_{ij}(c_j), c_j) = \sum_{i=1}^{N} \sum_{j=1}^{C} \underbrace{\left[ \sum_{k=1}^{C} \left( \frac{\sum_{p=1}^{Q} d_p^2(x_i^p, c_j^p)}{\sum_{p=1}^{Q} d_p^2(x_i^p, c_k^p)} \right)^{\frac{1}{m-1}} \right]^{-m} \sum_{p=1}^{Q} d_p^2(x_i^p, c_j^p)}_{A_1}$$

$$= \sum_{i=1}^{N} \sum_{j=1}^{C} \left( \left( \sum_{p=1}^{Q} d_p^2(x_i^p, c_j^p) \right)^{\frac{1}{m-1}} S_i \right)^{-m} \sum_{p=1}^{Q} d_p^2(x_i^p, c_j^p) = \sum_{i=1}^{N} \sum_{j=1}^{C} \left( \sum_{p=1}^{Q} d_p^2(x_i^p, c_j^p) \right)^{-\frac{1}{m-1}} S_i^{-m} \quad (8)$$

$$= \sum_{i=1}^{N} S_i^{-m} \sum_{j=1}^{C} \left( \sum_{p=1}^{Q} d_p^2(x_i^p, c_j^p) \right)^{-\frac{1}{m-1}} = \sum_{i=1}^{N} S_i^{1-m} = \sum_{i=1}^{N} \underbrace{\left( \sum_{j=1}^{C} \left( \sum_{p=1}^{Q} d_p^2(x_i^p, c_j^p) \right)^{-\frac{1}{m-1}} \right)^{1-m}}_{A_2}.$$

Let $S_i = \sum_{j=1}^{C} \left( \sum_{p=1}^{Q} d_p^2(x_i^p, c_j^p) \right)^{-\frac{1}{m-1}}$ be an intermediate variable introduced during simplification. By simplifying to form $A_2$, the objective function $J_{FK}$ is expressed solely in terms of $\{c_j\}$. This enables Riemannian optimization to be performed directly on $\{c_j\}$, without alternating between $\{u_{ij}\}$ and $\{c_j\}$.

It is important to note that the simplification from $A_1$ to $A_2$ is necessary, because computing the gradient of Equation (4) requires evaluating a triple sum, while differentiating $A_1$ involves a quadruple sum. Only by converting to the $A_2$ form, also involving a triple sum, can we ensure that this step does not introduce additional computational cost.

Analyze the time complexity of optimizing $J_{FK}$: since the time complexity of taking the derivative of Equation (4) and that of $A_2$ are the same (both have closed-form solutions), and operations such as computing the Riemannian gradient during the optimization process also have identical complexity, while Equation (4) requires $\omega$ times alternating updates between $\{u_{ij}\}$ and $\{c_j\}$, $A_2$ only requires one optimization. Therefore, we have successfully reduced the time complexity from $\mathcal{O}(\nu\omega)$ to $\mathcal{O}(\nu)$.

Specifically, when the distance on the product manifold is replaced by the distance on the manifold $\mathcal{M}$, $A_2$ can be further simplified as Equation (9). **For convenience, we will also use the notation in Equation** (9) **in the following sections.**

$$J_{FK}(u_{ij}(c_j), c_j) = \sum_{i=1}^{N} \left( \sum_{j=1}^{C} d(x_i, c_j)^{-\frac{2}{m-1}} \right)^{1-m}, d(x_i, c_j) = \sqrt{\sum_{p=1}^{Q} d_p^2(x_i^p, c_j^p)}, c_j \in \mathcal{M} = \otimes_{p=1}^{Q} \mathcal{M}_p$$

$$(9)$$

## 3.3 RADAN ON PRODUCT MANIFOLDS

To further accelerate the RFK algorithm, we modify the Adan optimizer (Xie et al., 2024) and adapt it to product manifolds. Adan is an algorithm that incorporates Nesterov acceleration (Zhou et al., 2024) into adaptive optimization (Yue et al., 2021). We expect that this type of Nesterov method can also be effective for optimization on product manifolds.

For Adan, we adopt a standard modification strategy (Boumal, 2023). Our adaptation of Adan consists of three main components: updating momentum via parallel transport, maintaining the second-order moment as a scalar, and performing updates using the exponential map. Specifically, let Riemannian Adan at the $t$-th iteration involve parameters $\{g_t^p, m_t^p, v_t^p, z_t^p, n_t^p, u_t^p, \alpha_t^p\}, p \in \{1, \ldots, Q\}$, where $g$ denotes the Riemannian gradient, $m$ the momentum, $v$ an estimate of the Riemannian gradient difference, $z$ and $n$ the estimations of the second-order moment of the gradient, $u$ the update direction, and $\alpha$ the learning rate, with $p$ indicating the component on the $p$-th manifold $\mathcal{M}_p$. During the update

of $m_{t+1}^p$, we apply parallel transport, i.e., $m_t^p = \beta_{1t}^p \varphi^p(m_{t-1}^p) + (1 - \beta_{1t}^p)g_t^p$, similar updates are applied to all other vector-based quantities involving subtraction. For the scalar maintenance of $n_t^p$, we use the update $n_t^p = \beta_{3t}^p n_{t-1}^p + (1 - \beta_{3t}^p)\|z_t^p\|_{y_t^p}^2$. Finally, the parameter update is conducted via the exponential map: $y_{t+1}^p = \mathrm{Exp}^p(-\alpha_t^p u_t^p)$.

$$
\begin{cases}
m_t = \beta_{1t}m_{t-1} + (1-\beta_{1t})g_t \\
v_t = \beta_{2t}v_{t-1} + (1-\beta_{2t})(g_t - g_{t-1}) \\
z_t = g_t + \beta_{2t}(g_t - g_{t-1}) \\
n_t = \beta_{3}n_{t-1} + (1-\beta_{3})(z_t \odot z_t) \\
u_t = m_t + \beta_{2t}v_t \\
\alpha_t = \dfrac{\eta_t}{\sqrt{n_t}} + \epsilon_t \\
y_{t+1} = y_t - \alpha_t u_t
\end{cases}
\qquad
\begin{cases}
m_t = \beta_{1t}\varphi(m_{t-1}) + (1-\beta_{1t})g_t \\
v_t = \beta_{2t}\varphi(v_{t-1}) + (1-\beta_{2t})(g_t - \varphi(g_{t-1})) \\
z_t = g_t + \beta_{2t}(g_t - \varphi(g_{t-1})) \\
n_t = \beta_{3t}n_{t-1} + (1-\beta_{3t})\|z_t\|_{y_t}^2 \\
u_t = m_t + \beta_{2t}v_t \\
\alpha_t = \dfrac{\eta_t}{\sqrt{n_t}} + \epsilon_t \\
y_{t+1} = \mathrm{Exp}_{y_t}(-\alpha_t u_t)
\end{cases}
\qquad
\begin{cases}
m_t = \beta_{1}\varphi(m_{t-1}) + (1-\beta_{1t})g_t \\[4pt]
v_t = \beta_{2}\varphi(v_{t-1}) + (1-\beta_{2})\|g_t\|_{y_t}^2 \\[4pt]
u_t = m_t \\[4pt]
\alpha_t = \dfrac{\eta_t}{\sqrt{v_t}} + \epsilon_t \\[4pt]
y_{t+1} = \mathrm{Exp}_{y_t}(-\alpha_t u_t)
\end{cases}
$$

(a) Adan optimizer      (b) Radan optimizer      (c) Radam optimizer

Figure 2: Update Process Illustration of Adan, Radan, and Radam Optimizers.

Figure 2 presents the update details of Adan, Radan, and Radam on the product manifold, using the simplified notation from Equation (9). Here, $m_t = (m_t^1, \ldots, m_t^Q)$, $\beta_{1t} = (\beta_{1t}^1, \ldots, \beta_{1t}^Q)$, and other variables are similarly updated on each component manifold.

To characterize its local convergence rate, We adopt a standard approach by analyzing the algorithm in a region where geodesic convexity holds, as the vicinity of a local minimum is guaranteed to be geodesically convex under standard second-order optimality conditions (Boumal, 2023). In this setting, we assume the product manifold $\mathcal{M}$ is bounded by a diameter $D_\infty$ and has a curvature function $\zeta(\kappa, c)$. This is also a common assumption in the literature (Becigneul & Ganea, 2019).

**Theorem 3.1.** *Let $y_t$ be the sequence generated by the Radan algorithm. Under the standard assumptions, the regret bound $R_T$ satisfies the following. The proof is in Appendix A.1.*

$$
\begin{aligned}
R_T \leq{} & \frac{\zeta(\kappa, c) \cdot (3 - 2\beta_1)\eta G^2 \sqrt{T}\sqrt{1 + \log T}}{2(1 - \beta_1)^3} + \frac{2\eta\beta_1 G}{(1 - \beta_1)^3}\sqrt{T} \\
& + \frac{GD_\infty^2(1 + 2\beta_2)\sqrt{T}}{2(1 - \beta_1) \cdot \eta} + \sum_{t=1}^{T} \frac{4D_\infty^2 G^2 \beta_{2t}}{1 - \beta_1} + \sum_{t=1}^{T} \frac{\sqrt{t}(1 + 2\beta_2)GD_\infty^2 \beta_{1t}}{\eta(1 - \beta_1)}
\end{aligned}
\tag{10}
$$

**Theorem 3.2.** *In the bound Equation (10), any non-summation term $K(T)$ satisfies $\mathbf{o}\left(\frac{K(T)}{T}\right) = 0$. For the summation terms, as long as the parameter decay conditions $\mathbf{o}\left(\frac{\sum_{t=1}^{T} \beta_{1t}\sqrt{t}}{T}\right) = 0$, $\mathbf{o}\left(\frac{\sum_{t=1}^{T} \beta_{2t}}{T}\right) = 0$ and $\beta_{3t} = 1 - \frac{1}{t}$ are met, Radan converges to the optimum. Here, $\mathbf{o}(\cdot)$ represent asymptotically vanishing terms. The proof is in Appendix A.2.*

While our convergence proof requires decaying $\beta$, our experiments adopt the standard practice of using fixed values for their proven empirical effectiveness and simplicity (Becigneul & Ganea, 2019; Kochurov et al., 2020). By optimizing Equation (9), we obtain the final cluster centers $\{c_j\}$ upon completion. Then, by applying Equation (7), we compute the assignment results $\{u_{ij}\}$, completing the clustering process.

### 3.4 CALCULATE RIEMANNIAN GRADIENT

During the Riemannian optimization process, it is also necessary to compute the Riemannian gradient. Below, we provide the expressions for the Riemannian gradient on three constant curvature manifolds: Euclidean space, hyperspherical manifold, and hyperbolic space.

**Theorem 3.3.** *On a single constant-curvature manifold $\mathbb{R}^r$, $\mathbb{S}^{s,K}$, or $\mathbb{H}^{h,K}$, the Riemannian gradient of the Riemannian Fuzzy K-Means objective function $J_{FK}$ with respect to the cluster center $c_k$ is uniformly expressed as:*

$$
\mathrm{grad}_{c_k} J_{FK} = -2 \sum_{i=1}^{N} S_i^{-m} d(x_i, c_k)^{-\frac{2m}{m-1}} Log_{c_k}(x_i),
\tag{11}
$$

*where $Log_{c_k}(x_i)$ denotes the logarithmic map of point $x_i$ at $c_k$. The $Log_{c_k}(x_i)$ on three types of constant-curvature manifolds are given as follows. The proof is in Appendix A.3.*

$$Log_c(x) = \begin{cases} x - c, & \text{if } x, c \in \mathbb{R}^r, \\ \dfrac{\theta}{\sin(\theta)} \left( x - \cos(\theta)\, c \right), \quad \theta = \cos^{-1}(K^2 \langle c, x \rangle), & \text{if } x, c \in \mathbb{S}^{s,K}, \\ \dfrac{\theta}{\sinh(\theta)} \left( x + K^2 \langle c, x \rangle_h\, c \right), \quad \theta = \cosh^{-1}(K^2 \langle c, x \rangle_h), & \text{if } x, c \in \mathbb{H}^{h,K}. \end{cases} \tag{12}$$

After computing according to Equation (11), the expression of the Riemannian gradient can be obtained. By combining the Riemannian gradient with the corresponding logarithmic map, exponential map, and other operations on different manifolds, all steps of the Riemannian optimization to solve RFK can be completed. Thereafter, we conduct extensive experiments on the RFK and Radan algorithms to validate their speed and superior performance.

## 4 EXPERIMENTS

In this section, we conducted extensive experiments aiming to answer the following three questions:

- Q1: How much faster is the RFK algorithm compared to the NRF algorithm when run on product manifolds? Does it achieve a lower loss value?
- Q2: When running Radan on product manifolds, does it accelerate the RFK algorithm compared to Radam with standard hyperparameters?
- Q3: Compared to the current state-of-the-art clustering algorithms, can RFK demonstrate better advantages for data represented on product manifolds?

We also provide several sensitivity analyses, including those on the fuzziness index $m$, the number of cluster centers, and other key hyperparameters in the Appendix F.2.

### 4.1 DATASETS

The datasets on product manifolds include four parts: synthetic data, graph embedding data and mixed-curvature VAE latent space data. More details are in Table 5.

**Synthetic Data:** We use the *'gaussian mixture'* function from Manify (Chlenski et al., 2025b) to generate data with 3 clusters on different product manifolds, and generate a set of labels for clustering.

**Graph Embedding Data:** For the graph embedding data, it is divided into two parts. One part selects the optimal embedding from $\left\{ \left(\mathbb{H}^2\right)^2, \, \mathbb{H}^2\mathbb{E}^2, \, \mathbb{H}^2\mathbb{S}^2, \, \mathbb{S}^2\mathbb{E}^2, \, \left(\mathbb{S}^2\right)^2, \, \mathbb{H}^4, \, \mathbb{E}^4, \, \mathbb{S}^4 \right\}$ by means of curvature estimation (Gu et al., 2018). The other part embeds the data into the 2D Poincaré disk $\mathbb{D}$.

**Mixed-curvature VAE Latent Space:** We use data from the latent space of a mixed-curvature variational autoencoder (Skopek et al., 2020) as the datasets, including the MNIST with over 600,000 samples. These product manifold representations are derived from the Manify (Chlenski et al., 2025b).

We emphasize that the data already lying on the manifolds are the actual data we use, without requiring any additional preprocessing.

### 4.2 EXPERIMENTS SETUP

#### 4.2.1 EXPERIMENT SETUP FOR Q1

To verify that our RFK algorithm is faster than NRK, we ran both algorithms on the aforementioned datasets and recorded their execution times. To ensure fair timing comparisons, we replaced non-vectorized operations with matrix-based implementations for NRK (see Equation (7)). For the optimization part, we used the proposed Radan optimizer for both methods, with parameters set as {Radan: $\beta_1^p = 0.7$, $\beta_2^p = 0.99$, $\beta_3^p = 0.99$}, and a common learning rate of $0.5$ for testing. For RFK, the stopping criterion for Radan was that the change in loss between iteration $t$ and $t + 1$ was less than $1e - 4$. For NRK, there are two convergence criteria: the condition for updating the Fréchet mean is

Table 1: RFK & NRK Time (s) and Cost on Datasets, OT means out-of-time

| Method | Gauss $\mathbb{R}^4$ | | Gauss $\mathbb{H}^4$ | | Gauss $\mathbb{S}^2\mathbb{H}^2$ | | Gauss $\mathbb{R}^2\mathbb{S}^2\mathbb{H}^2$ | | Gauss $\mathbb{S}^2(\mathbb{H}^2)^2$ | | Gauss $\mathbb{R}^4\mathbb{S}^4\mathbb{H}^4$ | | Gauss $\mathbb{R}^{16}\mathbb{S}^{16}\mathbb{H}^{16}$ | | CiteSeer | |
|---|---|---|---|---|---|---|---|---|---|---|---|---|---|---|---|---|
| | Time | Loss | Time | Loss | Time | Loss | Time | Loss | Time | Loss | Time | Loss | Time | Loss | Time | Loss |
| RFK | **0.07** | **1499.84** | **0.21** | **1832.57** | **0.19** | **791.87** | **0.23** | **1569.92** | **0.45** | **1518.82** | **0.28** | **3549.86** | **0.25** | **35869.52** | **1.02** | **17.77** |
| NRK | 0.60 | 1451.24 | 36.27 | 1845.32 | 2.37 | 791.87 | 6.12 | 1569.92 | 63.90 | 1518.82 | 2.78 | 3549.86 | 0.82 | 35869.54 | 52.23 | 17.89 |
| Method | Cora | | PolBlogs | | Olsson | | Paul | | PoolBooks | | CIFAR-100 | | Lymphoma | | MNIST | |
| | Time | Loss | Time | Loss | Time | Loss | Time | Loss | Time | Loss | Time | Loss | Time | Loss | Time | Loss |
| RFK | **0.29** | **16.00** | **0.13** | **39.54** | **0.17** | **65.88** | **0.25** | **88.06** | **0.07** | **127.00** | **67.28** | **46450.93** | **3.61** | **878.25** | **82.76** | **668268.50** |
| NRK | 68.46 | 16.01 | 0.44 | 39.54 | 4.82 | 65.88 | 606.23 | 88.47 | 2.40 | 127.01 | OT | OT | OT | OT | OT | OT |

the same as in RFK, while the global convergence condition is that the distance between the Fréchet centers of two consecutive iterations is less than $1e-4$.

### 4.2.2 EXPERIMENT SETUP FOR Q2

To evaluate the optimization capabilities of Radan and Radam on the RFK loss function, we designed Experiment 2, where both optimizers adopt their standard parameter settings: {Radan: $\beta_1^p = 0.7$, $\beta_2^p = 0.99$, $\beta_3^p = 0.99$}, {Radam: $\beta_1^p = 0.99$, $\beta_2^p = 0.999$}. We trained using a range of learning rates $\{0.1, 0.3, 0.5, 0.7, 1\}$, comparing the minimum and last values of the mean RFK loss under different learning rates. Each optimizer was run for 300 iterations. Notably, we use standard hyperparameters since adaptive optimizers are considered insensitive to them (Gkouti et al., 2024), and we aim to spare users from tuning when applying RFK.

### 4.2.3 EXPERIMENT SETUP FOR Q3

To compare the clustering performance of the RFK algorithm, we evaluated it on the above datasets against 10 competitive algorithms (Hu et al., 2023; Abdullah et al., 2024; Nie et al., 2024; Zhong & Pun, 2021; Chen et al., 2017; Huang et al., 2019; Nie et al., 2023; Liu et al., 2012; Elhamifar & Vidal, 2013), using five metrics: ACC (Yuan et al., 2025a; Wang et al., 2025), NMI (Xie et al., 2025), ARI (Yuan et al., 2024), F1 (Du et al., 2024), and Purity (Huang et al., 2024). RFK was optimized by Radan. Detailed experimental settings are in Appendix E.2.

### 4.3 EXPERIMENTS RESULT

### 4.3.1 EXPERIMENT RESULT FOR Q1

Table 1 presents the runtime and final loss of the RFK and NRK algorithms. As shown, RFK achieves speedups of over 100× compared to NRK on some datasets. On certain large-scale datasets, NRK runs out of time. Although RFK and NRK optimize the same objective, RFK generally attains a lower final loss. Figure 3 shows the loss curves of RFK and NRK. The NRK curves exhibit step-like drops due to alternating updates of the assignment and the Fréchet center, whereas the RFK curves decrease more smoothly, require significantly fewer iterations, and converge to a lower final value.

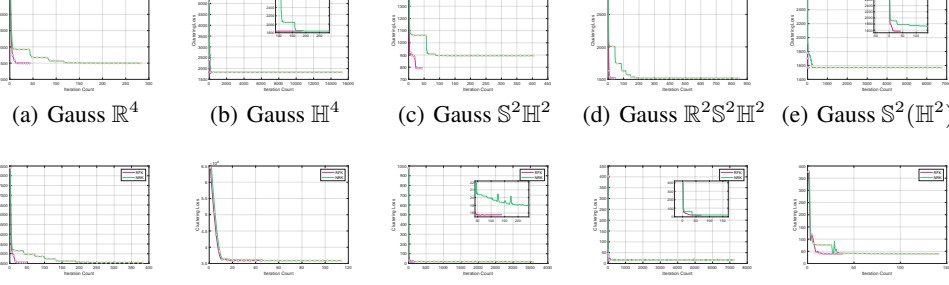

(a) Gauss $\mathbb{R}^4$  (b) Gauss $\mathbb{H}^4$  (c) Gauss $\mathbb{S}^2\mathbb{H}^2$  (d) Gauss $\mathbb{R}^2\mathbb{S}^2\mathbb{H}^2$  (e) Gauss $\mathbb{S}^2(\mathbb{H}^2)^2$

(f) Gauss $\mathbb{R}^4\mathbb{S}^4\mathbb{H}^4$  (g) Gauss $\mathbb{R}^{16}\mathbb{S}^{16}\mathbb{H}^{16}$  (h) CiteSeer  (i) Cora  (j) PolBlogs

Figure 3: Clustering loss curves for RFK and NRK

### 4.3.2 EXPERIMENT RESULT FOR Q2

Table 2 presents the average loss reduction results using the Radan and Radam optimizers. It can be seen that Radan generally achieves lower loss values than Radam. Figure 4 shows the loss curves,

Table 2: Radan & Radam Min and Last Loss on Various Datasets, OT means out-of-time

| Method | Gauss $\mathbb{R}^4$ | | Gauss $\mathbb{H}^4$ | | Gauss $\mathbb{S}^2\mathbb{H}^2$ | | Gauss $\mathbb{R}^2\mathbb{S}^2\mathbb{H}^2$ | | Gauss $\mathbb{S}^2(\mathbb{H}^2)^2$ | | Gauss $\mathbb{R}^4\mathbb{S}^4\mathbb{H}^4$ | | Gauss $\mathbb{R}^{16}\mathbb{S}^{16}\mathbb{H}^{16}$ | | CiteSeer | |
|---|---|---|---|---|---|---|---|---|---|---|---|---|---|---|---|---|
| | Min | Last | Min | Last | Min | Last | Min | Last | Min | Last | Min | Last | Min | Last | Min | Last |
| Radan | **1499.84** | **1499.84** | **1832.57** | **1832.58** | **791.87** | **792.61** | **1518.82** | **1518.89** | **1569.92** | **1569.98** | **3459.86** | **3459.86** | **35869.50** | **35869.50** | **18.61** | **18.62** |
| Radam | 1533.27 | 1533.27 | 2016.92 | 2016.92 | 814.98 | 814.98 | 1544.57 | 1546.15 | 1619.51 | 1621.24 | 3627.76 | 3627.76 | 36047.06 | 36090.44 | 39.35 | 41.45 |

| Method | Cora | | PolBlogs | | Olsson | | Paul | | PoolBooks | | CIFAR-100 | | Lymphoma | | MNIST | |
|---|---|---|---|---|---|---|---|---|---|---|---|---|---|---|---|---|
| | Min | Last | Min | Last | Min | Last | Min | Last | Min | Last | Min | Last | Min | Last | Min | Last |
| Radan | **17.23** | **17.23** | **39.60** | **39.74** | 66.92 | 66.93 | 83.95 | 84.11 | **126.65** | **126.71** | **48850.73** | 49127.77 | **878.26** | **878.27** | **662611.31** | 667403.93 |
| Radam | 36.60 | 44.79 | 61.43 | 64.19 | **66.08** | **66.08** | **80.70** | **80.70** | 126.86 | 126.86 | 70860.14 | 71248.29 | 3381.86 | 3716.99 | 746378.43 | 727619.73 |

Table 3: ACC for all benchmarks. OT means out-of-time

| | Dataset | Signature | RFK | NRK | K-Means | Ncut | FCM | UFCM | LRR | SSC | SBMC | USPEC | Fast-CD |
|---|---|---|---|---|---|---|---|---|---|---|---|---|---|
| Synthetic | Gaussian | $\mathbb{R}^4$ | 96.00 | 95.40 | 96.02 | **99.00** | 96.00 | 96.10 | 88.80 | 37.30 | 78.05 | 73.60 | **97.20** |
| | | $\mathbb{H}^4$ | **99.80** | 99.00 | 40.42 | 99.10 | 55.40 | 39.30 | **99.80** | 47.80 | 60.30 | 90.41 | 66.50 |
| | | $\mathbb{S}^2\mathbb{H}^2$ | **95.20** | 94.80 | 83.91 | 88.00 | 87.70 | 87.30 | **97.70** | 41.20 | 68.46 | 74.00 | 87.70 |
| | | $\mathbb{R}^2\mathbb{S}^2\mathbb{H}^2$ | **96.20** | 95.80 | 61.20 | **99.40** | 75.80 | 60.36 | 93.70 | 45.10 | 82.07 | 46.22 | 86 |
| | | $\mathbb{S}^2(\mathbb{H}^2)^2$ | **97.80** | **97.80** | 44.23 | 77.05 | 47.70 | 44.67 | 96.60 | 43.60 | 73.21 | 60.37 | 61.20 |
| | | $\mathbb{R}^4\mathbb{S}^4\mathbb{H}^4$ | **99.10** | **98.90** | 39.55 | 68.40 | 64.70 | 41.50 | 97.20 | 38.70 | 87.17 | 62.21 | 95.90 |
| | | $\mathbb{R}^{16}\mathbb{S}^{16}\mathbb{H}^{16}$ | **98.00** | **77.10** | 37.98 | 76.30 | 40.50 | 37.71 | 38.60 | 37.40 | 55.38 | 63.93 | 53.50 |
| Graph | CiteSeer | $(\mathbb{H}^2)^2$ | **25.36** | 20.09 | 20.80 | 24.91 | 20.05 | 21.60 | 19.86 | **25.35** | 19.81 | 23.92 | 20.62 |
| | Cora | $\mathbb{H}^4$ | **29.22** | 18.19 | 18.06 | 20.57 | 18.27 | 18.83 | 18.15 | **29.10** | 17.08 | 20.00 | 18.19 |
| | PolBlogs | $(\mathbb{S}^2)^2$ | **94.36** | 93.62 | 93.90 | 54.66 | 93.54 | **94.01** | 68.66 | 51.96 | 54.65 | 59.16 | 93.70 |
| | Olsson | $\mathbb{D}$ | **67.72** | **67.45** | 61.57 | 60.24 | 60.37 | 60.71 | 44.16 | 60.73 | 51.31 | 57.25 | 60.21 |
| | Paul | $\mathbb{D}$ | **52.73** | **48.15** | 47.05 | 45.47 | 46.57 | 46.86 | 22.94 | 13.88 | 26.01 | 46.00 | 44.03 |
| | PolBooks | $\mathbb{D}$ | **81.90** | **68.57** | 39.68 | 34.27 | 36.34 | 42.44 | OT | OT | 8.81 | 44.12 | 36.12 |
| VAE | CIFAR-100 | $(\mathbb{H}^2)^4$ | **71.19** | OT | 5.75 | OT | 6.00 | 5.53 | OT | OT | OT | 5.21 | OT |
| | Lymphoma | $(\mathbb{S}^2)^2$ | **100.00** | OT | 78.28 | OT | 78.28 | OT | OT | OT | OT | 78.28 | OT |
| | MNIST | $\mathbb{S}^2\mathbb{E}^2\mathbb{H}^2$ | **96.09** | OT | 12.09 | OT | 15.40 | 13.01 | OT | OT | OT | 11.42 | OT |

with the red curve representing the mean loss of Radan and the blue representing Radam. The shaded areas indicate variance. Radan consistently converges faster than Radam, typically within 50–100 iterations, whereas Radam requires around 300 iterations. Additionally, Radan generally achieves lower final loss values.

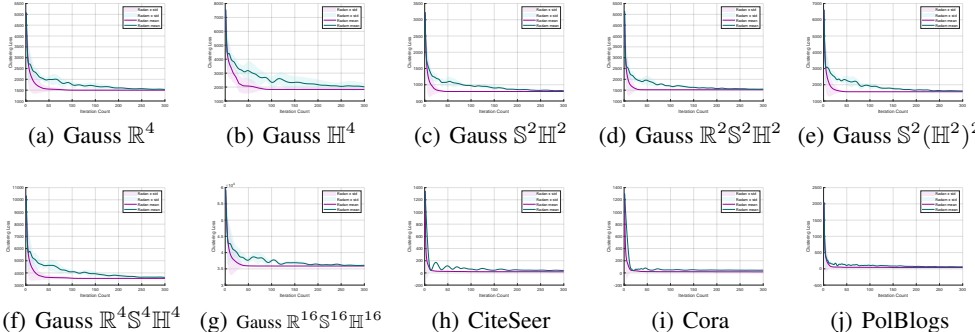

(a) Gauss $\mathbb{R}^4$    (b) Gauss $\mathbb{H}^4$    (c) Gauss $\mathbb{S}^2\mathbb{H}^2$    (d) Gauss $\mathbb{R}^2\mathbb{S}^2\mathbb{H}^2$    (e) Gauss $\mathbb{S}^2(\mathbb{H}^2)^2$

(f) Gauss $\mathbb{R}^4\mathbb{S}^4\mathbb{H}^4$    (g) Gauss $\mathbb{R}^{16}\mathbb{S}^{16}\mathbb{H}^{16}$    (h) CiteSeer    (i) Cora    (j) PolBlogs

Figure 4: Clustering loss curves for Radan and Radam

### 4.3.3 EXPERIMENT RESULT FOR Q3

Table 3 presents the ACC metric of different clustering algorithms across various datasets. The Dataset column lists all the datasets used, and Signature indicates the geometric structure of each dataset. RFK is our proposed algorithm. As shown in the table, our method achieves the best performance on nearly every dataset. In particular, for the MNIST dataset with 600,000 data points, most clustering algorithms fail to produce results; K-Means achieves only about **12%** accuracy, whereas RFK reaches **96.09%** accuracy, which is a remarkable outcome. This result is reasonable because MNIST is well represented in non-Euclidean space, where existing algorithms cannot respect the intrinsic geometric structure, while RFK effectively operates in non-Euclidean space, yielding this impressive performance. Other results can be found in Appendix F.1. Here, OT denotes out-of-time. All experiments were run on an Intel(R) Core(TM) i5-10200H CPU @ 2.40 GHz, with a predefined time limit of 3600 seconds (1 hour). Any algorithm that fails to converge within this time window is marked as out-of-time (OT).

## 5 LIMITATIONS

We acknowledge that this work still has several limitations that warrant further investigation. First, our theoretical assumptions rely on geodesic convexity, meaning that the convergence analysis of Radan focuses on its behavior in a neighborhood around a local optimum. In future work, we aim to establish convergence guarantees under more general conditions. Second, our analysis of Radan's convergence relies on a decaying learning rate, whereas our experiments use a fixed learning rate. Although this is a common practice in Riemannian adaptive optimization, we plan to explore how to bridge this gap. Finally, Riemannian Fuzzy K-Means requires access to closed-form geodesic distance formulas for the manifolds on which it operates. While these formulas are known for commonly used manifolds such as spheres, hyperbolic spaces, and their product manifolds, future work will investigate how to extend our method to manifolds whose geodesic distances lack closed-form expressions.

## 6 CONCLUSION

In this paper, we address an open problem and propose the RFK algorithm, which reduces the time complexity from $\mathcal{O}(\nu\omega)$ to $\mathcal{O}(\nu)$. Furthermore, we introduce Radan as an optimizer for product manifolds. Extensive experiments demonstrate that our algorithm achieves remarkable performance: on some certain datasets, it runs over 100 times faster than NRK while achieving better clustering results and lower loss values. Additionally, Radan converges faster than Radam under the RFK loss with standard hyperparameters. Across almost all datasets, RFK significantly outperforms other state-of-the-art clustering algorithms in all clustering metrics.

## 7 STATEMENT

For the reproducibility of this paper, we have submitted the complete anonymized code, data, and experiment files with fixed random seeds, as detailed in Appendix G. In addition, large language models (LLMs) were only used for language polishing.

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

CONTENTS

## A   PROOFS OF THEOREMS

### A.1   PROOF OF THEOREM 3.1

In this section, we will prove the Theorem 3.1, which provides the regret bound for the Radan algorithm. Before proceeding, we make the following assumption. **These assumptions are all standard assumptions in Riemannian optimization.**

#### A.1.1   ASSUMPTIONS

Assumption 1. In the optimization problem solved by the Radan algorithm, the feasible domain is geodesically bounded (Fujioka, 2022). That is, for any geodesic $\gamma(t)$ within the feasible domain $D$, its length satisfies that:

$$\int_{t_0}^{\infty} \|\gamma(t)\|_{y_t}\, dt \leq D_{\infty} \tag{13}$$

Furthermore, let $Log$ denote the logarithmic map. Then we have the inequality

$$\|Log_{y_t}(y)\|_{y_t} \leq D_{\infty} \tag{14}$$

which we state as Lemma 1, and will prove later in the paper.

Assumption 2. We assume that in the Riemannian optimization problem solved by the Radan algorithm, the curvature $\zeta$ of the Riemannian manifold on which the constraints are defined is bounded. Specifically, in the Riemannian cosine law (Arnaudon & Nielsen, 2013):

$$d^2(y_{t+1}, y^*) \leq d^2(y_{t+1}, y_t) \cdot \zeta(\kappa, c) + d^2(y_t, y^*) - 2d(y_{t+1}, y_t)d(y_t, y^*)\cos A \tag{15}$$

For general spherical, hyperbolic, and their product manifolds, the curvature function $\zeta(\kappa, c)$ also admits a unified formulation:

$$\zeta(\kappa, c) = \begin{cases} \dfrac{\sqrt{|\kappa|}\, c}{\tanh\left(\sqrt{|\kappa|}\, c\right)}, & \kappa < 0, \\[2ex] \dfrac{\tan(\sqrt{\kappa}\, c)}{\sqrt{\kappa}\, c}, & \kappa > 0, \end{cases} \tag{16}$$

which is a function of curvature and distance, and is commonly used in the convergence analysis of Riemannian optimization algorithms. Here, $c$ denotes the distance function, and it satisfies $c \leq D_{\infty}$. The function $\zeta(\kappa, c)$ is assumed to be bounded (Becigneul & Ganea, 2019).

Assumption 3. We assume that the gradient is bounded, i.e., the norm of the gradient at $y_t$ satisfies $\|g_t\|_{y_t} \leq G$, which is a standard assumption commonly used in the proof of the theorem.

Assumption 4. Let the parallel transport of the vector $m_{t-1}$ from $y_{t-1}$ to $y_t$ be denoted by $\varphi_{y_{t-1} \to y_t}(m_{t-1})$, which we abbreviate as $\varphi(m_{t-1})$ when there is no ambiguity. We assume that the parallel transport preserves the inner product of the vector, i.e.,

$$\langle m_{t-1}, v_{t-1}\rangle_{y_{t-1}} = \langle \varphi(m_{t-1}), \varphi(v_{t-1})\rangle_{y_t}. \tag{17}$$

Assumption 5. We assume that in the Riemannian optimization problem solved by the Radan algorithm, the objective function is geodesically convex (Alimisis & Vandereycken, 2024). That is, for any $p, q \in \mathcal{M}$ and $t \in [0, 1]$, the following holds:

$$f(\gamma(t)) \leq (1 - t)f(p) + tf(q), \tag{18}$$

where $\gamma$ is the geodesic connecting $p$ and $q$. Furthermore, it can be shown that

$$f(y_t) - f(y^*) \leq \langle -g_t, Log_{y_t}(y^*)\rangle_{y_t}, \tag{19}$$

and we will provide a proof of this in Lemma 2.

### A.1.2 PROOF DETAILS

We now present Theorem 3.1 along with its proof.

**Theorem A.1.** *Let $y_t$ be the sequence generated by the Radan algorithm. Under the standard assumptions, the regret bound $R_T$ satisfies the following.*

$$R_T \leq \frac{\eta\sqrt{T}\sqrt{1+logT}G}{(1-\beta_1)^2}\left[\frac{\zeta(\kappa,c)(3-2\beta_1)G}{2(1-\beta_1)} + \frac{\beta_1}{\sqrt{1-\beta_{3t}}(1-\delta)}\right]$$
$$+ \frac{GD_\infty^2(1+2\beta_2)\sqrt{T}}{2(1-\beta_1)\cdot\eta} + \sum_{t=1}^{T}\frac{4D_\infty^2 G^2\beta_{2t}}{1-\beta_1} + \sum_{t=1}^{T}\frac{\sqrt{t}(1+2\beta_2)GD_\infty^2\beta_{1t}}{\eta(1-\beta_1)} \tag{20}$$

*Proof.* According to the Radan algorithm, the update from step $t$ to step $t+1$ is as follows:

$$\begin{cases} m_t = \beta_{1t}\varphi(m_{t-1}) + (1-\beta_{1t})g_t \\ v_t = \beta_{2t}\varphi(v_{t-1}) + (1-\beta_{2t})(g_t - \varphi(g_{t-1})) \\ z_t = g_t + \beta_{2t}(g_t - \varphi(g_{t-1})) \\ n_t = \beta_{3t}n_{t-1} + (1-\beta_{3t})\|z_t\|_{y_t}^2 \\ u_t = m_t + \beta_{2t}v_t \\ \alpha_t = \frac{\eta_t}{\sqrt{n_t}} + \epsilon_t \\ y_{t+1} = Exp_{y_t}(-\alpha_t u_t) \end{cases} \tag{21}$$

Here, $\varphi$ is a parallel translation, which is assumed to preserve the inner product. $Exp(\cdot)$ is the exponential map, and $Log(\cdot)$ is the logarithmic map. Also, $\beta_{1t} = \beta_1 \cdot f_1(t)$, where $f_1(t)$ is a decay function, and $\beta_{2t} = \beta_2 \cdot f_2(t)$, where $f_2(t)$ is a decay function.

Given $y_{t+1} = Exp(-\alpha_t u_t)$, according to the cosine theorem on the manifold, we have:

$$a^2 \leq b^2\zeta(\kappa,c) + c^2 - 2bc\cos A \tag{22}$$

Based on the assumption of bounded curvature, we have that $\zeta(\kappa,c)$ is bounded, let:

$$a = d(y_{t+1}, y^*), \quad b = d(y_{t+1}, y_t), \quad c = d(y_t, y^*). \tag{23}$$

Then , that is:

$$d^2(y_{t+1}, y^*) \leq d^2(y_{t+1}, y_t)\cdot\zeta(\kappa,c) + d^2(y_t, y^*) - 2d(y_{t+1}, y_t)d(y_t, y^*)\cos A \tag{24}$$

According to the definition of $\cos A$, we have:

$$d(y_{t+1}, y_t)d(y_t, y^*)\cos A = \langle Log_{y_t}(y_{t+1}), Log_{y_t}(y^*)\rangle_{y_t} = -\alpha_t\langle u_t, Log_{y_t}(y^*)\rangle_{y_t} \tag{25}$$

Substituting this into the above Equation (24), we get:

$$2d(y_{t+1}, y_t)d(y_t, y^*)\cos A = -2\alpha_t\langle u_t, Log_{y_t}(y^*)\rangle_{y_t}$$
$$\leq d^2(y_t, y^*) - d^2(y_{t+1}, y^*) + \zeta(\kappa,c)\cdot d^2(y_{t+1}, y_t) \tag{26}$$
$$= d^2(y_t, y^*) - d^2(y_{t+1}, y^*) + \zeta(\kappa,c)\alpha_t^2\|u_t\|_{y_t}^2$$

By rearranging the terms and dividing both sides by $\alpha_t$, we obtain the following expression:

$$\langle -u_t, Log_{y_t}(y^*)\rangle_{y_t} \leq \frac{1}{2\alpha_t}[d^2(y_t, y^*) - d^2(y_{t+1}, y^*)] + \frac{\zeta(\kappa,c)\alpha_t\|u_t\|_{y_t}^2}{2} \tag{27}$$

Since $u_t = m_t + \beta_{2t}v_t$, then:

$$\langle -m_t, Log_{y_t}(y^*)\rangle_{y_t} \leq \frac{1}{2\alpha_t}[d^2(y_t, y^*) - d^2(y_{t+1}, y^*)] + \frac{\zeta(\kappa,c)\alpha_t\|u_t\|_{y_t}^2}{2} + \beta_{2t}\langle v_t, Log_{y_t}(y^*)\rangle_{y_t} \tag{28}$$

Also, because $m_t = \beta_{1t}\varphi(m_{t-1}) + (1-\beta_{1t})g_t$, finally we have:

$$\langle -(1-\beta_{1t})g_t, Log_{y_t}(y^*)\rangle_{y_t} \leq \frac{1}{2\alpha_t}[d^2(y_t, y^*) - d^2(y_{t+1}, y^*)] + \frac{\zeta(\kappa,c)\alpha_t\|u_t\|_{y_t}^2}{2}$$
$$+ \beta_{2t}\langle v_t, Log_{y_t}(y^*)\rangle_{y_t} + \beta_{1t}\langle\varphi(m_{t-1}), Log_{y_t}(y^*)\rangle_{y_t} \tag{29}$$

Dividing both sides of the equation by $(1 - \beta_{1t})$, we get the following expression:

$$\langle -g_t, Log_{y_t}(y^*)\rangle_{y_t} \leq \frac{1}{2\alpha_t(1 - \beta_{1t})}[d^2(y_t, y^*) - d^2(y_{t+1}, y^*)] + \frac{\zeta(\kappa, c) \cdot \alpha_t}{2(1 - \beta_{1t})}\|u_t\|_{y_t}^2$$

$$+ \frac{\beta_{2t}}{(1 - \beta_{1t})}\langle v_t, Log_{y_t}(y^*)\rangle_{y_t} + \frac{\beta_{1t}}{1 - \beta_{1t}}\langle \varphi(m_{t-1}), Log_{y_t}(y^*)\rangle_{y_t} \quad (30)$$

Since $f(x)$ is geodesically convex, according to Lemma 2, we have the following:

$$f(y_t) - f(y^*) \leq \langle -g_t, Log_{y_t}(y^*)\rangle_{y_t} \quad (31)$$

The regret bound is:

$$R_T = \sum_{t=1}^{T} f(y_t) - f(y^*) \leq \sum_{t=1}^{T}\langle -g_t, Log_{y_t}(y^*)\rangle_{y_t}$$

$$\leq \underbrace{\sum_{t=1}^{T} \frac{1}{2\alpha_t(1 - \beta_{1t})}[d^2(y_t, y^*) - d^2(y_{t+1}, y^*)]}_{B_1} + \underbrace{\sum_{t=1}^{T} \frac{\zeta(\kappa, c)\alpha_t}{2(1 - \beta_{1t})}\|u_t\|_{y_t}^2}_{B_2} \quad (32)$$

$$+ \underbrace{\sum_{t=1}^{T} \frac{\beta_{2t}}{(1 - \beta_{1t})}\langle v_t, Log_{y_t}(y^*)\rangle_{y_t}}_{B_3} + \underbrace{\sum_{t=1}^{T} \frac{\beta_{1t}}{1 - \beta_{1t}}\langle \varphi(m_{t-1}), Log_{y_t}(y^*)\rangle_{y_t}}_{B_4}$$

For $B_1$: First, we estimate the $B_1$ term and identify its upper bound.

$$B_1 \leq \frac{1}{2(1 - \beta_1)}\left[\sum_{t=1}^{T}\left(\frac{1}{\alpha_t}d^2(y_t, y^*) - \frac{1}{\alpha_t}d^2(y_{t+1}, y^*)\right)\right]$$

$$= \frac{1}{2(1 - \beta_1)}\left[\sum_{t=2}^{T}\left(\frac{1}{\alpha_t} - \frac{1}{\alpha_{t-1}}\right)d^2(y_t, y^*) + \frac{1}{\alpha_1}d^2(y_1, y^*) - \frac{1}{\alpha_T}d^2(y_{T+1}, y^*)\right]$$

$$\leq \frac{1}{2(1 - \beta_1)} \cdot \sum_{t=2}^{T}\left(\frac{1}{\alpha_t} - \frac{1}{\alpha_{t-1}}\right)D_\infty^2 + \frac{1}{\alpha_1}D_\infty^2 \quad (33)$$

$$= \frac{1}{2(1 - \beta_1)} \cdot \frac{1}{\alpha_T}D_\infty^2 - \frac{1}{\alpha_1}D_\infty^2 + \frac{1}{\alpha_1}D_\infty^2$$

$$= \frac{1}{2(1 - \beta_1)\alpha_T}D_\infty^2$$

$$= \frac{D_\infty^2}{2(1 - \beta_1) \cdot \eta_T}\sqrt{n_T} \quad (\text{where } \eta_T = \frac{\eta}{\sqrt{T}})$$

The first inequality follows from the fact that $\beta_{1t} = \beta_1 f_1(t)$, which decays term by term. Therefore, $\frac{1}{1 - \beta_{1t}} \leq \frac{1}{1 - \beta_1}$. The second inequality follows from the assumption that the feasible domain is bounded, i.e.,

$$d(y_t, y^*) \leq \sup_{x \in D} d(x, y^*) \leq D_\infty. \quad (34)$$

The mathematical logic behind the second inequality also includes $\frac{1}{\alpha_t} \geq \frac{1}{\alpha_{t-1}}$, which is derived from $\beta_{3t} = 1 - \frac{1}{t}$.

For $B_4$: Next, we provide the upper bound for the fourth term $B_4$. By Young's inequality (Alzer & Kwong, 2019), after making simple transformations, we can obtain the following expression:

$$\langle \varphi(m_{t-1}), Log_{y_t}(y^*)\rangle_{y_t} \leq \underbrace{\frac{\eta_t}{\sqrt{n_t}}\|\varphi(m_{t-1})\|_{y_t}^2}_{B_{41}} + \underbrace{\frac{\sqrt{n_t}}{\eta_t}\|Log_{y_t}(y^*)\|_{y_t}^2}_{B_{42}} \quad (35)$$

Considering $B_{41}$: Since $\varphi$ preserves the inner product, we can obtain the following equality:

$$\|\varphi(m_{t-1})\|_{y_t}^2 = \langle \varphi(m_{t-1}), \varphi(m_{t-1})\rangle_{y_t} = \langle m_{t-1}, m_{t-1}\rangle_{y_{t-1}} = \|m_{t-1}\|_{y_{t-1}}^2. \quad (36)$$

Furthermore, we can perform an equivalence transformation on $B_{41}$.

$$\sum_{t=1}^{T} \frac{\eta_t}{\sqrt{n_t}} \|\varphi(m_{t-1})\|_{y_t}^2 = \sum_{t=1}^{T} \frac{\eta_t}{\sqrt{n_t}} \|m_{t-1}\|_{y_{t-1}}^2 \tag{37}$$

Since $\|g_{t+1}\|_{y_{t+1}}$ is bounded, it is evident that $\|m_{t+1}\|_{y_{t+1}}$ is also bounded. Therefore, to prove that $B_{41}$ is bounded, it suffices to show that $\sum_{t=1}^{T} \frac{\eta_t}{\sqrt{n_t}} \|m_t\|_{y_t}^2$ is bounded.

Since $m_t = \beta_{1t}\varphi(m_{t-1}) + (1 - \beta_{1t})g_t$, by using the recurrence relation and mathematical induction, it can be proven that:

$$\begin{cases} m_1 = \beta_{11}\varphi(m_0) + (1 - \beta_{11})g_1 = (1 - \beta_{11})g_1 \\ m_2 = \beta_{12}\varphi(m_1) + (1 - \beta_{12})g_2 = \beta_{12}(1 - \beta_{11})\varphi(g_1) + (1 - \beta_{12})g_2 \\ m_3 = \beta_{13}\varphi(m_2) + (1 - \beta_{13})g_3 = \beta_{12}\beta_{13}(1 - \beta_{11})\varphi(g_1) + (1 - \beta_{12})\beta_{13}\varphi(g_2) + (1 - \beta_{13})g_3 \\ \quad \vdots \\ m_t = \sum_{j=1}^{t}(1 - \beta_{1j})\left(\prod_{k=1}^{t-j}\beta_{1,(t-k+1)}\right)\varphi(g_j) \end{cases}$$

$$\tag{38}$$

According to Lemma 3, using the inequality

$$\left\|\sum_{i=1}^{n} a_i p_i\right\|^2 \leq \left(\sum_{i=1}^{n} a_i\right)\left(\sum_{i=1}^{n} a_i\|p_i\|^2\right), \tag{39}$$

we can derive the following.

$$\begin{aligned} \|m_t\|_{y_t}^2 &= \|\sum_{j=1}^{t}(1 - \beta_{1j})\left(\prod_{k=1}^{t-j}\beta_{1,(t-k+1)}\right)\varphi(g_j)\|_{y_t}^2 \\ &\leq \left(\sum_{j=1}^{t}(1 - \beta_{1j})\prod_{k=1}^{t-j}\beta_{1,(t-k+1)}\right)\left(\sum_{j=1}^{t}(1 - \beta_{1j})\prod_{k=1}^{t-j}\beta_{1,(t-k+1)} \cdot \|g_j\|_{y_j}^2\right) \\ &\leq \left(\sum_{j=1}^{t}(1 - \beta_{1j})\beta_1^{t-j}\right)\left(\sum_{j=1}^{t}(1 - \beta_{1j})\beta_1^{t-j}\|g_j\|_{y_j}^2\right) \end{aligned} \tag{40}$$

Using the formula for the sum of a geometric series combined with the fact that $\beta_1 < 1$:

$$1 - \beta_{1j} < 1, \quad \sum_{j=1}^{t}\beta_1^{t-j} = \beta_1^t\sum_{j=1}^{t}\beta_1^{-j} = \beta_1^t \cdot \frac{\beta_1^{-1}(1 - \beta_1^{-t})}{1 - \beta_1^{-1}} \leq \frac{1}{1 - \beta_1} \tag{41}$$

we can proceed to derive the desired result.

$$\|m_t\|_{y_t}^2 \leq \left(\sum_{j=1}^{t}(1 - \beta_{1j})\beta_1^{t-j}\right)\left(\sum_{j=1}^{t}(1 - \beta_{1j})\beta_1^{t-j}\|g_j\|_{y_j}^2\right) \leq \frac{1}{1 - \beta_1}\sum_{j=1}^{t}\beta_1^{t-j}\|g_j\|_{y_j}^2 \tag{42}$$

For $n_t$, because $n_t = \beta_{3t}n_{t-1} + (1 - \beta_{3t})\|z_t\|_{y_t}^2$, a similar discussion still applies:

$$\begin{cases} n_1 = \beta_{31} \cdot n_0 + (1 - \beta_{31})\|z_1\|_{y_1}^2 = \beta_{31} \cdot 0 + (1 - \beta_{31})\|z_1\|_{y_1}^2 \\ n_2 = \beta_{32}(1 - \beta_{32})\|z_1\|_{y_1}^2 + (1 - \beta_{32})\|z_2\|_{y_2}^2 \\ \quad \vdots \\ n_t = \frac{1}{t}\sum_{j=1}^{t}\|z_j\|_{y_j}^2 \end{cases} \tag{43}$$

For $z_j$, because of following:

$$z_j = g_j + \beta_{2t}(g_j - \varphi(g_{j-1})) = (1 + \beta_{2t})g_j - \beta_{2t}\varphi(g_{j-1}) \tag{44}$$

we have that:

$$\|z_j\|_{y_j} \geq (1 + \beta_{2t})\|g_j\|_{y_j} - \beta_{2t}\|\varphi(g_{j-1})\|_{y_j} = \|g_j\|_{y_j} + \beta_{2t}(\|g_j\|_{y_j} - \|\varphi(g_{j-1})\|_{y_j}) \tag{45}$$

$g_j$ is the gradient at $y_j$, $g_{j-1}$ is the gradient at $y_{j-1}$. For the step - size formula, $\eta_t = \frac{\eta}{\sqrt{t}}$, when $t$ is large, $y_j \approx y_{j-1}$, assume $\|g_j\|_{y_j} \approx \|\varphi(g_{j-1})\|_{y_j}$, similar to the Lipschitz continuity of the gradient. Therefore, we have $\|z_j\|_{y_j}^2 \geq \|g_j\|_{y_j}^2$. From another perspective, if $\|z_j\|_{y_j}^2 \leq \|g_j\|_{y_j}^2$, one can always restrict $\beta_{2t} = 0$, in which case $\|z_j\|_{y_j}^2 \geq \|g_j\|_{y_j}^2$. Then for $n_t$:

$$n_t = \frac{1}{t}\sum_{j=1}^{t}\|z_j\|_{y_j}^2 \geq \frac{1}{t}\sum_{j=1}^{t}\|g_j\|_{y_j}^2 \tag{46}$$

Next, consider the sum:

$$\sum_{t=1}^{T}\frac{\eta_t}{\sqrt{n_t}}\|m_t\|_{y_t}^2 \leq \sum_{t=1}^{T}\frac{\eta_t}{(1-\beta_1)} \cdot \frac{\sum_{j=1}^{t}(\beta_1^{t-j}\|g_j\|_{y_j}^2)}{\sqrt{\frac{1}{t}\sum_{j=1}^{t}\|g_j\|_{y_j}^2}}$$

$$= \sum_{t=1}^{T}\frac{\eta}{(1-\beta_1)} \cdot \frac{\sum_{j=1}^{t}(\beta_1^{t-j}\|g_j\|_{y_j}^2)}{\sqrt{\sum_{j=1}^{t}\|g_j\|_{y_j}^2}} \tag{47}$$

$$\leq \frac{2\eta}{(1-\beta_1)^2} \cdot \sqrt{\sum_{j=1}^{T}\|g_j\|_{y_j}^2}$$

Since we assume the gradient is bounded, i.e., $\|g_j\|_{y_j} \leq G$, we can proceed accordingly in the analysis.

$$\sum_{t=1}^{T}\frac{\eta_t}{\sqrt{n_t}}\|m_t\|_{y_t}^2 \leq \frac{2\eta}{(1-\beta_1)^2} \cdot \sqrt{\sum_{j=1}^{T}\|g_j\|_{y_j}^2} \leq \frac{2\eta}{(1-\beta_1)^2} \cdot \sqrt{TG^2} = \frac{2\eta G}{(1-\beta_1)^2} \cdot \sqrt{T} \tag{48}$$

In summary, we have:

$$\sum_{t=1}^{T}\frac{\beta_{1t}}{1-\beta_{1t}} \cdot \frac{\eta_t}{\sqrt{n_t}}\|m_t\|_{y_t}^2 \leq \sum_{t=1}^{T}\frac{\beta_1}{1-\beta_1} \cdot \frac{\eta_t}{\sqrt{n_t}}\|m_t\|_{y_t}^2 \leq \frac{2\eta\beta_1 G}{(1-\beta_1)^3} \cdot \sqrt{T} \tag{49}$$

Considering $B_{42}$, we can directly use the boundedness of the feasible domain to obtain the following expression:

$$\sum_{t=1}^{T}\frac{\sqrt{n_t}}{\eta_t}\|Log_{y_t}(y^*)\|_{y_t}^2 \cdot \frac{\beta_{1t}}{1-\beta_{1t}} \leq \frac{1}{1-\beta_1}\sum_{t=1}^{T}\frac{\sqrt{t}\cdot\sqrt{n_t}}{\eta}D_\infty^2 \cdot \beta_{1t} \tag{50}$$

Since $n_t = \frac{1}{t}\sum_{j=1}^{t}\|z_j\|_{y_j}^2$, we have the following:

$$n_t = \frac{1}{t}\sum_{j=1}^{t}\|z_j\|_{y_j}^2$$

$$\leq \frac{1}{t}\sum_{j=1}^{t}\left(\|g_j\|_{y_j} + \beta_{2t}\|g_j\|_{y_j} + \beta_{2t}\|\varphi(g_{j-1})\|_{y_j}\right)^2 \tag{51}$$

$$\leq \frac{1}{t}G^2\sum_{j=1}^{t}\left(1 + 2\beta_{2t}\right)^2$$

$$\leq \frac{1}{t}G^2 t\left(1 + 2\beta_2\right)^2 \leq (1 + 2\beta_2)^2 G^2$$

The above expression still uses the bounded gradient assumption. Substituting the earlier result, we obtain:

$$\sum_{t=1}^{T} \frac{\sqrt{n_t}}{\eta_t} \|Log_{y_t}(y^*)\|_{y_t}^2 \cdot \frac{\beta_{1t}}{1 - \beta_{1t}} \leq \frac{(1 + 2\beta_2)GD_\infty^2}{1 - \beta_1} \sum_{t=1}^{T} \frac{\sqrt{t}\beta_{1t}}{\eta} \tag{52}$$

For $B_2$, we aim to provide an upper bound for $\sum_{t=1}^{T} \frac{\zeta(\kappa,c)\alpha_t}{2(1-\beta_{1t})} \|u_t\|_{y_t}^2$. According to the update rules:

$$\begin{cases} u_t = m_t + \beta_{2t}v_t \\ v_t = \beta_{2t}\varphi(v_{t-1}) + (1 - \beta_{2t})(g_t - \varphi(g_{t-1})) \end{cases} \tag{53}$$

According to the update rule for $v_t$ and using the triangle inequality, we have

$$\|v_t\|_{y_t} \leq \left((1-\beta_{2t})\cdot\|g_t-\varphi(g_{t-1})\|_{y_t}+\beta_{2t}\|\varphi(v_{t-1})\|_{y_t}\right) \leq \left((1-\beta_{2t})\cdot(\|g_t\|_{y_t}+\|\varphi(g_{t-1})\|_{y_t})+\beta_{2t}\|\varphi(v_{t-1})\|_{y_t}\right) \tag{54}$$

Since $(1 - \beta_{2t}) \cdot (\|g_t\| + \|\varphi(g_{t-1})\|_{y_t}) + \beta_{2t}\|\varphi(v_{t-1})\|_{y_t}$ can be viewed as a convex combination of $(\|g_t\| + \varphi(g_{t-1})\|_{y_t})$ and $\|\varphi(v_{t-1})\|_{y_t}$, we have:

$$(1 - \beta_{2t}) \cdot (\|g_t\|_{y_t} + \|\varphi(g_{t-1})\|_{y_t}) + \beta_{2t}\|\varphi(v_{t-1})\|_{y_t} \leq \sup_{y_t}(\|g_t\|_{y_t} + \|\varphi(g_{t-1})\|_{y_t}) \leq 2G. \tag{55}$$

Therefore, based on the update rule for $u_t$, together with the above result and the triangle inequality, we obtain the following inequality:

$$\begin{aligned} \|u_t\|_{y_t}^2 &= \|m_t + \beta_{2t}v_t\|_{y_t}^2 \\ &\leq (\|m_t\|_{y_t} + \beta_{2t} \cdot \|v_t\|_{y_t})^2 \\ &\leq (sup_{y_t}\|m_t\|_{y_t} + sup_{y_t}\beta_{2t} \cdot \|v_t\|_{y_t})^2 \\ &\leq \left(\frac{G}{1 - \beta_1} + 2\beta_2 G\right)^2 \\ &\leq \left(\frac{3 - 2\beta_1}{1 - \beta_1}\right)^2 G^2 \end{aligned} \tag{56}$$

Therefore, we can obtain the upper bound for $B_2$ as follows:

$$\begin{aligned} B_2 = \sum_{t=1}^{T} \frac{\zeta(\kappa,c)\alpha_t}{2(1-\beta_{1t})} \|u_t\|_{y_t}^2 &\leq \sum_{t=1}^{T} \frac{\zeta(\kappa,c)\cdot\eta}{2(1-\beta_1)\sqrt{t}}\left(\frac{3 - 2\beta_1}{1 - \beta_1}\right)^2 G^2 \\ &\leq \frac{\zeta(\kappa,c)\cdot\eta(3 - 2\beta_1)}{2(1-\beta_1)^3}\sqrt{\sum_{t=1}^{T}\frac{1}{t}}\sqrt{\sum_{t=1}^{T}G^4} \leq \frac{\zeta(\kappa,c)\cdot(3 - 2\beta_1)\eta G^2\sqrt{T}\sqrt{1 + logT}}{2(1-\beta_1)^3} \end{aligned} \tag{57}$$

For $B_3$, we can directly apply the Cauchy-Schwarz inequality to estimate it.

$$\begin{aligned} \sum_{t=1}^{T} \frac{\beta_{2t}}{1 - \beta_{1t}} \langle v_t, Log_{y_t}(y^*)\rangle_{y_t} &\leq \sum_{t=1}^{T} \frac{\beta_{2t}}{1 - \beta_1} \|v_t\|_{y_t}^2 \|Log_{y_t}(y^*)\|_{y_t}^2 \\ &\leq \sum_{t=1}^{T} \frac{\beta_{2t}}{1 - \beta_1} \cdot (2G)^2 \cdot D_\infty^2 \\ &\leq \sum_{t=1}^{T} \frac{4\beta_{2t}}{1 - \beta_1} D_\infty^2 G^2 \end{aligned} \tag{58}$$

We also need to slightly rearrange and simplify the previously obtained expression for $B_1$.

$$B_1 = \frac{D_\infty^2}{2(1-\beta_1)\cdot\eta_T}\sqrt{n_T} \leq \frac{D_\infty^2\sqrt{T}}{2(1-\beta_1)\cdot\eta}\sqrt{(1 + 2\beta_2)^2 G^2} = \frac{GD_\infty^2(1 + 2\beta_2)\sqrt{T}}{2(1-\beta_1)\cdot\eta} \tag{59}$$

By organizing all the terms, we obtain the regret bound:

$$R_T = \sum_{t=1}^{T} (f(y_t) - f(y^*)) \leq \sum_{t=1}^{T} \langle -g_t, Log_{y_t}(y^*) \rangle_{y_t}$$

$$\leq \frac{GD_\infty^2(1+2\beta_2)\sqrt{T}}{2(1-\beta_1)\cdot\eta} + \frac{\zeta(\kappa,c)\cdot(3-2\beta_1)\eta G^2\sqrt{T}\sqrt{1+logT}}{2(1-\beta_1)^3} + \sum_{t=1}^{T}\frac{4\beta_{2t}}{1-\beta_1}D_\infty^2 G^2 \quad (60)$$

$$+ \frac{2\eta\beta_1 G}{(1-\beta_1)^3}\cdot\sqrt{T} + \frac{(1+2\beta_2)GD_\infty^2}{1-\beta_1}\sum_{t=1}^{T}\frac{\sqrt{t}\beta_{1t}}{\eta}$$

Simplification yields the final expression for the regret bound:

$$R_T \leq \frac{\zeta(\kappa,c)\cdot(3-2\beta_1)\eta G^2\sqrt{T}\sqrt{1+logT}}{2(1-\beta_1)^3} + \frac{2\eta\beta_1 G}{(1-\beta_1)^3}\sqrt{T}$$

$$+ \frac{GD_\infty^2(1+2\beta_2)\sqrt{T}}{2(1-\beta_1)\cdot\eta} + \sum_{t=1}^{T}\frac{4D_\infty^2 G^2\beta_{2t}}{1-\beta_1} + \sum_{t=1}^{T}\frac{\sqrt{t}(1+2\beta_2)GD_\infty^2\beta_{1t}}{\eta(1-\beta_1)} \tag{61}$$

### A.1.3 PROOF OF LEMMA

In this section, we will provide proofs for the three lemmas used in Theorem 3.1.

**Lemma 1.** If the feasible domain $D \subset \mathcal{M}$ is geodesically bounded (i.e., there exists a constant $D_\infty$ such that $d(x,y) \leq D_\infty$ for all $x, y \in D$), then for any $x \in D$,

$$\|Log_{y_t}(x)\|_{y_t} \leq D_\infty, \tag{62}$$

where $x$ is any point, and $Log_{y_t}(\cdot)$ is the logarithmic map on $\mathcal{M}$.

*Proof.* By definition, the logarithmic map $Log_x(y)$ maps a point $y \in \mathcal{M}$ to a tangent vector in $T_x\mathcal{M}$ whose norm equals the geodesic distance $d(x,y)$:

$$\|Log_x(y)\|_x = d(x,y). \tag{63}$$

Since $D$ is geodesically bounded, for any $y_t \in D$ and $x \in D$, $d(y_t,x) \leq D_\infty$. Combining the above two results,

$$\|Log_{y_t}(x)\|_{y_t} = d(y_t,x) \leq D_\infty. \tag{64}$$

This completes the proof.

**Lemma 2.** If $f : \mathcal{M} \to \mathbb{R}$ is a geodesically convex function, then for any $y_t \in \mathcal{M}$,

$$f(y_t) - f(y^*) \leq \left\langle -grad f(y_t), Log_{y_t}(y^*) \right\rangle_{y_t}, \tag{65}$$

where $grad f(y_t)$ is the Riemannian gradient of $f$ at $y_t$.

*Proof.* A function $f$ is geodesically convex if, for any geodesic $\gamma : [0,1] \to \mathcal{M}$,

$$f(\gamma(t)) \leq (1-t)f(\gamma(0)) + tf(\gamma(1)), \quad \forall t \in [0,1]. \tag{66}$$

Let $\gamma(0) = y_t$ and $\gamma(1) = y^*$. Then,

$$f(\gamma(t)) \leq (1-t)f(y_t) + tf(y^*). \tag{67}$$

Expand $f(\gamma(t))$ around $t = 0$ using the exponential map $\gamma(t) = Exp_{y_t}(t \cdot Log_{y_t}(y^*))$:

$$f(\gamma(t)) = f(y_t) + t\left\langle grad f(y_t), Log_{y_t}(y^*) \right\rangle_{y_t} + o(t). \tag{68}$$

Substituting into the geodesic convexity inequality:

$$f(y_t) + t\left\langle grad f(y_t), Log_{y_t}(y^*) \right\rangle_{y_t} + o(t) \leq (1-t)f(y_t) + tf(y^*). \tag{69}$$

Rearranging terms and dividing by $t > 0$:

$$\left\langle \mathrm{grad} f(y_t), \mathrm{Log}_{y_t}(y^*) \right\rangle_{y_t} + \frac{o(t)}{t} \leq f(y^*) - f(y_t). \tag{70}$$

Taking $t \to 0$, the higher-order term $\frac{o(t)}{t} \to 0$, yielding:

$$f(y_t) - f(y^*) \leq - \left\langle \mathrm{grad} f(y_t), \mathrm{Log}_{y_t}(y^*) \right\rangle_{y_t}. \tag{71}$$

This completes the proof.

**Lemma 3.** Let $p_1,..., p_k \in \mathbb{R}^d$ and weights $a_1,...,a_k \geq 0$. Then

$$\left\| \sum_{i=1}^{k} a_i p_i \right\|^2 \leq \left( \sum_{i=1}^{k} a_i \right) \left( \sum_{i=1}^{k} a_i \|p_i\|^2 \right). \tag{72}$$

*Proof.* Define $w_i = \sqrt{a_i}$, $v_i := \sqrt{a_i} p_i$. Then

$$\left\| \sum_{i=1}^{k} a_i p_i \right\|^2 = \left\| \sum_{i=1}^{k} w_i v_i \right\|^2 = \left\langle \sum_{i=1}^{k} w_i v_i, \sum_{j=1}^{k} w_j v_j \right\rangle = \sum_{i=1}^{k} \sum_{j=1}^{k} w_i w_j \langle v_i, v_j \rangle$$

$$\leq \underbrace{\left( \sum_{i=1}^{k} w_i^2 \right) \left( \sum_{j=1}^{k} \|v_j\|^2 \right)}_{\text{(by Cauchy Schwarz)}} = \left( \sum_{i=1}^{k} a_i \right) \left( \sum_{j=1}^{k} a_j \|p_j\|^2 \right), \tag{73}$$

which proves the claim.

### A.2 PROOF OF THEOREM 3.2

**Theorem A.2.** *In the bound Equation* (10)*, any non-summation term $K(T)$ satisfies $\mathbf{o}\left( \frac{K(T)}{T} \right) = 0$. For the summation terms, as long as the parameter decay conditions $\mathbf{o}\left( \frac{\sum_{t=1}^{T} \beta_{1t}\sqrt{t}}{T} \right) = 0$, $\mathbf{o}\left( \frac{\sum_{t=1}^{T} \beta_{2t}}{T} \right) = 0$ and $\beta_{3t} = 1 - \frac{1}{t}$ are met, Radan converges to the optimum.*

*Proof.* Recall from Theorem 3.2 that the total regret $R_T$ obeys:

$$R_T \leq K(T) + \sum_{t=1}^{T} A_t = K(T) + \sum_{t=1}^{T} \underbrace{\frac{4D_\infty^2 G^2}{1-\beta_1} \beta_{2t}}_{=:a_{2t}} + \sum_{t=1}^{T} \underbrace{\frac{\sqrt{t}(1+2\beta_2)GD_\infty^2 \beta_{1t}}{\eta(1-\beta_1)}}_{=:a_{1t}}, \tag{74}$$

where we have that:

$$K(T) = \frac{\zeta(\kappa,c) \cdot (3-2\beta_1)\eta G^2 \sqrt{T}\sqrt{1+logT}}{2(1-\beta_1)^3} + \frac{2\eta\beta_1 G}{(1-\beta_1)^3}\sqrt{T} + \frac{GD_\infty^2(1+2\beta_2)\sqrt{T}}{2(1-\beta_1)\cdot\eta} \tag{75}$$

Dividing both sides by $T$ gives:

$$\frac{R_T}{T} \leq \frac{K(T)}{T} + \frac{1}{T}\sum_{t=1}^{T} a_{1t} + \frac{1}{T}\sum_{t=1}^{T} a_{2t}. \tag{76}$$

Set the constants $c_1 = \frac{(1+2\beta_2)GD_\infty^2}{\eta(1-\beta_1)}$, $c_2 = \frac{4D_\infty^2 G^2}{1-\beta_1}$, so that:

$$\frac{1}{T}\sum_{t=1}^{T} a_{1t} = \frac{c_1}{T}\sum_{t=1}^{T} \beta_{1t}\sqrt{t}, \qquad \frac{1}{T}\sum_{t=1}^{T} a_{2t} = \frac{c_2}{T}\sum_{t=1}^{T} \beta_{2t}. \tag{77}$$

Each summand in $K(T)$ scales like $T^{-1/2}$ (up to logarithmic factors), hence $\frac{K(T)}{T} = \mathbf{o}(1) \implies$ $\mathbf{o}\left(\frac{K(T)}{T}\right) = 0$. By hypothesis, $\mathbf{o}\left(\frac{1}{T}\sum_{t=1}^{T}\beta_{1t}\sqrt{t}\right) = 0$, $\mathbf{o}\left(\frac{1}{T}\sum_{t=1}^{T}\beta_{2t}\right) = 0$. Multiplying by the constants $c_1, c_2$ preserves the vanishing rate, so $\frac{1}{T}\sum_{t=1}^{T}a_{1t} = \mathbf{o}(1)$ and $\frac{1}{T}\sum_{t=1}^{T}a_{2t} = \mathbf{o}(1)$. Combining these,

$$\frac{R_T}{T} \leq \underbrace{\mathbf{o}(1)}_{K(T)/T} + \underbrace{\mathbf{o}(1)}_{(1/T)\sum a_{1t}} + \underbrace{\mathbf{o}(1)}_{(1/T)\sum a_{2t}} = \mathbf{o}(1). \tag{78}$$

Hence $\lim_{T\to\infty} R_T/T = 0$, i.e. Radan attains vanishing average regret and converges to the global optimum.

### A.3 PROOF OF THEOREM 3.3

#### A.3.1 PROOF DETAILS

**Theorem A.3.** *On a single constant-curvature manifold $\mathbb{R}^r$, $\mathbb{S}^{s,K}$, or $\mathbb{H}^{h,K}$, the Riemannian gradient of the Riemannian Fuzzy K-Means objective function $J_{FK}$ with respect to the cluster center $c_k$ is uniformly expressed as:*

$$\operatorname{grad}_{c_k} J_{FK} = -2\sum_{i=1}^{N} S_i^{-m} \, d(y_i, c_k)^{-\frac{2m}{m-1}} \, Log_{c_k}(y_i), \tag{79}$$

*where $Log_{c_k}(x_i)$ denotes the logarithmic map of point $x_i$ at $c_k$. The $Log_{c_k}(x_i)$ on three types of constant-curvature manifolds are given as follows.*

$$Log_c(x) = \begin{cases} x - c, & \text{if } x, c \in \mathbb{R}^r, \\ \dfrac{\theta}{\sin(\theta)}\left(x - \cos(\theta)\, c\right), & \theta = \cos^{-1}(K^2\langle c, x\rangle), & \text{if } x, c \in \mathbb{S}^{s,K}, \\ \dfrac{\theta}{\sinh(\theta)}\left(x + K^2\langle c, x\rangle_h \, c\right), & \theta = \cosh^{-1}(K^2\langle c, x\rangle_h), & \text{if } x, c \in \mathbb{H}^{h,K}. \end{cases} \tag{80}$$

*Proof.* To transform it into the above form, we simplify $J_{FK}$ using the expression of $S_i$.

$$\begin{cases} J_{FK}\left(u_{ij}(c_j), c_j\right) = \displaystyle\sum_{i=1}^{N}\left(\sum_{j=1}^{C} d(x_i, c_j)^{-\frac{2}{m-1}}\right)^{1-m}, \\ S_i = \displaystyle\sum_{j=1}^{C}\left(\sum_{p=1}^{Q} d_p^2(x_i^p, c_j^p)\right)^{-\frac{1}{m-1}} = \sum_{j=1}^{C} d(x_i, c_j)^{-\frac{2}{m-1}} \end{cases} \tag{81}$$

Due to Equation 81, we can simply express $J_{FK}$ as Equation 82.

$$J_{FK}\left(u_{ij}(c_j), c_j\right) = \sum_{i=1}^{N}\left(\sum_{j=1}^{C} d(x_i, c_j)^{-\frac{2}{m-1}}\right)^{1-m} = \sum_{i=1}^{N} S_i^{1-m} \tag{82}$$

Consider taking the Riemannian gradient with respect to the kk-th center $c_k$. Obviously, when differentiating $S_i$ with respect to $c_k$, only the term with $j = k$ is nonzero. Therefore, according to the chain rule of Riemannian gradients, we obtain Equation 83.

$$\operatorname{grad}_{c_k} J_{FK} = (1-m)\sum_{i=1}^{N} S_i^{-m} \operatorname{grad}_{c_k}\left(d(x_i, c_k)^{-\frac{2}{m-1}}\right) \tag{83}$$

According to the lemma $\operatorname{grad}_c d(x, c) = -\frac{Log_c(x)}{d(x,c)}$ (proved later), we further simplify Equation 83 and obtain Equation 84.

$$\operatorname{grad}_c d(x, c)^a = a\,d(x,c)^{a-1} \operatorname{grad}_c d(x,c) = -a\,d(x,c)^{a-2} Log_c(x). \tag{84}$$

By setting $a = -\frac{2}{m-1}$, we obtain Equation 85.

$$\text{grad}_{c_k}\left(d(x_i, c_k)^{-\frac{2}{m-1}}\right) = -\frac{2}{m-1}d(x_i, c_k)^{\frac{2}{m-1}-2}Log_{c_k}(x_i) = -\frac{2}{m-1}d(x_i, c_k)^{\frac{-2m}{m-1}}Log_{c_k}(x_i) \tag{85}$$

Simply substituting Equation 85 into Equation 83 yields Equation 11.

$$\begin{aligned}
\text{grad}_{c_k} J_{FK} &= (1-m)\sum_{i=1}^{N} S_i^{-m} \text{grad}_{c_k}\left(d(x_i, c_k)^{-\frac{2}{m-1}}\right) \\
&= (1-m)\sum_{i=1}^{N} S_i^{-m}\left(-\frac{2}{m-1}d(x_i, c_k)^{\frac{-2m}{m-1}}Log_{c_k}(x_i)\right) \tag{86} \\
&= -2\sum_{i=1}^{N} S_i^{-m} d(x_i, c_k)^{-\frac{2m}{m-1}} Log_{c_k}(x_i)
\end{aligned}$$

### A.3.2 PROOF OF LEMMA

We now prove a key lemma.

**Lemma A.4.** *Let $x, c \in \mathcal{M}$, and let $d(x, c)$ denote the geodesic distance between $x$ and $c$. Then we have $\text{grad}_c\, d(x, c) = -\frac{Log_c(x)}{d(x,c)}$.*

*Proof.* First, consider the function $f(c) = \frac{1}{2}d^2(x, c)$ and its directional derivative along the direction $w$, denoted by $\frac{\partial f}{\partial w}$.

$$\frac{\partial f}{\partial w} = \lim_{t \to 0}\frac{1}{2}\frac{d^2(x, Exp_c(tw)) - d^2(x, c)}{t} \tag{87}$$

Let $\gamma(t) = Exp_c(tw)$, which is the geodesic starting from $c$ along $w$. The directional derivative can then be written as:

$$\frac{\partial f}{\partial w} = \lim_{t \to 0}\frac{1}{2}\frac{d^2(x, Exp_c(tw)) - d^2(x, c)}{t} = \frac{d}{dt}\Big|_{t=0}\frac{1}{2}d^2(x, \gamma(t)) \tag{88}$$

According to the standard formula in Riemannian geometry (You), we have:

$$\frac{d}{dt}\Big|_{t=0}\frac{1}{2}d^2(x, \gamma(t)) = \langle -Log_c(x), w\rangle_c \tag{89}$$

Therefore, we obtain the final equation:

$$\frac{\partial f}{\partial w} = \frac{d}{dt}\Big|_{t=0}\frac{1}{2}d^2(x, \gamma(t)) = \langle -Log_c(x), w\rangle_c = \langle \text{grad}_c(\frac{1}{2}d^2(x, c)), w\rangle_c \tag{90}$$

So that:

$$\text{grad}_c(d(x, c)) = \frac{\text{grad}_c(\frac{1}{2}d^2(x, c))}{d(x, c)} = -\frac{Log_c(x)}{d(x, c)} \tag{91}$$

With this, we complete all the proofs.

## B  NOTATIONS

Table 4 lists all the symbols used and their corresponding meanings.

Table 4: Notations in this paper.

| Notation | Description |
|---|---|
| $X = \{x_1, \ldots, x_N\}$ | Dataset notation, consisting of $N$ samples |
| $x_i$ | The $i$-th sample |
| $y_t$ | The coordinate of optimization variable $y$ at step $t$ |
| $c_j$ | The $j$-th cluster center |
| $\mathcal{M}_p$ | The $p$-th component manifold |
| $\otimes_{p=1}^{Q} \mathcal{M}_p$ | The product manifold of $Q$ component manifolds |
| $(x_i^1, x_i^2, \ldots, x_i^Q)$ | The coordinate representation of $x_i$ under $Q$ product manifolds $\otimes_{p=1}^{Q} \mathcal{M}_p$ |
| $d_p(x^p, y^p)$ | The geodesic distance computed from the coordinates of $x$ and $y$ on the $p$-th component manifold |
| $d(x, y)$ | The distance between $x$ and $y$ on the product manifold $\otimes_{p=1}^{Q} \mathcal{M}_p$ |
| $\mathbb{H}^{h_i, K}$ | The hyperbolic space of dimension $h_i$ and curvature $K$, with $K < 0$ |
| $\mathbb{H}^{h_i}$ | The hyperbolic space of dimension $h_i$ and curvature $K$, with $K = -1$ |
| $\mathbb{S}^{s_i, K}$ | The spherical space of dimension $s_i$ and curvature $K$, with $K > 0$ |
| $\mathbb{S}^{s_i}$ | The spherical space of dimension $s_i$ and curvature $K$, with $K = 1$ |
| $\mathbb{R}^{r_i}$ | The Euclidean space of dimension $r_i$ |
| $\mathbb{D}$ | The 2-dimensional Poincaré disk |
| $T_{x^p} \mathcal{M}_p$ | The tangent space at $x^p$ on the $p$-th component manifold |
| $T_x \mathcal{M}$ | The tangent space at $x$ on the product manifold $\otimes_{p=1}^{Q} \mathcal{M}_p$ |
| $\| \cdot \|$ | Euclidean norm |
| $\| \cdot \|_{x_t}$ | Riemannian norm at $x_t$ on the product manifold |
| $\varphi^P_{x^p \to y^p}(u^p)$ | On the $p$-th component manifold, parallel transport $u^p$ from $x^p$ to $y^p$. |
| $\varphi_{x \to y}(u)$ | On the product manifold, parallel transport $u$ from $x$ to $y$. |
| $\mathrm{Exp}^P_{c^p}(u^p)$ | Apply the exponential map to $u^p$ at $c^p$ on the $p$-th component manifold. |
| $\mathrm{Exp}_c(u)$ | Apply the exponential map to $u$ at $c$ on the product manifold. |
| $Log^P_{c^p}(x^p)$ | Apply the logarithmic map to $u^p$ at $c^p$ on the $p$-th component manifold. |
| $Log_c(x)$ | Apply the logarithmic map to $u$ at $c$ on the produc manifold. |
| $log(\cdot)$ | Logarithmic function |
| $\{g_t^p, m_t^p, v_t^p, z_t^p, n_t^p, u_t^p, \alpha_t^p\}$ | Intermediate quantity of Radan on the $p$-th component manifold |
| $\{g_t, m_t, v_t, z_t, n_t, u_t, \alpha_t\}$ | Intermediate quantity of Radan on the product manifold |
| $\zeta(\kappa, c)$ | Curvature function |
| $D_\infty$ | Upper bound of the size of the geodesically convex region |
| $\gamma(t)$ | Geodesic |
| $\langle \cdot, \cdot \rangle_{y_t}$ | Riemannian inner product at $y_t$ |
| $\langle \cdot, \cdot \rangle_h$ | hyperbolic inner product |
| $S_i$ | Intermediate variable of RFK |
| $\beta_{1t}$ | First hyperparameter of Radan |
| $\beta_{2t}$ | Second hyperparameter of Radan |
| $\beta_{3t}$ | Third hyperparameter of Radan |
| $\mathcal{O}(\cdot)$ | Infinitely large of the same order |
| $\mathbf{o}(\cdot)$ | infinitely small of the same order |
| $R_T$ | Regret bound |

## C  RELATED WORK ABOUT CLUSTERING ON MANIFOLD

In terms of clustering algorithm design for data distributed on manifolds, there has not been extensive research so far. In (Miolane et al., 2020), an iterative Riemannian K-Means–style algorithm was implemented by alternately updating the assignments $\{u_{ij}\}$ and the centers $\{c_j\}$, with a time complexity of $\mathcal{O}(\omega\nu)$. Many application scenarios adopt this alternating update paradigm, such as (Wu & Pan, 2025b). Some recent methods for clustering data distributed in hyperbolic spaces have been proposed (Jaćimović & Crnkić, 2025; Ghosh & Das, 2024; Lin et al., 2022). However, these approaches are not applicable to product manifolds and therefore cannot be compared with RFK. There also exist deep learning–based clustering methods (Sun et al., 2023b). However, they lack flexibility, lightweight implementation, and interpretability compared to machine learning–based algorithms. Moreover, deep clustering frameworks often require a clustering procedure similar to RFK to generate pseudo-labels for the learned deep representations. Hence, RFK can serve as a natural and effective replacement for NRK in this context. Moreover, some clustering algorithms assume data lie on an unknown submanifold; algorithms based on this idea still fail to fully respect the data's geometry. Such methods, e.g., Zhong & Pun (2021), are included in our comparisons. In addition, we further compare with several clustering approaches defined on other manifolds (Subbarao & Meer, 2009; Ashizawa et al., 2017; Zhao et al., 2016).

## D  BACKGROUND

Since Riemannian machine learning is a relatively novel research direction and not yet widely familiar to all readers, we provide a detailed introduction to the background of this field in this section.

### D.1  WHAT KIND OF DATA DO WE LEARN?

In fact, Riemannian machine learning focuses on data that have already been represented on manifolds. With the development of representation learning, researchers commonly use neural networks to automatically extract features and obtain new representations of data (Bengio et al., 2013). However, scientists soon realized that many types of data possess non-Euclidean structures, and forcibly embedding them into Euclidean space causes distortions (Yang et al., 2023; Ren et al., 2025).

For example, for periodic data such as cells at different stages of a division cycle, Euclidean embedding fails to capture periodicity, representing them on spheres, hyperspheres, or tori is more appropriate (Davidson et al., 2018). Data with hierarchical structures—such as graphs or trees—are better represented on hyperbolic manifolds (Sala et al., 2018; Mishne et al., 2023). Hyperbolic representations have been widely used in video retrieval (Li et al., 2025), bioinformatics (Ding & Regev, 2021), and large language models (Mandica et al., 2024b).

Moreover, if data simultaneously exhibit multiple structural properties, they are often embedded into product manifolds composed of several manifolds (Chlenski et al., 2025a).

Riemannian machine learning focuses on such data that are already represented on manifolds, aiming to perform classification (Bachmann et al., 2020), clustering (Ashizawa et al., 2017), and regression (Zhou et al., 2025). In the narrow sense, Riemannian machine learning extracts information from these non-Euclidean data, whereas obtaining these manifold-valued representations is the task of Riemannian representation learning.

### D.2  DIFFERENCE FROM MANIFOLD LEARNING

A standard assumption in machine learning is that data lie on some unknown manifold (Izenman, 2012). Manifold learning typically exploits local Euclidean approximations—for example, constructing a KNN graph (Costa & Hero, 2004) and applying spectral clustering methods such as Ncut. The key distinction from Riemannian machine learning is that manifold learning does not know the underlying manifold structure. As a result, its algorithms do not leverage manifold geometry explicitly and often perform poorly when data lie on a known manifold with known structure.

### D.3  BASIC PRINCIPLES OF RIEMANNIAN MACHINE LEARNING

The fundamental principle of Riemannian machine learning is that problems should be considered from the perspective of the manifold itself rather than the Euclidean embedding space (Miolane et al., 2020).

A simple example is the construction of an airport at the geometric center of several countries: the center should be computed using a manifold center (the Fréchet mean) and distances measured on the manifold (the Earth's sphere is a manifold). In contrast, using an Euclidean center could lead to a meaningless point, such as somewhere inside the Earth's interior.

### D.4  MISCELLANEOUS QUESTIONS

**Q1:** How do we determine which manifold a dataset belongs to?

**A1:** Several established methods can identify intrinsic structures in raw data (such as periodicity or hierarchy) and recommend an appropriate embedding manifold (Tabaghi et al., 2021).

**Q2:** How do we embed data onto these manifolds?

**A2:** This has also been extensively studied. Methods such as graph neural networks (Wang et al., 2021), UMAP (McInnes et al., 2018), and coordinate-learning approaches (Gu et al., 2018) can effectively map data into their corresponding manifolds.

Table 5: Description Table of the benchmark datasets

|  | Dataset | Signature | Dimension | Class | Objects |
|---|---|---|---|---|---|
| Synthetic | Gaussian | $\mathbb{R}^4$ | 4 | 3 | 1000 |
| | | $\mathbb{H}^4$ | 5 | 3 | 1000 |
| | | $\mathbb{S}^2\mathbb{H}^2$ | 6 | 3 | 1000 |
| | | $\mathbb{R}^2\mathbb{S}^2\mathbb{H}^2$ | 8 | 3 | 1000 |
| | | $\mathbb{S}^2(\mathbb{H}^2)^2$ | 9 | 3 | 1000 |
| | | $\mathbb{R}^4\mathbb{S}^4\mathbb{H}^4$ | 14 | 3 | 1000 |
| | | $\mathbb{R}^{16}\mathbb{S}^{16}\mathbb{H}^{16}$ | 50 | 3 | 1000 |
| Graph | CiteSeer | $(\mathbb{H}^2)^2$ | 6 | 6 | 2110 |
| | Cora | $\mathbb{H}^4$ | 5 | 7 | 2485 |
| | PolBlogs | $(\mathbb{S}^2)^2$ | 6 | 2 | 1222 |
| | Olsson | $\mathbb{D}$ | 2 | 9 | 382 |
| | Paul | $\mathbb{D}$ | 2 | 20 | 2730 |
| | PolBooks | $\mathbb{D}$ | 2 | 3 | 106 |
| VAE | CIFAR-100 | $(\mathbb{H}^2)^4$ | 12 | 10 | 500000 |
| | Lymphoma | $(\mathbb{S}^2)^2$ | 6 | 10 | 134100 |
| | MNIST | $\mathbb{S}^2\mathbb{E}^2\mathbb{H}^2$ | 8 | 10 | 600000 |

# E    DETAILS OF THE EXPERIMENTAL SETUP

## E.1    DATASETS DESCRIPTION

Table 5 presents the basic information of the datasets we used. Here, Signature refers to the type of manifold onto which the dataset is embedded, Dimension indicates the dimensionality of the embedding space, Class denotes the number of clusters in the data, and Objects specifies the total number of samples in the dataset.

Here, we also provide a brief introduction to the background of these datasets, along with the sources from which each dataset can be obtained.

- All Gaussian datasets are generated using Manify's *'gaussian mixture'* function, with the specific code as follows:

```
from manify.manifolds import ProductManifold
signature = [
    (0.0, 16),   # R^16 (Euclidean space)
    (1.0, 16),   # S^16 (Spherical space)
    (-1.0, 16),  # H^16 (Hyperbolic space)
]
P = ProductManifold(signature, device="cpu", stereographic=False)
n_clusters = 3
X, y_true = P.gaussian_mixture(
    num_points=1000,
    num_classes=n_clusters,
    task="classification",
    cov_scale_points=.1
)
```

  To ensure reproducibility, we also saved the generated data, which can be found here[1].

- CiteSeer, Cora, and PolBlogs are graph datasets, which can be represented in non-Euclidean spaces using the following code:

```
import manify
from manify.utils.dataloaders import load_hf
```

---

[1]https://anonymous.4open.science/r/Manifold-Clustering-Data-3C53/

```
features, dists, adj, labels = load_hf("polblogs")

pm = manify.ProductManifold(signature=[(1.0, 4), (-1.0, 4)])

embedder = manify.CoordinateLearning(pm=pm)
X_embedded = embedder.fit_transform(X=None, D=dists,
    burn_in_iterations=200, training_iterations=800)
```

In fact, the Manify GitHub repository already provides the pre-trained embeddings of these datasets, which you can access there[2], or alternatively obtain from our anonymous GitHub repository[3].

- Olsson, Paul, and PolBooks are also graph datasets, which are embedded in the Poincaré disk. You can access the data here[4], or alternatively obtain it through our anonymous link.
- The datasets CIFAR-100, Lymphoma, and MNIST are obtained using the VAE method provided in Manify. The reference code is as follows:

```
encoder = torch.nn.Sequential(
    torch.nn.Linear(784, 128),
    torch.nn.ReLU(),
    torch.nn.Linear(128, 2 * euclidean_manifold.dim),  # The
        INTRINSIC dimension of the manifold
)
decoder = torch.nn.Sequential(
    torch.nn.Linear(euclidean_manifold.ambient_dim, 128),  # The
        AMBIENT dimension of the manifold
    torch.nn.ReLU(),
    torch.nn.Linear(128, 784),
    torch.nn.Sigmoid(),
)

vae = manify.ProductSpaceVAE(pm=euclidean_manifold, encoder=
    encoder, decoder=decoder)

mnist_embeddings = vae.fit_transform(
    X=mnist_features.reshape(-1, 784), burn_in_iterations=1,
        training_iterations=9, batch_size=128
)
```

Manify also provides the precomputed embeddings of these datasets, which can be accessed here[5] or through our anonymous link. In particular, MNIST performs poorly under small learning rates. In the RFK algorithm, its learning rate is set to 3, while in Experiment 2 we adopt the settings $\{2.1, 2.3, 2.5, 2.7, 3.0\}$.

### E.2 EXPERIMENT 3 SETUP

#### E.2.1 BENCHMARK CLUSTERING ALGORITHMS

We compare it with 10 benchmark clustering algorithms across 7 toy datasets and 9 real-world datasets. These algorithms include K-Means-based methods, graph-based methods, and subspace-based methods. A detailed introduction to each algorithm is provided below.

- NRK, i.e., Naive Riemannian K-Means, is a K-Means-based algorithm that respects the manifold structure but requires double loops. Our main contribution is to modify it in order to reduce its complexity.
- KM partitions data into predefined clusters by minimizing the sum of squared distances between data points and their corresponding cluster centers. It is simple but sensitive to initial centroids and struggles with non-spherical clusters.

---

[2]https://github.com/pchlenski/manify/tree/Dataset-Generation/data/graphs/embeddings
[3]https://anonymous.4open.science/r/Manifold-Clustering-Data-3C53/
[4]https://github.com/drewwilimitis/hyperbolic-learning/tree/master/data/ucidata-zachary
[5]https://github.com/pchlenski/manify/tree/Dataset-Generation/data/mnist/embeddings

- Ncut improves on Ratio-Cut by normalizing the cut, balancing the partition while considering the total graph weight. It's better suited for non-convex and unevenly distributed clusters.

- FCM Fuzzy C-Means (or Fuzzy K-Means), can be regarded as a relaxation of K-Means. Instead of hard assignments, it computes the similarity between each sample and each cluster center as the assignment criterion. It is also a well-known clustering algorithm.

- UFCM This is an unconstrained Fuzzy C-Means algorithm, which aims to replace the constrained alternating optimization in traditional Fuzzy C-Means with an unconstrained gradient descent approach.

- LRR This is a subspace-based clustering method, which leverages low-rank representations to obtain robust subspace clustering results.

- SSC This is also a subspace clustering method, characterized by sparse representation. Through sparse representation, SSC can often identify the core low-rank structure of the data, achieving excellent clustering performance while simultaneously reducing data dimensionality.

- SBMC is a graph-based balanced clustering method. Being graph-based means it clusters data by constructing a graph adjacency matrix. Balanced clustering indicates that the clustering results tend to have roughly equal numbers of samples in each cluster.

- USPEC is one of the representative ensemble clustering algorithms. Ensemble clustering integrates the information from multiple base clusterers to produce a final result, achieving performance far superior to any single clusterer.

- Fast-CD This is a fast and stable clustering algorithm for solving the Ncut loss function, which often achieves clustering results with lower loss than the Ncut itself, combining efficiency and robustness.

To evaluate the clustering performance comprehensively, three metrics are applied, which are clustering accuracy (ACC), normalized mutual information (NMI) and adjusted rand index (ARI). The calculation of these three metrics are displayed below.

### E.2.2 CLUSTERING ACCURACY (ACC)

Clustering Accuracy measures the proportion of correctly clustered data points by aligning predicted cluster labels with ground truth labels. Since clustering algorithms do not inherently assign specific labels, a permutation mapping is applied, often using the Hungarian algorithm, to maximize alignment. The formula for ACC is:

$$\text{ACC} = \frac{\delta(map(\hat{y}_i), y_i)}{n} \tag{92}$$

where $\delta(a, b)$ is an indicator function defined as:

$$\delta(a, b) = \begin{cases} 1, & \text{if } a = b \\ 0, & \text{otherwise,} \end{cases} \tag{93}$$

Here, $\hat{y}_i$ is the predicted label, $y_i$ is the true label, $n$ is the total number of data points, and $map(\hat{y}_i)$ is the permutation mapping function that aligns predicted labels with ground truth labels. ACC ranges from 0 to 1, with higher values indicating better clustering performance.

### E.2.3 NORMALIZED MUTUAL INFORMATION (NMI)

Normalized Mutual Information quantifies the mutual dependence between clustering results and ground truth labels, normalized to account for differences in label distributions. It evaluates the overlap between clusters and true classes using information theory. Given predicted partitions $\{\hat{C}_i\}_{i=1}^{c}$ and ground truth partitions $\{C_i\}_{i=1}^{c}$, NMI is calculated as:

$$\text{NMI} = \frac{\sum_{i=1}^{c} \sum_{j=1}^{c} \left| \hat{C}_i \cap C_j \right| \log \frac{n |\hat{C}_i \cap C_j|}{|\hat{C}_i||C_j|}}{\sqrt{\left( \sum_{i=1}^{c} \left| \hat{C}_i \right| \log \frac{|\hat{C}_i|}{n} \right) \left( \sum_{j=1}^{c} |C_j| \log \frac{C_j}{n} \right)}} \tag{94}$$

Here, $|\cdot|$ denotes the size of a set, and $\hat{C}_i \cap C_j$ represents the number of data points belonging to both the $i$-th predicted cluster and the $j$-th ground truth class. NMI ranges from 0 to 1, where 1 indicates perfect agreement between clustering results and ground truth. It is particularly effective in scenarios with imbalanced class distributions.

### E.2.4 ADJUSTED RAND INDEX (ARI)

The Adjusted Rand Index measures the similarity between predicted clustering and ground truth by comparing all pairs of samples and evaluating whether they are assigned to the same cluster in both results. A contingency table $H$ is first constructed, where each element $h_{ij}$ represents the number of samples in both predicted cluster $\hat{C}_i$ and ground truth cluster $C_j$. The formula for ARI is:

$$\text{ARI}(\bar{C}, C) = \frac{\sum_{ij} \binom{n_{ij}}{2} - \left[\sum_i \binom{n^i}{2} \sum_j \binom{n_j}{2}\right] / \binom{n}{2}}{\frac{1}{2}\left[\sum_i \binom{n^i}{2} + \sum_j \binom{n_j}{2}\right] - \left[\sum_i \binom{n^i}{2} \sum_j \binom{n_j}{2}\right] / \binom{n}{2}} \tag{95}$$

where $\binom{n_{ij}}{2} = \frac{n_{ij}(n_{ij}-1)}{2}$. ARI ranges from -1 to 1, where 1 indicates perfect clustering, 0 represents random assignments, and negative values indicate worse-than-random clustering. ARI is robust to differences in cluster sizes and does not favor a large number of clusters.

### E.2.5 F1 SCORE

The F1 Score evaluates the balance between clustering precision and recall, capturing both the completeness and exactness of the clustering results. It is computed based on pairwise precision and recall between predicted clusters and ground truth classes. The F1 Score is defined as:

$$\text{F1} = \frac{2 \cdot \text{Precision} \cdot \text{Recall}}{\text{Precision} + \text{Recall}} \tag{96}$$

where Precision and Recall are given by:

$$\text{Precision} = \frac{\text{TP}}{\text{TP} + \text{FP}}, \quad \text{Recall} = \frac{\text{TP}}{\text{TP} + \text{FN}} \tag{97}$$

Here, TP (true positives) is the number of data point pairs correctly assigned to the same cluster, FP (false positives) is the number of pairs incorrectly assigned to the same cluster, and FN (false negatives) is the number of pairs that belong to the same ground truth cluster but are assigned to different clusters. F1 Score ranges from 0 to 1, with higher values indicating better clustering quality.

### E.2.6 PURITY

Purity measures the extent to which clusters contain data points from a single ground truth class. For each cluster, the class with the maximum frequency is identified, and the sum of these maximum frequencies over all clusters is normalized by the total number of data points. Purity is defined as:

$$\text{Purity} = \frac{1}{n} \sum_k \max_j |C_k \cap L_j| \tag{98}$$

where $C_k$ denotes the set of data points in cluster $k$, $L_j$ denotes the set of data points in ground truth class $j$, and $n$ is the total number of data points. Purity ranges from 0 to 1, with higher values indicating that clusters are more homogeneous with respect to the true labels.

Table 6: NMI for all benchmarks. OT means out-of-time

| Dataset | | Signature | RFK | NRK | K-Means | Ncut | FCM | UFCM | LRR | SSC | SBMC | USPEC | Fast-CD |
|---|---|---|---|---|---|---|---|---|---|---|---|---|---|
| Synthetic | Gaussian | $\mathbb{R}^4$ | 88.49 | 88.37 | 88.39 | **95.87** | 88.36 | 88.54 | 69.01 | 0.51 | 52.81 | 62.61 | **91.18** |
| | | $\mathbb{H}^4$ | **98.89** | **98.24** | 3.82 | 96.02 | 20.88 | 6.37 | 98.87 | 7.71 | 29.22 | 80.41 | 38.75 |
| | | $\mathbb{S}^2\mathbb{H}^2$ | **84.16** | 84.16 | 69.67 | 73.00 | 71.90 | 71.58 | **89.96** | 0.35 | 42.12 | 62.28 | 72.39 |
| | | $\mathbb{R}^2\mathbb{S}^2\mathbb{H}^2$ | **84.27** | **83.95** | 39.94 | **96.22** | 45.06 | 39.84 | 74.79 | 0.25 | 61.67 | 1.27 | 60.58 |
| | | $\mathbb{S}^2(\mathbb{H}^2)^2$ | **90.37** | **89.25** | 0.53 | 58.31 | 8.44 | 4.50 | 85.62 | 3.99 | 40.49 | 42.56 | 29.40 |
| | | $\mathbb{R}^4\mathbb{S}^4\mathbb{H}^4$ | **95.70** | **95.42** | 7.13 | 57.73 | 57.58 | 8.97 | 87.58 | 5.46 | 68.57 | 45.00 | 86.70 |
| | | $\mathbb{R}^{16}\mathbb{S}^{16}\mathbb{H}^{16}$ | **91.98** | **73.62** | 0.53 | 55.95 | 1.99 | 0.52 | 0.43 | 1.12 | 20.02 | 29.85 | 23.68 |
| Graph | CiteSeer | $(\mathbb{H}^2)^2$ | 0.28 | 0.54 | 0.63 | 0.57 | 0.48 | 0.58 | **0.60** | 0.29 | 0.53 | 0.59 | **0.66** |
| | Cora | $\mathbb{H}^4$ | 0.00 | **0.74** | 0.70 | 0.65 | 0.71 | 0.60 | 0.70 | 0.24 | 0.48 | 0.70 | **0.74** |
| | PolBlogs | $(\mathbb{S}^2)^2$ | **68.76** | 65.77 | 66.94 | 4.02 | 65.41 | **67.26** | 18.61 | 0.08 | 1.41 | 3.79 | 66.09 |
| | Olsson | $\mathbb{D}$ | 70.34 | **70.26** | 67.35 | 66.44 | 66.93 | 66.77 | 37.73 | 58.29 | 54.92 | 51.96 | 65.77 |
| | Paul | $\mathbb{D}$ | **61.70** | **59.78** | 58.28 | 55.95 | 58.11 | 58.24 | 27.25 | 0.67 | 32.06 | 58.59 | 56.41 |
| | PolBooks | $\mathbb{D}$ | **45.48** | **41.59** | 36.83 | 34.71 | 34.36 | 36.17 | 7.34 | 7.34 | 30.13 | 29.50 | 39.34 |
| VAE | CIFAR-100 | $(\mathbb{H}^2)^4$ | 88.24 | OT | 0.52 | OT | 0.62 | 0.24 | OT | OT | OT | 0.17 | OT |
| | Lymphoma | $(\mathbb{S}^2)^2$ | 100.00 | OT | 0.00 | OT | 0.00 | OT | OT | OT | OT | 0.00 | OT |
| | MNIST | $\mathbb{S}^2\mathbb{E}^2\mathbb{H}^2$ | 93.00 | OT | 0.56 | OT | 2.76 | 0.99 | OT | OT | OT | 0.20 | OT |

Table 7: ARI for all benchmarks. OT means out-of-time

| Dataset | | Signature | RFK | NRK | K-Means | Ncut | FCM | UFCM | LRR | SSC | SBMC | USPEC | Fast-CD |
|---|---|---|---|---|---|---|---|---|---|---|---|---|---|
| Synthetic | Gaussian | $\mathbb{R}^4$ | 90.12 | 89.63 | 90.17 | **97.42** | 90.12 | 90.35 | 72.99 | 0.46 | 52.68 | 65.73 | **95.09** |
| | | $\mathbb{H}^4$ | **99.47** | **99.07** | -0.27 | 97.35 | 17.68 | -0.55 | 99.48 | 2.50 | 27.41 | 83.48 | 62.88 |
| | | $\mathbb{S}^2\mathbb{H}^2$ | **87.70** | 85.45 | 70.89 | 74.88 | 74.24 | 73.64 | **93.58** | 0.16 | 41.03 | 65.74 | 82.90 |
| | | $\mathbb{R}^2\mathbb{S}^2\mathbb{H}^2$ | **88.04** | 87.56 | 33.34 | **98.15** | 38.76 | 32.77 | 83.47 | 0.09 | 60.88 | 0.02 | 74.92 |
| | | $\mathbb{S}^2(\mathbb{H}^2)^2$ | **92.48** | **92.12** | -1.55 | 42.72 | -1.06 | -1.27 | 88.69 | -1.50 | 39.29 | 36.48 | 60.84 |
| | | $\mathbb{R}^4\mathbb{S}^4\mathbb{H}^4$ | **97.34** | 96.25 | 0.99 | 55.54 | 53.28 | 2.07 | 91.78 | 0.32 | 68.88 | 44.29 | 92.57 |
| | | $\mathbb{R}^{16}\mathbb{S}^{16}\mathbb{H}^{16}$ | **94.62** | **64.65** | -0.09 | 52.69 | 0.08 | -0.07 | 0.05 | -0.30 | 18.54 | 29.25 | 50.03 |
| Graph | CiteSeer | $(\mathbb{H}^2)^2$ | 0.04 | 0.20 | **0.33** | -0.09 | 0.18 | 0.31 | 0.30 | 0.07 | 0.15 | **0.50** | 0.19 |
| | Cora | $\mathbb{H}^4$ | 0.00 | **0.20** | 0.15 | -0.30 | **0.20** | 0.15 | 0.18 | 0.00 | 0.07 | -0.29 | 0.14 |
| | PolBlogs | $(\mathbb{S}^2)^2$ | **78.67** | 76.08 | 77.09 | 1.17 | 75.79 | 77.46 | 13.82 | -0.01 | 1.83 | 4.84 | **88.67** |
| | Olsson | $\mathbb{D}$ | **51.10** | 50.88 | 49.34 | 47.10 | 48.02 | 48.74 | 22.86 | 44.50 | 33.58 | 44.06 | 45.33 |
| | Paul | $\mathbb{D}$ | **37.36** | **33.48** | 35.26 | 31.71 | 34.84 | 35.76 | 10.34 | -0.02 | 12.19 | 35.24 | 30.90 |
| | PolBooks | $\mathbb{D}$ | **55.38** | 44.87 | 46.47 | 43.99 | 44.66 | 46.01 | 8.58 | **53.34** | 35.74 | 36.40 | 51.02 |
| VAE | CIFAR-100 | $(\mathbb{H}^2)^4$ | **78.67** | OT | 0.05 | OT | 0.10 | 0.01 | OT | OT | OT | 0.01 | OT |
| | Lymphoma | $(\mathbb{S}^2)^2$ | 100.00 | OT | 0.00 | OT | 0.00 | OT | OT | OT | OT | 0.00 | OT |
| | MNIST | $\mathbb{S}^2\mathbb{E}^2\mathbb{H}^2$ | **91.52** | OT | 0.06 | OT | 1.20 | 0.42 | OT | OT | OT | 4.84 | OT |

# F ADDITIONAL EXPERIMENTAL RESULTS

## F.1 EXPERIMENTAL 3 RESULTS

In this section, we present the experimental results of NMI, ARI, F1, and Purity from Experiment 3. Tables 6, 7, 8, and 9 respectively present the NMI, ARI, F1, and Purity metrics of different algorithms across various datasets. It can be observed that, except for the first dataset, RFK consistently and significantly outperforms the other methods on all metrics. This is because the Gauss $\mathbb{R}^4$ dataset lies in Euclidean space, where RFK degenerates to Fuzzy K-Means, thus yielding results similar to K-Means and other implementations of Fuzzy K-Means. Moreover, it is worth noting that for large-scale datasets, RFK is always able to complete execution while achieving highly competitive results.

In addition, we further compare our method with several clustering approaches defined on other manifolds, such as those presented in (Subbarao & Meer, 2009; Ashizawa et al., 2017), and (Zhao et al., 2016). Result is shown in Table 10.

## F.2 SENSITIVITY ANALYSIS

### F.2.1 SENSITIVITY ANALYSIS OF $m$

In addition, we conducted a sensitivity analysis on the parameter $m$ in RFK. The parameter $m$ represents the fuzziness, reflecting the degree of uncertainty in the assignment. In typical implementations of Fuzzy K-Means, $m$ is usually set to the default value of 2. Similarly, in the RFK algorithm, we consistently use the default $m = 2$. This choice is justified because within a sufficiently wide range, the influence of $m$ on the final results is minimal, as illustrated in Figure 5. Specifically, we set $m = \{1.5, 1.75, 2, 2.25, 2.5\}$ and computed the evaluation metrics. It can be observed that $m = 2$ consistently achieves good performance, and the metrics vary only slightly with changes in $m$.

Table 8: F1 for all benchmarks. OT means out-of-time

| | Dataset | Signature | RFK | NRK | K-Means | Ncut | FCM | UFCM | LRR | SSC | SBMC | USPEC | Fast-CD |
|---|---|---|---|---|---|---|---|---|---|---|---|---|---|
| Synthetic | Gaussian | $\mathbb{R}^4$ | **95.49** | 95.35 | 93.53 | 98.30 | 93.50 | 93.65 | 82.33 | 41.26 | 68.74 | 79.84 | 95.39 |
| | | $\mathbb{H}^4$ | **99.77** | **98.42** | 48.25 | 98.26 | 51.35 | 46.87 | 99.66 | 50.51 | 52.04 | 90.09 | 60.52 |
| | | $\mathbb{S}^2\mathbb{H}^2$ | **94.60** | **94.55** | 81.35 | 83.48 | 83.06 | 82.66 | 93.81 | 51.11 | 61.03 | 79.84 | 83.17 |
| | | $\mathbb{R}^2\mathbb{S}^2\mathbb{H}^2$ | **96.27** | 96.27 | 62.32 | 98.85 | 61.73 | 62.31 | 89.57 | 54.79 | 74.80 | 54.41 | 74.35 |
| | | $\mathbb{S}^2(\mathbb{H}^2)^2$ | **98.17** | **98.05** | 52.07 | 66.03 | 51.15 | 52.51 | 92.97 | 52.41 | 60.87 | 67.71 | 55.66 |
| | | $\mathbb{R}^4\mathbb{S}^4\mathbb{H}^4$ | **99.09** | **98.25** | 48.18 | 74.07 | 72.15 | 48.17 | 94.52 | 48.71 | 79.31 | 69.17 | 92.48 |
| | | $\mathbb{R}^{16}\mathbb{S}^{16}\mathbb{H}^{16}$ | **97.75** | 64.93 | 50.26 | 70.18 | 48.14 | 50.46 | 50.90 | 50.46 | 46.16 | 54.03 | 46.08 |
| Graph | CiteSeer | $(\mathbb{H}^2)^2$ | 0.07 | 18.47 | 19.21 | 31.32 | 18.29 | 20.22 | 18.23 | 31.98 | 18.00 | 23.74 | 19.31 |
| | Cora | $\mathbb{H}^4$ | 0.06 | 16.89 | 16.52 | 20.77 | 16.17 | 17.72 | 16.28 | 30.12 | 16.08 | 20.64 | 16.15 |
| | PolBlogs | $(\mathbb{S}^2)^2$ | **94.33** | **93.60** | 88.56 | 64.80 | 87.92 | 88.74 | 64.53 | 66.66 | 50.99 | 53.83 | 88.21 |
| | Olsson | $\mathbb{D}$ | **64.74** | **64.45** | 56.54 | 54.59 | 54.90 | 56.59 | 32.96 | 53.55 | 42.44 | 55.14 | 52.51 |
| | Paul | $\mathbb{D}$ | **47.75** | **46.08** | 40.14 | 36.40 | 39.36 | 41.22 | 16.01 | 14.95 | 17.98 | 40.74 | 35.41 |
| | PolBooks | $\mathbb{D}$ | **72.60** | 57.23 | 66.96 | 67.24 | 66.20 | 66.85 | 43.29 | 71.12 | 50.09 | 63.27 | 70.79 |
| VAE | CIFAR-100 | $(\mathbb{H}^2)^4$ | **69.08** | OT | 6.83 | OT | 6.01 | 8.71 | OT | OT | OT | 6.57 | OT |
| | Lymphoma | $(\mathbb{S}^2)^2$ | **100.00** | OT | 79.51 | OT | 79.51 | OT | OT | OT | OT | 79.51 | OT |
| | MNIST | $\mathbb{S}^2\mathbb{E}^2\mathbb{H}^2$ | **96.18** | OT | 18.17 | OT | 18.13 | 18.18 | OT | OT | OT | 18.21 | OT |

Table 9: Purity for all benchmarks. OT means out-of-time

| | Dataset | Signature | RFK | NRK | K-Means | Ncut | FCM | UFCM | LRR | SSC | SBMC | USPEC | Fast-CD |
|---|---|---|---|---|---|---|---|---|---|---|---|---|---|
| Synthetic | Gaussian | $\mathbb{R}^4$ | 96.00 | 95.84 | 93.80 | 98.47 | 93.77 | 93.92 | 81.75 | 34.58 | 69.47 | 66.61 | 97.20 |
| | | $\mathbb{H}^4$ | **99.80** | **99.00** | 34.25 | 98.62 | 43.13 | 34.14 | 99.62 | 35.24 | 52.68 | 86.04 | 66.50 |
| | | $\mathbb{S}^2\mathbb{H}^2$ | **95.20** | 94.80 | 79.31 | 83.73 | 83.36 | 82.93 | 95.02 | 34.40 | 61.78 | 66.63 | 87.70 |
| | | $\mathbb{R}^2\mathbb{S}^2\mathbb{H}^2$ | **96.20** | 95.80 | 54.55 | 98.76 | 62.12 | 53.89 | 91.64 | 37.77 | 79.25 | 37.75 | 86.00 |
| | | $\mathbb{S}^2(\mathbb{H}^2)^2$ | **97.80** | **97.80** | 37.22 | 61.24 | 37.37 | 37.32 | 92.65 | 37.25 | 64.56 | 51.98 | 68.50 |
| | | $\mathbb{R}^4\mathbb{S}^4\mathbb{H}^4$ | **99.10** | **98.90** | 33.81 | 59.43 | 59.73 | 34.22 | 94.64 | 33.55 | 79.18 | 54.16 | 95.90 |
| | | $\mathbb{R}^{16}\mathbb{S}^{16}\mathbb{H}^{16}$ | **98.00** | **77.10** | 34.17 | 66.26 | 34.23 | 34.18 | 34.22 | 34.10 | 46.59 | 52.82 | 57.20 |
| Graph | CiteSeer | $(\mathbb{H}^2)^2$ | **25.36** | 20.09 | 19.29 | 18.98 | 19.17 | 19.26 | 19.28 | 19.05 | 19.15 | 19.33 | 25.31 |
| | Cora | $\mathbb{H}^4$ | **29.22** | 18.19 | 17.89 | 17.55 | 17.93 | 17.87 | 17.91 | 17.75 | 17.81 | 17.56 | 29.22 |
| | PolBlogs | $(\mathbb{S}^2)^2$ | **94.36** | 93.62 | 88.49 | 50.38 | 87.85 | 88.70 | 54.88 | 50.04 | 50.96 | 52.37 | 93.70 |
| | Olsson | $\mathbb{D}$ | **76.38** | **76.38** | 73.69 | 68.40 | 74.45 | 71.36 | 50.10 | 65.71 | 61.15 | 58.85 | 71.20 |
| | Paul | $\mathbb{D}$ | **58.78** | 59.51 | 58.21 | 57.57 | 57.57 | 54.04 | 35.78 | 14.32 | 35.19 | 51.19 | 56.30 |
| | PolBooks | $\mathbb{D}$ | **81.90** | 77.14 | 78.80 | 78.10 | 77.14 | 77.62 | 55.24 | 79.05 | 76.48 | 75.05 | 80.95 |
| VAE | CIFAR-100 | $(\mathbb{H}^2)^4$ | **79.57** | OT | 5.04 | OT | 5.08 | 5.01 | OT | OT | OT | 5.00 | OT |
| | Lymphoma | $(\mathbb{S}^2)^2$ | **100.00** | OT | 65.99 | OT | 65.99 | OT | OT | OT | OT | 65.99 | OT |
| | MNIST | $\mathbb{S}^2\mathbb{E}^2\mathbb{H}^2$ | **96.09** | OT | 10.06 | OT | 10.61 | 10.23 | OT | OT | OT | 10.03 | OT |

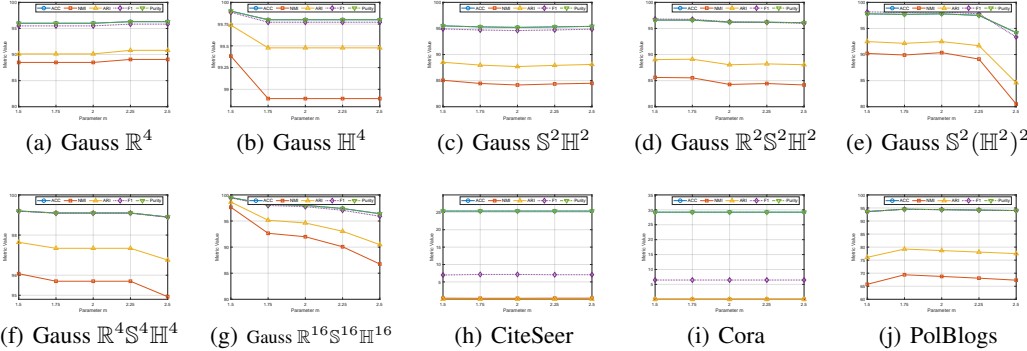

(a) Gauss $\mathbb{R}^4$    (b) Gauss $\mathbb{H}^4$    (c) Gauss $\mathbb{S}^2\mathbb{H}^2$    (d) Gauss $\mathbb{R}^2\mathbb{S}^2\mathbb{H}^2$    (e) Gauss $\mathbb{S}^2(\mathbb{H}^2)^2$

(f) Gauss $\mathbb{R}^4\mathbb{S}^4\mathbb{H}^4$    (g) Gauss $\mathbb{R}^{16}\mathbb{S}^{16}\mathbb{H}^{16}$    (h) CiteSeer    (i) Cora    (j) PolBlogs

Figure 5: Sensitivity Analysis of m

### F.2.2 SENSITIVITY ANALYSIS OF RANDOM INITIALIZATION

We have provided the complete implementation of Riemannian Fuzzy K-Means. It is worth noting that our algorithm adopts random initialization of cluster centers. Therefore, it is necessary to include a sensitivity analysis with respect to random initialization.

We have released the full experimental code from the original paper and fixed all parameters and random seeds. In our experiments, the default seed is set to 1. To assess the robustness of random initialization, we additionally run the algorithm on several datasets with seeds set to 2, 3, and 4, respectively, and obtain the following results:

The column origin corresponds to our default random seed, while each subsequent column reports results obtained using different random seeds. As shown, our algorithm is not sensitive to random initialization of cluster centers. When using different seeds, most of the best results are even better

| Method (ACC) | RFK | NMS | LSLDGC | KGRP |
|---|---|---|---|---|
| Gaussian $\mathbb{R}^4$ | **96.00** | 95.40 | 94.30 | 95.40 |
| Gaussian $\mathbb{H}^4$ | **99.80** | 94.40 | 96.30 | 98.10 |
| Gaussian $\mathbb{S}^2\mathbb{H}^2$ | **95.20** | 94.50 | 93.10 | 95.00 |
| Gaussian $\mathbb{R}^2\mathbb{S}^2\mathbb{H}^2$ | **96.20** | 95.10 | 94.70 | 94.70 |
| Gaussian $\mathbb{S}^2(\mathbb{H}^2)^2$ | **97.80** | 90.20 | 96.40 | 96.00 |
| Gaussian $\mathbb{R}^4\mathbb{S}^4\mathbb{H}^4$ | **99.10** | 99.00 | 98.80 | 95.00 |
| Gaussian $\mathbb{R}^{16}S^{16}H^{16}$ | **98.00** | 88.10 | 90.50 | 94.90 |
| CiteSeer | **25.36** | 20.04 | 21.66 | 22.23 |
| Cora | **29.22** | 26.27 | 25.03 | 20.93 |
| PolBlogs | **94.36** | 94.07 | 92.94 | 90.45 |
| CIFAR-100 | **71.19** | OT | OT | OT |
| Lymphoma | **100.00** | OT | OT | OT |
| MNIST | **96.09** | OT | OT | OT |

Table 10: Comparison of RFK with other manifold-based clustering methods.

| Seed (ACC) | origin | seed=2 | seed=3 | seed=4 |
|---|---|---|---|---|
| Gaussian $\mathbb{R}^4$ | 96.00 | 96.00 | 96.00 | 96.10 |
| Gaussian $\mathbb{H}^4$ | 99.80 | 99.80 | 99.80 | 99.80 |
| Gaussian $\mathbb{S}^2\mathbb{H}^2$ | 95.20 | 95.30 | 95.10 | 95.50 |
| CiteSeer | 25.36 | 25.45 | 25.21 | 24.79 |
| Cora | 29.22 | 29.25 | 28.49 | 29.22 |
| PolBlogs | 94.36 | 93.62 | 93.78 | 94.68 |
| CIFAR-100 | 71.19 | 71.23 | 71.11 | 71.16 |
| Lymphoma | 100.00 | 100.00 | 100.00 | 100.00 |
| MNIST | 96.09 | 95.91 | 96.12 | 95.93 |

Table 11: Sensitivity analysis of random initialization.

than those originally reported, and in all cases, the clustering performance obtained with different seeds remains close to our reported results.

### F.2.3 SENSITIVITY ANALYSIS OF THE NUMBER OF CLUSTER CENTERS

In Riemannian Fuzzy K-Means, an important hyperparameter is the number of cluster centers. Typically, the number of clusters is set to the known number of classes in the dataset. Nevertheless, it is still necessary to analyze the sensitivity of the algorithm to this hyperparameter.

We primarily conduct the sensitivity analysis on the GAUSS datasets, each of which contains three classes regardless of dimensionality. Accordingly, we vary the number of cluster centers to 2, 4, and 5, and evaluate how the clustering ACC changes. In addition, we include real datasets in our analysis, with the true number of classes labeled inside the table.

| Data (ACC) | $c = c_{\text{real}}$ | $c = 2$ | $c = 4$ | $c = 5$ |
|---|---|---|---|---|
| Gaussian $\mathbb{R}^4$ | **96.00** | 67.10 | 91.40 | 64.40 |
| Gaussian $\mathbb{H}^4$ | **99.80** | 73.80 | 87.00 | 64.40 |
| Gaussian $\mathbb{S}^2\mathbb{H}^2$ | **95.20** | 72.90 | 90.30 | 65.90 |
| Gaussian $\mathbb{R}^2\mathbb{S}^2\mathbb{H}^2$ | **96.20** | 58.60 | 75.20 | 59.10 |
| Gaussian $\mathbb{S}^2(\mathbb{H}^2)^2$ | **97.80** | 74.20 | 79.80 | 61.00 |
| Gaussian $\mathbb{R}^4\mathbb{S}^4\mathbb{H}^4$ | **99.10** | 69.30 | 90.50 | 94.60 |
| Gaussian $\mathbb{R}^{16}\mathbb{S}^{16}\mathbb{H}^{16}$ | **98.00** | 70.40 | 83.50 | 97.40 |
| CiteSeer ($c_{\text{real}} = 6$) | **25.36** | 25.26(c=5) | 23.60(c=7) | 24.93(c=8) |
| Cora ($c_{\text{real}} = 7$) | **29.22** | 29.21(c=6) | 23.98(c=8) | 28.37(c=9) |
| PolBlogs ($c_{\text{real}} = 2$) | **94.36** | 70.70(c=3) | 59.00(c=4) | 42.79(c=5) |

Table 12: Sensitivity analysis of the number of cluster centers.

We observe that Riemannian Fuzzy K-Means is relatively sensitive to the number of cluster centers. In fact, this is a well-known property of the entire K-Means family, which explains why the number of centers is the most critical hyperparameter in K-Means–type algorithms.

# G RUN AND REFERENCE CODE

## G.1 RUN THE CODE

The simplest way to run the code is by using the Anonymous Library. The library submitted in the supplementary material is not developed by us, but is part of a publicly available open-source community library. To ensure the double-blind review process, we have anonymized the library name and included only its minimal implementation unit in the supplementary material.

**First, import the package**[6]

```
import AnonymousLibrary
```

You can simply perform clustering with the following code.

```
pm=AnonymousLibrary.ProductManifold(signature=[(0, 16),(1, 16),(-1, 16)])

# Use classification labels, which identify clusters by their center
X_clustering, y_clustering = pm.gaussian_mixture(
    num_points=1000, num_classes=4, seed=2025, task="classification",
        cov_scale_points=0.1
)

# The RFK algorithm is essentially a sklearn-styled clustering algorithm,
     so we call it like this:
rfk = AnonymousLibrary.RiemannianFuzzyKMeans(pm=pm, n_clusters=4,
    random_state=2025)
rfk.fit(X_clustering)
y_pred = rfk.predict(X_clustering)

from sklearn.metrics import normalized_mutual_info_score
nmi = normalized_mutual_info_score(y_clustering, y_pred)
print(f"Riemannian Fuzzy K-Means nmi: {nmi:.2f}")
```

**We kindly suggest** that during the review process, reviewers refrain from searching for the source code related to Riemannian Fuzzy K-Means and Riemannian Adan, as this may violate the double-blind policy. We will provide the fully anonymized versions of RFK and Radan later in the paper.

## G.2 REPLICATION STATEMENT

We fully understand the astonishment when seeing the experimental results, especially the clustering outcomes in Experiment 3. On some datasets, traditional K-Means achieves only **12%** accuracy, while RFK reaches **96%**. Reporting such a striking gap obliges the authors to provide code during the review stage. We are not only willing to provide the source code of RFK but also offer a DEMO that can reproduce the experimental results with a single command, with parameters and random seeds fixed for verification. Our code is available here[7], and our datasets are available here[8].

---

[6]https://anonymous.4open.science/status/AnonymousLibrary-32EB

[7]https://anonymous.4open.science/r/Demo-of-RFK-243B/

[8]https://anonymous.4open.science/r/Manifold-Clustering-Data-3C53/

## G.3 CODE OF RIEMANNIAN FUZZY K-MEANS

```python
from __future__ import annotations

from typing import TYPE_CHECKING

import numpy as np
import torch
from geoopt import ManifoldParameter
from geoopt.optim import RiemannianAdam
from sklearn.base import BaseEstimator, ClusterMixin

if TYPE_CHECKING:
    from beartype.typing import Literal
    from jaxtyping import Float, Int

from ..manifolds import Manifold, ProductManifold
from ..optimizers.radan import RiemannianAdan

class RiemannianFuzzyKMeans(BaseEstimator, ClusterMixin):
    """Riemannian Fuzzy K-Means.

    Attributes:
        n_clusters: The number of clusters to form.
        pm: An initialized manifold object (from manifolds.py) on which
            clustering will be performed.
        m: Fuzzifier parameter. Controls the softness of the partition.
        lr: Learning rate for the optimizer.
        max_iter: Maximum number of iterations for the optimization.
        tol: Tolerance for convergence. If the change in loss is less
            than tol, iteration stops.
        optimizer: The optimizer to use for updating cluster centers.
        random_state: Seed for random number generation for
            reproducibility.
        verbose: Whether to print loss information during iterations.
        losses_: List of loss values during training.
        u_: Final fuzzy partition matrix.
        labels_: Cluster labels for each sample.
        cluster_centers_: Final cluster centers.

    Args:
        n_clusters: The number of clusters to form.
        manifold: An initialized manifold object (from manifolds.py) on
            which clustering will be performed.
        m: Fuzzifier parameter. Controls the softness of the partition.
        lr: Learning rate for the optimizer.
        max_iter: Maximum number of iterations for the optimization.
        tol: Tolerance for convergence. If the change in loss is less
            than tol, iteration stops.
        optimizer: The optimizer to use for updating cluster centers.
        random_state: Seed for random number generation for
            reproducibility.
        verbose: Whether to print loss information during iterations.
    """

    def __init__(
        self,
        n_clusters: int,
        pm: Manifold | ProductManifold,
        m: float = 2.0,
        lr: float = 0.1,
        max_iter: int = 100,
        tol: float = 1e-4,
        optimizer: Literal["adan", "adam"] = "adan",
```

```
2052          random_state: int | None = None,
2053          verbose: bool = False,
2054      ):
2055          self.n_clusters = n_clusters
2056          self.pm = pm
2057          self.m = m
2058          self.lr = lr
2059          self.max_iter = max_iter
2060          self.tol = tol
2061          if optimizer not in ("adan", "adam"):
2062              raise ValueError("optimizer must be 'adan' or 'adam'")
2063          self.optimizer = optimizer
2064          self.random_state = random_state
2065          self.verbose = verbose

      def _init_centers(self, X: Float[torch.Tensor, "n_points n_features"
      ]) -> None:
          if self.random_state is not None:
              torch.manual_seed(self.random_state)
              np.random.seed(self.random_state)

          # Input data X's second dimension should match the pm's ambient
              dimension
          if X.shape[1] != self.pm.ambient_dim:
              raise ValueError(
                  f"Input data X's dimension ({X.shape[1]}) does not match
              "
                  f"the manifold's ambient dimension ({self.pm.ambient_dim
                      })."
              )

          # Generate initial centers using the manifold's sample method
          # We want n_clusters points, each sampled around the manifold's
              origin (mu0)
          # The .sample() method in manifolds.py handles z_mean and sigma/
              sigma_factorized
          # defaulting to mu0 and identity covariances if z_mean or sigma
              are not fully specified
          # or are set to None in a way that triggers this default.

          # For sampling initial centers, we want n_clusters distinct
              points.
          # The .sample() method typically takes a z_mean of shape (
              num_points_to_sample, ambient_dim).
          # If we provide self.pm.mu0 repeated n_clusters times,
          # it samples n_clusters points, each around mu0.
          centers = self.pm.sample(self.n_clusters)

          # IMPORTANT: Use self.manifold.manifold for ManifoldParameter,
          # as self.manifold is our wrapper and self.manifold.manifold is
              the geoopt object.
          self.mu_ = ManifoldParameter(
              centers.clone().detach(),  # type: ignore
              manifold=self.pm.manifold,
          )  # Ensure centers are detached
          self.mu_.requires_grad_(True)

          if self.optimizer == "adan":
              self.opt_ = RiemannianAdan([self.mu_], lr=self.lr, betas
                  =[0.7, 0.999, 0.999])
          else:
              self.opt_ = RiemannianAdam([self.mu_], lr=self.lr, betas
                  =[0.99, 0.999])
```

```python
def fit(self, X: Float[torch.Tensor, "n_points n_features"], y: None
    = None) -> "RiemannianFuzzyKMeans":
    """Fit the Riemannian Fuzzy K-Means model to the data X.

    Args:
        X: Input data. Features should match the manifold's geometry.
        y: Ignored, present for compatibility with scikit-learn's API
            .

    Returns:
        self: Fitted RiemannianFuzzyKMeans instance.

    Raises:
        ValueError: If the input data's dimension does not match the
            manifold's ambient dimension.
        RuntimeError: If the optimizer is not set correctly or if the
            model has not been initialized properly.
    """
    if isinstance(X, np.ndarray):
        X = torch.from_numpy(X).type(torch.get_default_dtype())
    elif not isinstance(X, torch.Tensor):
        X = torch.tensor(X, dtype=torch.get_default_dtype())

    # Ensure X is on the same device as the manifold
    X = X.to(self.pm.device)

    if X.shape[1] != self.pm.ambient_dim:
        raise ValueError(
            f"Input data X's dimension ({X.shape[1]}) in fit() does
                not match "
            f"the manifold's ambient dimension ({self.pm.ambient_dim
                })."
        )

    self._init_centers(X)
    m, tol = self.m, self.tol
    losses = []
    for i in range(self.max_iter):
        self.opt_.zero_grad()
        # self.pm.dist is implemented in manifolds.py and handles
            broadcasting
        d = self.pm.dist(X, self.mu_)  # X is (N,D), mu_ is (K,D) ->
            d is (N,K)
        # Original RFK: d = self.pm.dist(X.unsqueeze(1), self.mu_.
            unsqueeze(0))
        # The .dist in manifolds.py uses X[:, None] and Y[None, :],
            so direct call should work if mu_ is (K,D)

        S = torch.sum(d.pow(-2 / (m - 1)) + 1e-8, dim=1)  # Add
            epsilon for stability
        loss = torch.sum(S.pow(1 - m))
        loss.backward()
        losses.append(loss.item())
        self.opt_.step()
        if self.verbose:
            print(f"RFK iter {i + 1}, loss={loss.item():.4f}")
        if i > 0 and abs(losses[-1] - losses[-2]) < tol:
            break

    # save the result
    self.losses_ = np.array(losses)
    with torch.no_grad():  # Ensure no gradients are computed for
        final calculations
        dfin = self.pm.dist(X, self.mu_)  # Re-calculate dist to
            final centers
```

```python
            inv = dfin.pow(-2 / (m - 1)) + 1e-8   # Add epsilon
            u_final = inv / (inv.sum(dim=1, keepdim=True) + 1e-8)   # Add
                epsilon
        self.u_ = u_final.detach().cpu().numpy()
        self.labels_ = np.argmax(self.u_, axis=1)
        self.cluster_centers_ = self.mu_.data.clone().detach().cpu().
            numpy()
        return self

    def predict(self, X: Float[torch.Tensor, "n_points n_features"]) ->
        Int[torch.Tensor, "n_points"]:
        """Predict the closest cluster each sample in X belongs to.

        Args:
            X: Input data. Features should match the manifold's geometry.

        Returns:
            labels: Cluster labels for each sample in X.

        Raises:
            ValueError: If the input data's dimension does not match the
                manifold's ambient dimension.
            RuntimeError: If the model has not been fitted yet.
        """
        if isinstance(X, np.ndarray):
            X = torch.from_numpy(X).type(torch.get_default_dtype())
        elif not isinstance(X, torch.Tensor):
            X = torch.tensor(X, dtype=torch.get_default_dtype())

        # Ensure X is on the same device as the manifold
        X = X.to(self.pm.device)

        if X.shape[1] != self.pm.ambient_dim:
            raise ValueError(
                f"Input data X's dimension ({X.shape[1]}) in predict()
                    does not match "
                f"the manifold's ambient dimension ({self.pm.ambient_dim
                    })."
            )

        if not hasattr(self, "mu_") or self.mu_ is None:
            raise RuntimeError("The RFK model has not been fitted yet.
                Call 'fit' before 'predict'.")

        with torch.no_grad():
            dmat = self.pm.dist(X, self.mu_)   # X is (N,D), mu_ is (K,D)
                -> dmat is (N,K)
            inv = dmat.pow(-2 / (self.m - 1)) + 1e-8  # Add epsilon
            u = inv / (inv.sum(dim=1, keepdim=True) + 1e-8)   # Add
                epsilon
            labels = torch.argmax(u, dim=1).cpu().numpy()
        return labels
```

### G.4 CODE OF RIEMANNIAN ADAN

```python
from __future__ import annotations

from typing import TYPE_CHECKING

import torch
from geoopt import ManifoldParameter, ManifoldTensor
from geoopt.optim.mixin import OptimMixin

if TYPE_CHECKING:
    from beartype.typing import Any, Callable
    from jaxtyping import Float

from . import _adan

class RiemannianAdan(OptimMixin, _adan.Adan):
    """Riemannian Adan with the same API as :class:adan.Adan.

    Attributes:
        param_groups: iterable of parameter groups, each containing
            parameters to optimize and optimization options
        _default_manifold: the default manifold used for optimization if
            not specified in parameters

    Args:
        params: iterable of parameters to optimize or dicts defining
            parameter groups
        lr: learning rate (default: 1e-3)
        betas: coefficients used for computing (default: (0.98, 0.92,
            0.99))
        eps: term added to the denominator to improve numerical stability
            (default: 1e-8)
        weight_decay: weight decay (L2 penalty) (default: 0)
    """

    def step(self, closure: Callable | None = None) -> Float[torch.Tensor
        , ""] | None:
        """Performs a single optimization step.

        Args:
            closure: A closure that reevaluates the model and returns the
                loss.

        Returns:
            The loss value if closure is provided, otherwise None.
        """
        loss = None
        if closure is not None:
            loss = closure()

        with torch.no_grad():
            for group in self.param_groups:
                betas = group["betas"]
                weight_decay = group["weight_decay"]
                eps = group["eps"]
                learning_rate = group["lr"]
                stablilize = False
                for point in group["params"]:
                    grad = point.grad
                    if grad is None:
                        continue
                    if isinstance(point, ManifoldParameter |
                        ManifoldTensor):
```

```
                            manifold = point.manifold
                    else:
                        manifold = self._default_manifold

                    if grad.is_sparse:
                        raise RuntimeError("RiemannianAdan does not
                            support sparse gradients")

                    state = self.state[point]

                    # State initialization
                    if len(state) == 0:
                        state["step"] = 0
                        # Exponential moving average of gradient values
                        state["exp_avg"] = torch.zeros_like(point)
                        # Exponential moving average of squared gradient
                            values
                        state["exp_avg_sq"] = torch.zeros_like(point)
                        # new param
                        state["exp_avg_diff"] = torch.zeros_like(point)
                        # last step grad
                        state["last_grad"] = torch.zeros_like(point)

                    state["step"] += 1
                    # make local variables for easy access
                    exp_avg = state["exp_avg"]
                    exp_avg_diff = state["exp_avg_diff"]
                    exp_avg_sq = state["exp_avg_sq"]
                    last_grad = state["last_grad"]
                    # actual step

                    grad.add_(point, alpha=weight_decay)
                    grad = manifold.egrad2rgrad(point, grad)
                    # grad_last_diff
                    grad_last_diff = grad - last_grad
                    exp_avg.mul_(betas[0]).add_(grad, alpha=1 - betas[0])
                    # grad_last_diff
                    exp_avg_diff.mul_(betas[1]).add_(grad_last_diff,
                        alpha=1 - betas[1])
                    # z_t
                    zt = grad_last_diff.mul(betas[1]).add_(grad)
                    # z_t^2
                    exp_avg_sq.mul_(betas[2]).add_(manifold.
                        component_inner(point, zt), alpha=1 - betas[2])
                    bias_correction1 = 1 - betas[0] ** state["step"]
                    bias_correction2 = 1 - betas[1] ** state["step"]
                    bias_correction3 = 1 - betas[2] ** state["step"]

                    denom = exp_avg_sq.div(bias_correction3).sqrt_()

                    # copy the state, we need it for retraction
                    # get the direction for ascend
                    direction = (
                        (exp_avg.div(bias_correction1)).add_((
                            exp_avg_diff.div(bias_correction2)), alpha=
                            betas[1])
                    ) / denom.add_(eps)

                    # transport the exponential averaging to the new
                        point
                    new_point, exp_avg_new = manifold.retr_transp(point,
                        -learning_rate * direction, exp_avg)

                    last_grad.copy_(manifold.transp(point, new_point,
                        grad))
```

```python
                        # transport v_t
                        exp_avg_diff.copy_(manifold.transp(point, new_point,
                            exp_avg_diff))
                        exp_avg.copy_(exp_avg_new)
                        point.copy_(new_point)

                        if group["stabilize"] is not None and state["step"] %
                            group["stabilize"] == 0:
                            stablilize = True
                if stablilize:
                    self.stabilize_group(group)
        return loss

    @torch.no_grad()  # type: ignore
    def stabilize_group(self, group: dict[str, Any]) -> None:
        """Stabilizes the parameters in the group by projecting them onto
            their respective manifolds.

        Args:
            group: A dictionary containing the parameters and their
                states.

        Returns:
            None
        """
        for p in group["params"]:
            if not isinstance(p, ManifoldParameter | ManifoldTensor):
                continue
            state = self.state[p]
            if not state:  # due to None grads
                continue
            manifold = p.manifold
            exp_avg = state["exp_avg"]
            exp_avg_diff = state["exp_avg_diff"]
            last_grad = state["last_grad"]
            p.copy_(manifold.projx(p))
            exp_avg.copy_(manifold.proju(p, exp_avg))
            exp_avg_diff.copy_(manifold.proju(p, exp_avg_diff))
            last_grad.copy_(manifold.proju(p, last
```