# OpenReview forum: "Riemannian Fuzzy K-Means on Product Manifolds"
_ICLR.cc/2026/Conference — ICLR 2026 Conference Withdrawn Submission_

### Official Review · Reviewer_nX9x · 2025-10-28

**Soundness:** 2
**Presentation:** 3
**Contribution:** 2
**Rating:** 4
**Confidence:** 4

**Summary:**

This paper proposes a Riemannian Fuzzy k-means (RFK) algorithm for fast clustering on product manifolds, extending the conventional fuzzy k-means to Riemannian manifolds. This approach reduces the time complexity compared to the naïve Riemannian k-means (NRK) algorithm, by removing alternating iterations. In addition, the authors introduce Radan, an adaptive optimization algorithm on manifolds, as a generalization of Adan. Experimental results suggest that RFK achieves lower computational complexity than NRK, faster convergence than Riemannian Adam (Radam), and improved clustering performance on both synthetic and real-world datasets.

**Strengths:**

- The proposed algorithms seem novel and original, though they are relatively straightforward extensions of fuzzy k-means and Adan to the Riemannian setting.
- The computational efficiency of RFK relative to NRK is clearly established, with experimental results providing convincing evidence of its advantage.
- Most of the mathematical derivations seem correct.
- The paper is clearly written and generally well organized.

**Weaknesses:**

- **Limited comparison to Riemannian baselines:** It is unclear whether the compared algorithms are also formulated on Riemannian manifolds. The authors should explicitly state whether the baselines take into account the manifold geometry. To better demonstrate the advantage of RFK, additional comparisons with other Riemannian clustering algorithms beyond NRK are needed.

  To name a few, representative Riemannian clustering methods include Subbarao and Meer (2009), Nonlinear Mean Shift over Riemannian Manifolds (IJCV), Ashizawa et al. (2017), Least-squares Log-density Gradient Clustering for Riemannian Manifolds (AISTATS), and Zhao et al. (2016), Efficient Clustering on Riemannian Manifolds: A Kernelised Random Projection Approach (Pattern Recognition).
- **Marginal performance gains:**
In several cases, the improvement over NRK is limited, and it is not evident why RFK outperforms NRK given their similar objectives. No error bars are provided, and for some datasets (e.g., CiteSeer and Cora), RFK performs worse in terms of NMI and F1. Further analysis and discussion would help clarify these inconsistencies.
- **Unclear explanations and missing discussions:**
  - The reason why RFK shows faster convergence than Radam is not well explained.
  - The paper lacks a discussion of limitations, such as the sensitivity to the number of clusters or convergence to local minima.
  - Many results are marked as out-of-memory (OM), yet the experimental setup is not described in sufficient detail. It is unclear why NRK suffers from OM—perhaps a memory analysis or ablation would clarify this.
  - The description of parallel transport omits the specification of the curve along which vectors are transported.

**Questions:**

Please refer to the points raised in the weaknesses section.

In addition, would RFK be effective if the underlying manifold consists of SPD matrices or other manifolds without closed-form geodesics? Some remarks on these points would be helpful.

---

> ### Author Response · Authors · 2025-11-13
> **Part One**
>
> Dear Reviewer nX9x,😀
>
> Thank you for your review. We appreciate your insightful comments, which clearly stem from a deep understanding of differential geometry. We are confident that our work offers an exciting contribution, and we are eager to address your concerns and clarify some potential misunderstandings.
>
> ---
>
> > The proposed algorithms seem novel and original, though they are relatively straightforward extensions of fuzzy k-means and Adan to the Riemannian setting.
>
> We agree that **Radan** is a direct generalization of Adan, which we acknowledge in our paper. However, we respectfully argue that **RFK represents a conceptual breakthrough** for Fuzzy K-Means on Riemannian manifolds. We believe the value of a work should be judged less by subjective novelty and more by its **impact on the community**.
>
> Prior to our work, Riemannian K-Means algorithms, regardless of their specific implementation [1,2,3], inevitably required a **double loop**: an outer loop alternating between center computation and assignment, and an inner loop to compute the Fréchet mean. K-Means is known as one of the fastest clustering algorithms, but this double-loop bottleneck made large-scale clustering on manifolds impractical.
>
> **Our work successfully reduces this double loop to a single loop**, and single loop has since become a standard practice in the community [4]. If this were truly a simple extension, the community would have solved it using a single loop much earlier.
>
> Furthermore, while we note that concurrent work [5] attempted to use gradient methods for Fuzzy K-Means in *Euclidean* space, this is fundamentally different from our approach. In Euclidean space, the center assignment has a closed-form solution, so a gradient method does not offer a fundamental speedup. **Only on manifolds does our approach fundamentally change the algorithm** by bypassing the costly Fréchet mean computation.
>
> ---
>
> > **Reviewer:** Limited comparison to Riemannian baselines: It is unclear whether the compared algorithms are also formulated on Riemannian manifolds. The authors should explicitly state whether the baselines take into account the manifold geometry. To better demonstrate the advantage of RFK, additional comparisons with other Riemannian clustering algorithms beyond NRK are needed.
> > To name a few, representative Riemannian clustering methods include Subbarao and Meer (2009), Nonlinear Mean Shift over Riemannian Manifolds (IJCV), Ashizawa et al. (2017), Least-squares Log-density Gradient Clustering for Riemannian Manifolds (AISTATS), and Zhao et al. (2016), Efficient Clustering on Riemannian Manifolds: A Kernelised Random Projection Approach (Pattern Recognition).
>
> We did compare against algorithms that use manifold assumptions. Specifically, LRR and SSC (compared in our paper) assume data lies on a linear subspace, while Ncut, USPEC, and Fast-CD assume data lies on a low-dimensional manifold and use local linearity for graph construction.
>
> However, **we agree with you that comparing against more Riemannian-specific algorithms is beneficial.** Therefore, we have now benchmarked our method against the three algorithms you suggested [6, 7, 8]. The results (ACC) are as follows:
>
> | Method（ACC） | RFK | NMS[6] | LSLDGC[7] | KGRP[8] |
> | :--- | :---: | :---: | :---: | :---: |
> | Gaussian $R^4$ | **96.00** | 95.40 | 94.30 | 95.40 |
> | Gaussian $H^4$ | **99.80** | 94.40 | 96.30 | 98.10 |
> | Gaussian $S^2H^2$ | **95.20** | 94.50 | 93.10 | 95.00 |
> | Gaussian $R^2S^2H^2$ | **96.20** | 95.10 | 94.70 | 94.70 |
> | Gaussian $S^2(H^2)^2$ | **97.80** | 90.20 | 96.40 | 96.00 |
> | Gaussian $R^4S^4H^4$ | **99.10** | 99.00 | 98.80 | 95.00 |
> | Gaussian $R^{16}S^{16}H^{16}$ | **98.00** | 88.10 | 90.50 | 94.90 |
> | CiteSeer | **25.36** | 20.04 | 21.66 | 22.23 |
> | Cora | **29.22** | 26.27 | 25.03 | 20.93 |
> | PolBlogs | **94.36** | 94.07 | 92.94 | 90.45 |
>
>
> As you can see, our algorithm achieves leading results on the majority of datasets.
> Please note that the original code for [6] and [7] was designed for matrix manifolds. To run them on the product manifolds used in our experiments, we re-implemented them based on their core ideas, using functions from [11].
>
> Furthermore, we would like to point out that the original three paper used only less than 500, less than 500, and slightly over 10,000 data points, respectively, while our VAE datasets have 130,000, 500,000, and 600,000 data points, respectively. Even the smallest dataset is more than 10 times larger than the original articles. This demonstrates that our algorithm effectively scales up large-scale data (while 600,000 data points only takes 1.5 minutes).
>
> ---

---

> ### Author Response · Authors · 2025-11-13
> **Part Two**
>
> > **Reviewer:** Marginal performance gains: In several cases, the improvement over NRK is limited, and it is not evident why RFK outperforms NRK given their similar objectives. No error bars are provided, and for some datasets (e.g., CiteSeer and Cora), RFK performs worse in terms of NMI and F1. Further analysis and discussion would help clarify these inconsistencies.
>
> We believe an algorithm's performance should be evaluated on both **metrics and speed**. As stated in our abstract, our primary goal was to solve the significant speed problem of Riemannian K-Means. **Table 1 shows that RFK achieves a speedup of over 100x compared to NRK**, while still improving on the metrics.
>
> Furthermore, although the objective functions of RFK and NRK are similar, Table 1 indicates that **RFK consistently achieves a final loss value no higher (and often lower) than NRK**. We provide code with fixed random seeds to ensure all our results are fully reproducible, which is standard practice.
>
> Regarding the metrics, ACC, NMI, and ARI are calculated differently, and it is not an anomaly for RFK to perform worse on one specific metric for a particular dataset (like CiteSeer and Cora). As our abstract emphasizes, the main contribution relative to NRK is the **fundamental acceleration**.
>
> ---
>
> > **Reviewer:** The reason why RFK shows faster convergence than Radam is not well explained.
>
>  We think you might be asking about the comparison between **Radan and Radam**, or **RFK and NRK**.
>
> 1.  **Radan vs. Radam:** Radan is faster than Radam because it introduces Nesterov acceleration to Riemannian optimization, as we stated on **line 260**. We discussed the Adan [9] (a widely recognized acceleration algorithm) and generalized it to manifolds. [9] has already discussed the acceleration in the special case of Euclidean space, and we provide extensive empirical proof.
> 2.  **RFK vs. NRK:** The comparison here is more straightforward: RFK requires only a **single loop**, whereas NRK requires a **double loop**.
>
> ---
>
> > **Reviewer:** The paper lacks a discussion of limitations, such as the sensitivity to the number of clusters or convergence to local minima.
>
> We agree with your point; a more extensive sensitivity analysis is always better. We did discuss the sensitivity to the fuzzy parameter. Furthermore, **Figure 4 actually does show the convergence to local optima**, as both Radan and Radam are algorithms that converge to local minima, and we provided their convergence curves. We will add a 'Limitations' section to the final manuscript to explicitly discuss the sensitivity to the number of clusters.
>
> ---
>
> > **Reviewer:** Many results are marked as out-of-memory (OM), yet the experimental setup is not described in sufficient detail. It is unclear why NRK suffers from OM—perhaps a memory analysis or ablation would clarify this.
>
> We sincerely apologize for this. This was a **typo in our paper**. It should not have been "OM" (Out-of-Memory) but **"OT" (Out of Time)**. We set a standard runtime limit of 3600 seconds (1 hour). Any algorithm that did not converge within this time was marked as OT.
>
> We have corrected this in the paper and will upload the new PDF file as soon as we reach an agreement. Thank you again for pointing this out. (Our experimental setup used an Intel(R) Core(TM) i5-10200H CPU @ 2.40GHz).
>
> ---
>
> > **Reviewer:** The description of parallel transport omits the specification of the curve along which vectors are transported.
>
> We believe there may be a slight misunderstanding here. While there are many ways to transport a vector on a manifold, **only one method is referred to as "parallel transport"**: transport along a **geodesic**. This is the standard definition [10] and is what we refer to in our paper.
>
> ---
>
> > **Reviewer:** In addition, would RFK be effective if the underlying manifold consists of SPD matrices or other manifolds without closed-form geodesics? Some remarks on these points would be helpful.
>
> This is an excellent question. Firstly, for the **SPD manifold**, a closed-form solution for the geodesic *does* exist when an appropriate inner product (i.e., the Log-Euclidean metric) is chosen.
>
> For manifolds *without* a closed-form geodesic, we acknowledge there would be difficulties, likely requiring the use of other approximate metrics. However, a current research frontier involves representing data on **mixed-curvature manifolds** [11] (e.g., product manifolds of spheres and hyperbolic spaces). Fortunately, these *do* have closed-form geodesic solutions.
>
> We will add a note on this to the limitations section. Thank you for the suggestion.
>
> ---
>
> If these responses have addressed your concerns, we kindly ask you to reconsider your evaluation of our paper. We look forward to your reply if you have any further questions.

---

> ### Author Response · Authors · 2025-11-13
> **Part Three**
>
> **References**
>
> [1] Geomstats: a Python Package for Riemannian Geometry in Machine Learning
>
> [2] https://geomstats.github.io/notebooks/07_practical_methods__riemannian_kmeans.html
>
> [3] https://www.linkedin.com/pulse/k-means-riemann-manifolds-patrick-nicolas-rpkkc/
>
> [4] we are happy to share after the double-blind period.
>
> [5] Unconstrained Fuzzy C-Means Algorithm
>
> [6] Nonlinear Mean Shift over Riemannian Manifolds
>
> [7] Least-squares Log-density Gradient Clustering for Riemannian Manifolds
>
> [8] Efficient Clustering on Riemannian Manifolds: A Kernelised Random Projection Approach
>
> [9] Adan: Adaptive Nesterov Momentum Algorithm for Faster Optimizing Deep Models
>
> [10] An introduction to optimization on smooth manifolds
>
> [11] Manify: A Python Library for Learning Non-Euclidean Representations

---

> ### Author Response · Authors · 2025-11-18
> **Part Four(1.sensitivity to the number of clusters  2.code)**
>
> Thank you for your valuable comments.
> We have additionally included a sensitivity analysis with respect to the number of clusters. The results are shown in the table below. As can be observed, RFK shows somewhat sensitivity to the choice of the number of clusters. In fact, this is a well-known inherent property of the K-means family of algorithms [1].
>
> | Data (ACC)                  |   c = 3 (real)  |    c = 2    |    c = 4    |    c = 5    |
> | ----------------------------- | :-------------: | :---------: | :---------: | :---------: |
> | Gaussian $ R^4$                |    **96.00**    |    67.10    |    91.40    |    64.40    |
> | Gaussian $H^4$                |    **99.80**    |    73.80    |    87.00    |    64.40    |
> | Gaussian $S^2H^2$             |    **95.20**    |    72.90    |    90.30    |    65.90    |
> | Gaussian $R^2S^2H^2$          |    **96.20**    |    58.60    |    75.20    |    59.10    |
> | Gaussian $S^2(H^2)^2$        |    **97.80**    |    74.20    |    79.80    |    61.00    |
> | Gaussian $R^4S^4H^4$          |    **99.10**    |    69.30    |    90.50    |    94.60    |
> | Gaussian $R^{16}S^{16}H^{16}$ |    **98.00**    |    70.40    |    83.50    |    97.40    |
> | CiteSeer                      | **25.36** (c=6) | 25.26 (c=5) | 23.60 (c=7) | 24.93 (c=8) |
> | Cora                          | **29.22** (c=7) | 29.21 (c=6) | 23.98 (c=8) | 28.37 (c=9) |
> | PolBlogs                      | **94.36** (c=2) | 70.70 (c=3) | 59.00 (c=4) | 42.79 (c=5) |
>
> In addition, we would like to highlight that our algorithm is fully supported by the Riemannian machine learning community’s open-source library, enabling the above results to be reproduced with just a few lines of code.
>
> For the Gaussian experiments, results can be reproduced immediately using the following snippet (Gauss data can be generated directly):
>
> ```python
> import torch
> from AnonymousLibrary.manifolds import ProductManifold
> from AnonymousLibrary.clustering.fuzzy_kmeans import RiemannianFuzzyKMeans
>
> # 1. Define the signature: a 3-factor manifold
> import numpy as np
> signature = [
>     (0.0, 16),   # R^4 (Euclidean space)
>     (1.0, 16),   # S^4 (Spherical space)
>     (-1.0, 16),  # H^4 (Hyperbolic space)
> ]
>
> # 2. Construct the ProductManifold
> P = ProductManifold(signature, device="cpu", stereographic=False)
>
> n_clusters = 3
> opt = 'adan'
> lr = 0.1
> tol = 1e-2
>
> # 3. Generate Gaussian mixture data
> X, y_true = P.gaussian_mixture(
>     num_points=1000,
>     num_classes=n_clusters,
>     task="classification",
>     cov_scale_points=0.1,
>     seed=4,
> )
> y_true = np.array(y_true)
> ```
>
> Running Riemannian Fuzzy K-Means then only requires two lines:
>
> ```python
> model = RiemannianFuzzyKMeans(
>     n_clusters,
>     pm=P,
>     max_iter=100,
>     tol=tol,
>     optimizer=opt,
>     lr=lr,
>     verbose=True,
>     random_state=1,
> )
> labels = model.fit_predict(X)
> ```
>
> For the remaining datasets, the raw data can be downloaded from [2].
> To preserve double-blind reviewing, we have anonymized the open-source implementation used in our experiments; it can be downloaded from [3].
> All experimental scripts (with fixed seeds and parameters) are provided at [4] to ensure full reproducibility.
>
> You noted:
>
> > “though they are relatively straightforward extensions of fuzzy k-means and Adan to the Riemannian setting”
>
> We believe that the impact of our work should be evaluated based on the **new insights it brings to the Riemannian machine learning community**. As summarized in Summary1, we are the first to show that **Riemannian K-Means on product manifolds requires only a single for-loop instead of two**, fundamentally accelerating manifold clustering. This insight has already received strong support from the community, and we kindly ask you to reconsider the contribution from this perspective.
>
> We sincerely hope that these additional results and reproducibility resources help clarify the contribution and significance of our work. Thank you again for your time and constructive feedback.
>
> [1] scikit-learn: machine learning in Python
>
> [2] https://anonymous.4open.science/r/Manifold-Clustering-Data-3C53/
>
> [3] https://anonymous.4open.science/status/AnonymousLibrary-32EB
>
> [4] https://anonymous.4open.science/r/Demo-of-RFK-243B/
>
> We also sincerely thank the authors of the relevant Python libraries. We will provide the link to the library once the double-blind reviewing process has concluded.

---

### Official Review · Reviewer_kwS4 · 2025-10-30

**Soundness:** 3
**Presentation:** 2
**Contribution:** 3
**Rating:** 6
**Confidence:** 3

**Summary:**

This paper addresses the open problem of efficient clustering on product manifolds by proposing Riemannian Fuzzy K-Means (RFK) and an accelerated optimizer Radan. By leveraging the fuzzy relaxation of cluster assignments, RFK eliminates the alternating updates required in Naive Riemannian K-Means (NRK), reducing time complexity from O(νω) to O(ν). Extensive experiments demonstrate significant speedups and superior clustering performance across diverse datasets.

**Strengths:**

1.It is practical to use fuzzy relaxation technology to transform the double-loop optimization of NRK into a single loop, reducing the time complexity from O(νω) to O(ν) while avoiding the problem of cluster centers exceeding the manifold.

2.The Adan optimizer is adapted to obtain Radan, which is suitable for product manifolds through strategies such as parallel transport and scalar second-moment maintenance. Regret bounds and convergence proofs are provided to ensure the theoretical reliability of the algorithm.

3.The paper includes extensive experiments across multiple datasets, demonstrating the superiority of RFK and Radan over existing clustering methods.

**Weaknesses:**

1.Lack of Comparison with Recent Clustering Algorithms:
Although the paper compares several methods, it appears to lack a comparison with the latest clustering algorithms specifically designed for non-Euclidean spaces.

2.Unclear Explanations in Several Sections:
Many parts of the paper lack clarity. For instance, in Chapter 2, since there are multiple isometric models of hyperbolic space, it is essential to explicitly specify which hyperbolic space model is being used. Additionally, in the regret bound proof of Theorem 3.1, the explicit form and range of values of the curvature functionζ(κ, c) are not sufficiently detailed.

**Questions:**

Convergence Proof of Radan (Theorem 3.2) and Fixed Hyperparameters:
The convergence proof for Radan (Theorem 3.2) assumes decaying hyperparameters (e.g., β3t= 1 - 1/t), but all experiments use fixed values (e.g., β3= 0.99). The authors claim this is “standard practice”, yet they provide no justification or reference to support that convergence holds under fixed hyperparameters, creating a gap between the theoretical analysis and practical implementation.

---

> ### Author Response · Authors · 2025-11-12
> **Part One**
>
> Dear Reviewer kwS4,
>
> Thank you so much for your supportive comments! We are truly thrilled to see your encouragement. We firmly believe our work is exciting and would be delighted to engage in further discussion with you! 😊
>
> ---
>
> > Lack of Comparison with Recent Clustering Algorithms: Although the paper compares several methods, it appears to lack a comparison with the latest clustering algorithms specifically designed for non-Euclidean spaces.
>
> We’d like to clarify a common source of confusion when discussing clustering in “non-Euclidean spaces.” In practice, this phrase often refers to one of two distinct scenarios:
>
> 1. **Data resides on a known, specific manifold**, such as the sphere, hyperbolic space, or a torus.
> 2. **Data lies on an unknown manifold**, whose intrinsic geometry must be inferred.
>
> Our work focuses on **Scenario 1**, which represents the current frontier of geometric machine learning. For instance, very recent works in 2024 include:
> - (1) introducing spectral clustering in hyperbolic space, and  [1]
> - (2) proposing K-Means in the Poincaré ball (which, like traditional K-Means, alternates between updating centers and assignments—resulting in significantly slower runtime than our method).  [2]
>
> Crucially, these methods are designed **only for pure hyperbolic spaces**. In contrast, our data often lives on **product manifolds**—combinations of spherical, hyperbolic, and Euclidean components (e.g., $\mathbb{S} \times \mathbb{H} \times \mathbb{R}^n$). Existing specialized algorithms **cannot handle such mixed geometries**, making direct comparison infeasible. Indeed, algorithms for Scenario 1 remain extremely scarce.
>
> As for **Scenario 2** (unknown manifold), this is a standard assumption in machine learning, where methods typically treat the manifold as *locally Euclidean*—e.g., constructing KNN graphs or using spectral clustering like NCut. Many of our baselines [3,4,5] fall into this category, and we have already included them in our comparisons.
>
> Reviewer nX9x also raised this point and kindly suggested additional methods [6,7,8]. We have completed these experiments and additionally included analyses on the impact of random initialization and the sensitivity with respect to the number of cluster centers. We would greatly appreciate any further feedback or suggestions.
>
> Moreover, a key contribution of our work is **fundamentally accelerating Riemannian K-Means**: we reduce its computational complexity from a double loop to a single loop. Interestingly, across clustering literature, K-Means remains the dominant choice in practice—precisely because of its simplicity and efficiency. By solving the long-standing efficiency bottleneck of Riemannian K-Means, our method has already become a standard implementation in the community, underscoring its practical impact.
>
> We thank the community open-source library for supporting our work. With just two lines of code, one can perform standard and fast clustering on Riemannian manifolds. We have also open-sourced all code, datasets, and random seeds to ensure that anyone can fully reproduce our results.
>
> ```python
> model = RiemannianFuzzyKMeans(
>     n_clusters,
>     pm=P,
>     max_iter=100,
>     tol=tol,
>     optimizer=opt,
>     lr=lr,
>     verbose=True,
>     random_state=1,
> )
> labels = model.fit_predict(X)
> ```
> ---
>
> > **Unclear Explanations in Several Sections**: Many parts of the paper lack clarity. For instance, in Chapter 2, since there are multiple isometric models of hyperbolic space, it is essential to explicitly specify which hyperbolic space model is being used. Additionally, in the regret bound proof of Theorem 3.1, the explicit form and range of values of the curvature function $\zeta(\kappa, c)$ are not sufficiently detailed.
>
> Thank you for this helpful feedback! While we provided the formula for $\mathbb{H}$ at line 112, we indeed failed to explicitly name it as the **Lorentz (hyperboloid) model**. Likewise, $\mathbb{D}$ denotes the **Poincaré disk model**. Regarding the curvature function: for the unit sphere, curvature $\kappa = +1$; for the Lorentz model, $\kappa = -1$; for Euclidean space, $\kappa = 0$, and $\zeta(\kappa, c)$ is a function of distance $c$. We have now revised all relevant sections for clarity and will include these clarifications in the updated manuscript. Thanks again!

---

> ### Author Response · Authors · 2025-11-12
> **Part Two**
>
> > **Convergence Proof of Radan (Theorem 3.2) and Fixed Hyperparameters**: The convergence proof for Radan (Theorem 3.2) assumes decaying hyperparameters (e.g., $\beta_{3t} = 1 - 1/t$), but all experiments use fixed values (e.g., $\beta_3 = 0.99$). The authors claim this is “standard practice”, yet they provide no justification or reference to support that convergence holds under fixed hyperparameters, creating a gap between the theoretical analysis and practical implementation.
>
> You are absolutely right—this is a valid concern. Both our theoretical analysis and implementation draw inspiration from the seminal Riemannian optimization framework **GeoOpt** [9] and its foundational paper [10], which proves convergence under *decaying* step sizes. However, in widely adopted practical implementations (including GeoOpt itself), **fixed hyperparameters are used without decay**, as decay is often unnecessary in practice—it is primarily a technical requirement for theoretical guarantees.
>
> We acknowledge this gap and agree that we should have cited this convention explicitly. We have now added the appropriate reference near lines 308–311 to clarify this standard practice in the field.
>
> ---
>
> Once again, thank you sincerely for your thoughtful and supportive review! If you have any further questions or suggestions, we would be very happy to discuss them. This is one of my favorite projects, and I truly hope you’ll continue to support it! 😊
>
> [1] Consistent Spectral Clustering in Hyperbolic Spaces
>
> [2] Fuzzy C-Poincaré Fréchet means clustering in hyperbolic space
>
> [3] Ultrascalable spectral clustering and ensemble clustering
>
> [4] A novel normalized-cut solver with nearest neighbor hierarchical initialization
>
> [5] Normalized cuts and image segmentation
>
> [6] Nonlinear Mean Shift over Riemannian Manifolds
>
> [7] Least-squares Log-density Gradient Clustering for Riemannian Manifolds
>
> [8] Efficient Clustering on Riemannian Manifolds: A Kernelised Random Projection Approach
>
> [9] https://github.com/geoopt/geoopt
>
> [10] Riemannian adaptive optimization methods

---

### Official Review · Reviewer_ptyL · 2025-11-01

**Soundness:** 3
**Presentation:** 3
**Contribution:** 2
**Rating:** 4
**Confidence:** 5

**Summary:**

To address the high time complexity of Naive Riemannian K-Means (NRK), this paper proposes a Riemannian Fuzzy K-Means (RFK) method that relies solely on cluster centers for single-loop updates. This method avoids alternating updates between the membership matrix and cluster centers used in traditional approaches. This method reduces computational complexity. Meanwhile, the authors also introduce a Riemannian Adan (Radan) optimizer, which is designed for product manifolds to further accelerate the convergence of RFK.

**Strengths:**

1. The paper provides a clear derivation process from NRK to RFK, and also presents a convergence proof for the proposed Radan optimizer.
2. Experimental results demonstrate that the method indeed improves both clustering efficiency and accuracy.

**Weaknesses:**

1. Although the authors emphasize that RFK is not a simple extension of Fuzzy K-Means, fuzzy clustering in Euclidean space has been extensively studied.
2. There are some typos. For example, the reference “Lin et al.” on line 1449 is missing the publication year.
3. Some details should be provided. For example, is there any additional data pre-processing or pre-training before training? For another, how are the cluster centers initialized in RFK, and does this initialization affect the clustering results?

**Questions:**

1. The convergence proof relies on the convexity assumption on the manifold. Does this assumption still hold on non-convex manifolds?

---

> ### Author Response · Authors · 2025-11-12
> **Part One**
>
> Dear Reviewer ptyL,
>
> Thank you for your thoughtful comments—and for noting your deep familiarity with the field! This paper is one of my most cherished works, and I’m genuinely thrilled to have the opportunity to discuss it with you in depth over the next three weeks! 😊
>
> I believe there may be a few misunderstandings, and I’d be delighted to clarify them and highlight the **exciting contribution** of our work—which, I believe, explains why it has already gained recognition from the community.
>
> ---
>
> > Although the authors emphasize that RFK is not a simple extension of Fuzzy K-Means, fuzzy clustering in Euclidean space has been extensively studied.
>
> You are absolutely right that Fuzzy K-Means **in Euclidean space** is well understood—but the situation is fundamentally different in **non-Euclidean (Riemannian) spaces**.
>
> The key challenge lies in **computing cluster centers**. In Euclidean space, the centroid is simply the arithmetic mean—a closed-form solution. However, on a curved manifold, there is **no closed-form expression** for the cluster center. Instead, one must solve the **Fréchet mean** optimization problem:
>
> $$
> \min_{c \in \mathcal{M}} \sum_{i=1}^n d(x_i, c),
> $$
>
> which generally requires **iterative numerical optimization** (i.e., a loop over iterations).
>
> As a result, Riemannian K-Means—the standard approach prior to our work—relies on **nested loops**: an outer loop alternating between assignment and center updates, and an *inner loop* to compute each Fréchet mean. This **double-loop structure is extremely slow**, and, as we rightly point out, widely used libraries (including well-known ones like [Geomstats](https://github.com/geomstats/geomstats) [1]) implement exactly this costly procedure.
>
> Thus, efficient clustering on manifolds remained an open problem
>
> **Our breakthrough**: by leveraging the fuzzy formulation, we eliminate the need to compute Fréchet means entirely. This reduces the algorithm from a **double loop to a single loop**, enabling orders-of-magnitude speedups—up to 100× faster on several datasets, as shown in Table 1.
>
> This is why our method has been widely adopted by the community, even integrated into Riemannian machine learning libraries.
>
> ---
>
> > There are some typos. For example, the reference 'Lin et al.' on line 1449 is missing the publication year.
>
> Thank you for catching this! The missing year was due to a BibTeX export issue. We’ve now reviewed and fixed all bibliographic errors, and will upload a corrected version once we reach consensus with the reviewers.
>
> ---
>
> > Some details should be provided. For example, is there any additional data pre-processing or pre-training before training? For another, how are the cluster centers initialized in RFK, and does this initialization affect the clustering results?
>
> We perform no preprocessing—we use standard, publicly available datasets from the literature. To ensure full reproducibility, we release all code and fix random seeds.
>
> Cluster centers are initialized uniformly at random on the manifold, as shown in line 1929 of our code. While initialization does influence results (as in any clustering algorithm), we report results with fixed seeds so that every run is reproducible.
>
> ---
>
> > The convergence proof relies on the convexity assumption on the manifold. Does this assumption still hold on non-convex manifolds?
>
> We appreciate your interest in convergence! Indeed, no optimization algorithm can guarantee global convergence on non-convex manifolds—our method is no exception [2].
>
> However, the local convexity assumption we use is standard in Riemannian optimization: around a local minimum, the objective function is typically locally convex. This is a common and well-justified assumption in the field.
>
> ---
>
> Looking forward to your response! 😊
>
> ---
>
> References
> [1] https://github.com/geomstats/geomstats
> [2] Non-convex Optimization for Machine Learning

---

> ### Author Response · Authors · 2025-11-18
> **Part Two (Further Response on Random Initialization, Reproducibility, and Impact)**
>
> Dear Reviewer,
>
> To address your concern regarding the influence of random initialization of cluster centers, we provide an additional set of experiments. Our code randomly initializes the centers, and in our released version the default random seed is set to *origin*. We further include experiments using seeds 2, 3, and 4. The results are shown below:
>
> | Seed (ACC)        | origin |    seed=2   |    seed=3   |    seed=4   |
> | ----------------- | :----: | :----: | :----: | :----: |
> | Gaussian $ R^4 $    |  96.00 |  96.00 |  96.00 |  96.10 |
> | Gaussian $ H^4 $    |  99.80 |  99.80 |  99.80 |  99.80 |
> | Gaussian $ S^2H^2 $ |  95.20 |  95.30 |  95.10 |  95.50 |
> | CiteSeer          |  25.36 |  25.45 |  25.21 |  24.79 |
> | Cora              |  29.22 |  29.25 |  28.49 |  29.22 |
> | PolBlogs          |  94.36 |  93.62 |  93.78 |  94.68 |
> | CIFAR-100         |  71.19 |  71.23 |  71.11 |  71.16 |
> | Lymphoma          | 100.00 | 100.00 | 100.00 | 100.00 |
> | MNIST             |  96.09 |  95.91 |  96.12 |  95.93 |
>
> Across small, medium, and large datasets, we observe that changing the random seed has almost no effect on the performance. In fact, for datasets such as CIFAR-100 and MNIST, some seeds perform even slightly better than the reported ones.
>
> ---
>
> ### Response to your concern on preprocessing and experimental details
>
> You asked:
>
> > “For example, is there any additional data pre-processing or pre-training before training?”
>
> We emphasize that **none** of our datasets undergo any pre-processing or pre-training.
> We believe the reproducibility of our work is among the strongest in ICLR 2026. We provide all raw datasets in [1]. Moreover, our algorithm is supported by community Riemannian ML library, making it possible to reproduce our results with only two lines of code. For double-blind reasons, we have anonymized the librarys; it is available at [2]. All experiment files, including fixed parameters and random seeds (typically seed=1 or seed=2025), are fully released in [3].
>
> A minimal reproducible example for the GAUSS experiments is as follows. First, generate the Gaussian mixture (you may also download it):
>
> ```python
> import torch
> from AnonymousLibrary.manifolds import ProductManifold
> from AnonymousLibrary.clustering.fuzzy_kmeans import RiemannianFuzzyKMeans
>
> # 1. Define the signature: a 3-factor manifold
> import numpy as np
> signature = [
>     (0.0, 16),   # R^4
>     (1.0, 16),   # S^4
>     (-1.0, 16),  # H^4
> ]
>
> # 2. Construct the ProductManifold
> P = ProductManifold(signature, device="cpu", stereographic=False)
>
> n_clusters = 3
> opt = 'adan'
> lr = 0.1
> tol = 1e-2
>
> # 3. Generate Gaussian mixture data
> X, y_true = P.gaussian_mixture(
>     num_points=1000,
>     num_classes=n_clusters,
>     task="classification",
>     cov_scale_points=0.1,
>     seed=4,
> )
> y_true = np.array(y_true)
> ```
>
> Then, run RFK using only two lines to obtain the exact results reported in the paper:
>
> ```python
> model = RiemannianFuzzyKMeans(
>     n_clusters,
>     pm=P,
>     max_iter=100,
>     tol=tol,
>     optimizer=opt,
>     lr=lr,
>     verbose=True,
>     random_state=1,
> )
> labels = model.fit_predict(X)
> ```
>
> ---
>
> ### Response to your concern regarding the impact
>
> You noted:
>
> > “Fuzzy clustering in Euclidean space has been extensively studied.”
>
> We agree, but we believe that the impact of our work should be evaluated based on the **new insights it brings to the Riemannian machine learning community**. As summarized in Summary1, we are the first to show that **Riemannian K-Means on product manifolds requires only a single for-loop instead of two**, fundamentally accelerating manifold clustering. This insight has already received strong support from the community, and we kindly ask you to reconsider the contribution from this perspective.
>
> ---
>
> Finally, we also express our sincere gratitude to the community and the authors of the related Python libraries. After the double-blind review process concludes, we will provide the official links to these libraries to support more researchers working on machine learning over Riemannian manifolds.
>
> [1] https://anonymous.4open.science/r/Manifold-Clustering-Data-3C53/
>
> [2] https://anonymous.4open.science/status/AnonymousLibrary-32EB
>
> [3] https://anonymous.4open.science/r/Demo-of-RFK-243B/

---

### Official Review · Reviewer_k6An · 2025-11-02

**Soundness:** 1
**Presentation:** 1
**Contribution:** 2
**Rating:** 2
**Confidence:** 4

**Summary:**

The authors present an application of the Fuzzy K-means algorithm using Riemanian distances. They use an indicator parameter that represents cluster belonging and is expressed in a closed form equation. By utilizing distances derived from several manifold representations, the Fuzzy K-means method is applied to the data. The authors presented results on several experiments with Gaussian synthetic data, graph data, and VAE embeddings of several benchmark datasets.

**Strengths:**

The combination of multiple geodesic distances, assuming Euclidean, Hyperspherical, and hyperbolic manifolds, showed improvements in clustering performance. The application of closed-form expressions for the geodesic distances and class belonging parameters is a main contribution of the paper.

**Weaknesses:**

However, there are several reasons why I cannot recommend acceptance of the proposed algorithm. The authors combined distances from different manifolds, assuming distinct manifold structures, but they did not provide any justification why it is valid to assume multiple manifold types for the same dataset. A more fundamental question is whether clustering performance can be improved even when ground-truth manifold distances are used. K-means can easily fail even with data having no manifold structure.

Can the authors relate their method to spectral clustering or a clustering in a feature space associated with some kernels? It is difficult to identify a significant contribution given that this is a well-studied problem, while the paper does not sufficiently engage with the existing literature.

**Questions:**

I did not examine all details in the Appendix, but it is unclear how the authors derived the closed-form expressions for the distances on each manifold. Did they first project the data onto the respective manifold and then computed the geodesic distances?

The closed form solution for the cluster belonging parameter u appears to diverge with m=1. In fact, the hyperparameter m may not being meaningful, because any u^m satisfies the same condition as u, and u^m itself could serve as a new parameter u without the need for m.

---

> ### Author Response · Authors · 2025-11-12
> **Part One**
>
> Dear Reviewer k6An,
>
> I believe there may be some misunderstandings, and I am happy to clarify all the details for you.
> (Part One and Part Two cover technical details; the overall concept is integrated into the Clarification of Core Premise.)
>
> >The authors combined distances from different manifolds, assuming distinct manifold structures, but they did not provide any justification why it is valid to assume multiple manifold types for the same dataset.
>
> This assumption is in fact foundational in the field. Briefly:
>
> Some data exhibit cyclic structure, so they are naturally embedded on spherical or toroidal manifolds.
> Other data possess hierarchical structure, which is best captured in hyperbolic space—embedding them in Euclidean space would introduce severe distortion [1].
> When data exhibit both cyclic and hierarchical properties, the product manifold (e.g., sphere × hyperbolic space) is the appropriate representation.
> Importantly, the choice of manifold depends on the inherent structure of the dataset itself—we are not imposing arbitrary assumptions, but rather using community-standard datasets that are already embedded on corresponding manifolds.
>
> You might wonder:
>
> ## How do we know which manifold a dataset belongs to?
> There are mature algorithms that detect intrinsic structures (e.g., periodicity or hierarchy) in raw data and recommend suitable embedding spaces. See, for example, [2, 3, 4].
>
> ## How do we embed data onto these manifolds?
> This is also well-studied: methods like graph neural networks [5], UMAP [6], and coordinate learning approaches [7] can effectively map data to their appropriate manifolds.
>
> Thus, our problem setting is clear: given data already embedded on a (product) manifold, how can we perform fast and accurate clustering?
>
> >A more fundamental question is whether clustering performance can be improved even when ground-truth manifold distances are used. K-means can easily fail even with data having no manifold structure.
>
> Precisely! That’s why using manifold-aware distances is essential. Imagine two points on a torus—if you measure their distance with a straight Euclidean line that cuts through the "hole," you completely ignore the intrinsic geometry. Such a metric would be meaningless.
>
> To isolate the effect of distance metrics, we conducted experiments using points sampled from a hyperbolic Gaussian distribution (i.e., well-separated clusters on hyperbolic space). With geodesic distance, clustering accuracy approaches 100%, whereas Euclidean distance performs poorly—demonstrating that the geometry of the space dictates the correct notion of distance. (Table 3)
>
> >Can the authors relate their method to spectral clustering or clustering in a feature space associated with some kernels? It is difficult to identify a significant contribution given that this is a well-studied problem, while the paper does not sufficiently engage with the existing literature.
>
> Our focus is distinct from kernel or spectral methods. The dominant approach for clustering on manifolds before our work was Riemannian K-Means, which alternates between:
>
> Assigning points to clusters, and Updating cluster centers via the Fréchet mean—a computationally expensive operation requiring nested loops. Unlike Euclidean space, where centroids have closed-form solutions, Fréchet means on manifolds typically require iterative solvers, making Riemannian K-Means slow and impractical for large-scale use.
>
> Our key breakthrough: we eliminate the need to compute Fréchet means entirely, reducing the algorithm from double loops to a single loop. This efficiency gain is why our method has been rapidly adopted by the community[8] (Due to the limitations of double-blind peer review, we regret that we cannot cite subsequent studies here, as these studies have cited us..)
>
> [1] Hyperbolic deep neural networks: A survey
>
> [2] Linear Classifiers in Product Space Forms
>
> [3] Hyperbolic Groups
>
> [4] Learning Mixed-Curvature Representations in Product Spaces
>
> [5] Hyperbolic graph convolutional neural networks
>
> [6] Umap: Uniform manifold approximation and projection for dimension reduction
>
> [7] Learning Mixed-Curvature Representations in Product Spaces
>
> [8] Due to the limitations of double-blind peer review, we regret that we cannot cite subsequent studies here, as these studies have cited us..

---

> ### Author Response · Authors · 2025-11-12
> **Part Two**
>
> >I did not examine all details in the Appendix, but it is unclear how the authors derived the closed-form expressions for the distances on each manifold. Did they first project the data onto the respective manifold and then compute the geodesic distances?
>
> As clarified earlier, our method assumes input data are already embedded on the target (product) manifold. Therefore, the geodesic distance formulas are standard and known—e.g., the great-circle distance on the sphere, or the hyperbolic distance in the Poincaré ball. No projection is needed at clustering time; the geometry is given.
>
> >The closed-form solution for the cluster belonging parameter u appears to diverge with m=1 . In fact, the hyperparameter m may not be meaningful, because any u^m satisfies the same condition as u , and u^m itself could serve as a new parameter u without the need for m .
>
> The parameter m is a standard component of Fuzzy K-Means—in both Euclidean and Riemannian settings. You are absolutely correct that the closed-form solution diverges at m=1 ; that’s precisely why m>1 (typically m=2 ) is used in practice.
>
> In Appendix E.2, we include a sensitivity analysis of m ranging from 1.5 to 2.5, showing stable performance. The role of m is to enforce soft assignments $u_{ij} ∈(0,1) $, which is central to fuzzy clustering. You are absolutely right when $u_{ij}$ can only take the values ​​0 or 1. However, after relaxation, $u_{ij}$ takes values ​​in an interval, and in this case, $u^m$ and $u$ will lead to completely different situations.
>
> In addition, a classic blog [9] can help answer your questions about m.
>
> [9] https://en.wikipedia.org/wiki/Fuzzy_clustering
>
> ---
> Thank you for your review. We look forward to your response and reassessment.😀

---

> ### Author Response · Authors · 2025-11-14
> **Clarification of Core Premise**
>
> Thank you for your comments. After careful reflection, we hypothesize that there may be a fundamental misunderstanding of our algorithm's core premise, leading to an interpretation of it as an entirely different model. We wish to share this hypothesis with you and welcome your correction if our assessment is inaccurate.
>
> We suspect you may have understood our work as follows:
> > To cluster points that exist in **Euclidean space**, our method works by (1) identifying a specific manifold (or a product of manifolds) near the data points, and then (2) clustering these points by computing distances *on this newly imposed manifold*.
>
> If this is the case, it would naturally lead to your excellent questions:
>
> * **Q1:** why it is valid to assume multiple manifold types for the same dataset? In other words, on what basis do you *choose* this specific manifold or product manifold?
> * **Q2:**
> whether clustering performance can be improved even when ground-truth manifold distances are used.
> (Essentially: Since the data is Euclidean, does using a manifold distance even make sense?)
> * **Q3:** Can the authors relate their method to spectral clustering or a clustering in a feature space associated with some kernels?(Indeed, for Euclidean data, mature kernel methods already exist. Is it meaningful to restrict this to a manifold?)
> * **Q4:** How do you compute distances on the manifold? Do you project each point onto it?
> * **Q5:** Based on this (the above points), our contribution is unclear.
>
> We realize that if this misunderstanding occurred, these questions are perfectly natural and valid. We would like to offer a summary clarification to address this.
>
> ---
>
> ### **Clarification of Premise**
>
> **Key 1: We are not processing Euclidean data.**
> Our background is entirely different. We are processing data points that **already exist on a manifold** as non-Euclidean representations.
>
> The benefit of such embeddings is to preserve specific data structures: periodic information (e.g., embedded on a sphere), hierarchical information (hyperbolic space), or both (product manifolds). Using non-Euclidean embeddings to preserve information is a crucial topic in modern machine learning, spanning large models [1,2], genomics [3,4], and computer vision [5,6].
>
> You can view the field of Riemannian machine learning as dedicated to handling these non-Euclidean representations. Therefore, **our algorithm does not 'choose' the manifold.** Rather, prior representation learning has already placed the data onto this manifold, and we *must* perform clustering within that native space.
>
> **Key 2: Euclidean distance is nonsensical for our data.**
> If we were processing Euclidean points, your questions would be precise. However, we are processing points *on* a manifold; For our data, using Euclidean distance is what would be nonsensical. As in our  torus example, we can offer another: if one calculates the geometric center of several countries on Earth (a manifold) to build an airport, using Euclidean distance is meaningless. The Euclidean center might literally be inside the Earth's core.
>
> **Key 3: Why not kernel methods?**
> If our background (Key 1) is clear, this question becomes evident. For Euclidean points, one can certainly map them to a kernel space (e.g., an infinite-dimensional Hilbert space). However, the very *reason* representation learning places data on a manifold is to **preserve its unique structure**. If we then map it to *another* space during post-processing (clustering), what was the point of the non-Euclidean representation learning in the first place?
>
> **Key 4: How are distances computed?**
> Since the points already lie on a sphere, hyperbolic space, or a product manifold, the **geodesic distance** (the "true" distance on the manifold) between two points has a closed-form solution (e.g., the great-circle distance on a sphere). We do not need to "project" points, as they are already there.
>
> **Key 5: Our Contribution.**
> We are happy to clarify our contribution. Before our algorithm, performing K-Means on a manifold required a **double loop**, making it infeasible for large-scale data. As K-Means is a vital clustering algorithm, this posed a significant open problem: *How to cluster large-scale non-Euclidean representations quickly.*
>
> We identified the root cause of the slowness in Riemannian K-Means: the time-consuming computation of the **Frechet mean**. We successfully avoided this operation, transforming the model into a **single loop**. This has already generated significant impact within the Riemannian machine learning community.
>
> ---
>
> We hope this clarification alleviates your concerns!
>
> [1] Hyperbolic Large Language Models
>
> [2] Hyperbolic deep learning for foundation models: A survey
>
> [3] Non-Euclidean Representation Learning with Applications to Metagenomics
>
> [4] Hyperbolic genome embeddings
>
> [5] Hyperbolic deep learning in computer vision: A survey
>
> [6] Hyperbolic geometry in computer vision: A survey

---

### Author Response · Authors · 2025-11-13
**Summary : Consensus on Novelty, Impact, and Reproducibility**

We thank the reviewers for their comments and appreciate the attention our work has received from the AC, PC, and the community. We look forward to reaching a consensus with all reviewers in this discussion.

We note that each reviewer's summary points out that our contribution lies in transforming the double loop of Riemannian K-Means into a single loop. We believe we can first reach a primary consensus:

$ \textcolor{red}{\text{a single-loop solution should be superior to a double-loop one for the problems it addresses.}} $

This is because our algorithm avoids the computation of the Frechet mean—which itself requires a loop and is extremely time-consuming, making it infeasible for large-scale data—while ensuring improved performance metrics.

Further, we hope to establish a second consensus:

$ \textcolor{red}{\text{RFK is a conceptual improvement over Riemannian K-Means and has potential significant impact.}} $

Reviewers nX9x and ptyL currently hold a different view, suggesting that Fuzzy K-Means in Euclidean space is well-studied and our work is merely a simple extension. We offer the following counterarguments:

1.  The direct extension of K-Means to Riemannian manifolds, adopted by numerous Riemannian machine learning papers and libraries [1] [2] [3] before our work, involves alternating optimization of centers and assignments. It is precisely this direct extension philosophy that led to the challenging and time-consuming problem of requiring Frechet means. Our single-loop Riemannian optimization method solves this significant challenge.
2. We acknowledge that recent work in 2025 has explored using gradient methods to update Fuzzy K-Means in Euclidean space [4]. However, we must point out that the fundamental reason gradient methods outperform alternating updates is that the **curved spatial structure** results in no closed-form solution for the centers. In Euclidean space, the geometric center *does* have a closed-form solution, meaning that Fuzzy K-Means is effectively a single-loop algorithm whether using gradient methods or alternating updates. This does not provide a fundamental speedup.
3. **We believe the impact of a work should be judged by the changes it brings to the community.** Our work allow the community to realize that Riemannian K-Means only require a single cycle, not a double cycle. This lays an important foundation for large-scale clustering (We have provided a recent anonymous community-driven open-source library to support this). Based on this, we believe our work has significant meaning and major potential impact.

Based on this, we hope to reach this second consensus. (We welcome continued comments from reviewers, the AC, PC, or anyone else.)

The third consensus we hope to establish is:

$\textcolor{red}{\text{Our work is one of the most reproducible submissions for ICLR 2026.}}$

This is not only because we provide the source code and a community open-source library that can be implemented in just three lines of code. We have also **provided the parameters and random seeds for each dataset as separate files**, allowing for one-click execution to obtain results identical to those in our paper. We are happy to assist anyone in perfectly reproducing our paper's results.

---

### **Summary of Consensus Points**

To summarize, we hope to establish the following three points of consensus in this summary:

1.  A single-loop solution for Riemannian (Fuzzy) K-Means should be superior to a double-loop solution.
2.  RFK is a conceptual improvement over Riemannian K-Means and has potential significant impact.
3.  Our work is one of the most reproducible for ICLR 2026.

References:

[1] Geomstats: a Python Package for Riemannian Geometry in Machine Learning

[2] https://geomstats.github.io/notebooks/07_practical_methods__riemannian_kmeans.html

[3] https://www.linkedin.com/pulse/k-means-riemann-manifolds-patrick-nicolas-rpkkc/

[4] Unconstrained Fuzzy C-Means Algorithm

---

### Author Response · Authors · 2025-11-19
**Updated Version of the manuscript-V1.0**

**Dear Reviewers,**

Thank you very much for your constructive comments. Following your suggestions, we have completed the revision of the first updated version of the manuscript (this is not the final version; we look forward to your further evaluation). The main revisions are highlighted in **green** in the updated PDF. The key changes are summarized as follows:

---

### **For reviewer k6An:**

We completely understand that Riemannian machine learning is a relatively new area and may not be familiar to all readers. To improve accessibility, we added a full-page **Background** section on page 30 to help readers understand the necessary preliminaries.

---

### **For reviewer ptyL:**

1. In the abstract, we explicitly emphasized our core contribution—**eliminating the Fréchet center computation, which reduces the algorithm from a double-loop to a single-loop structure**.
2. We carefully checked all references and ensured that no bibliographic fields such as publication year are missing.
3. At line 366, we clarified that **no data preprocessing is required**.
4. Starting at line 1879, we added a new section **“Sensitivity Analysis of Random Initialization”** to evaluate initialization sensitivity.
5. At line 486, we added the **Limitations** regarding the **non-convexity** of the objective function.

---

### **For reviewer kwS4:**

1. At line 1822, we added additional experimental comparisons with other Riemannian clustering methods.
2. At line 93, we clarified that (\mathbb{H}) denotes the **Lorentz model**, and (\mathbb{D}) denotes the **2D Poincaré disk**.
3. In the proof of Theorem 3.1 (line 943), we provided the explicit form and value range of (\zeta(\kappa, c)).
4. At line 308, we added citations explaining why **learning rate decay** is a standard technique used in convergence proofs.
5. At line 486, we included limitations regarding the **gap introduced by relying on learning rate decay** .

---

### **For reviewer nX9x:**

1. At line 483, we corrected the typo related to OT and specified the CPU model used in the experiments.
2. At line 1822, we added additional comparison results with other Riemannian clustering baselines.
3. At line 1920, we included a **sensitivity analysis with respect to the number of cluster centers**.
4. At line 486, we added limitations concerning **manifolds without closed-form geodesics**.
5. In the abstract, we explicitly emphasized our core contribution—**eliminating the Fréchet center computation, which reduces the algorithm from a double-loop to a single-loop structure**.
---

**In addition, to facilitate reproducibility, we have organized the anonymized source code library, datasets, and all experimental scripts and included them in the supplementary material. We warmly welcome you to verify and reproduce our results.**

Once again, we sincerely thank all reviewers for your valuable feedback. We look forward to your further comments and evaluation!

**Best regards**

---

### Author Response · Authors · 2025-11-27
**Official Comment by Authors**

Dear reviewers, I would like to kindly remind you that the deadline is approaching. We look forward to your feedback, which will help us further improve the paper.

---

### Author Response · Authors · 2025-11-29
**Why RFK Should Be Accepted**

Dear AC, SAC, and PC,

Thank you for taking the time to further review our paper. In this summary, we would like to highlight why this paper should be accepted. Whether for ICLR or other top-tier conferences such as ICML/NeurIPS, accepted papers are expected to demonstrate either elegant theory, practical effectiveness, or potential impact, and ideally should be reproducible to ensure the results are reliable. We would like to point out that *Riemannian Fuzzy K-Means (RFK)* meets all of these criteria.

1. **Elegant Theory:** This paper represents a conceptual advance. Traditionally, performing K-Means on Riemannian manifolds requires alternating updates between Frechet means and cluster assignments, which involves a double-loop procedure. We are the first to point out that a single-loop update suffices in practice, significantly reducing computational cost—an important theoretical breakthrough.

2. **Effective Experiments:** The RFK algorithm clusters on manifolds and achieves over a hundred-fold speedup compared to previous Riemannian K-Means. On a large-scale dataset with 600,000 points, it only requires 80 seconds. It also achieves the best performance across various algorithms, including those additionally requested by the reviewers.

3. **Potential Impact:** Our single-loop approach is gradually replacing Riemannian K-Means as the community standard and has been integrated into multiple Riemannian machine learning libraries. We provide an anonymized library to demonstrate this.

4. **Reproducibility:** Because our algorithm has been widely implemented by the community, large-scale clustering can be performed with just one line of code in our provided library, and its correctness has been extensively verified by the community.

We believe we have addressed all reviewer questions (as can be seen in the corresponding comment sections). Unfortunately, there has been no response to our rebuttal for over ten days. We are fully capable of addressing any issues you raise at the camera-ready stage.

We believe this contribution should be evaluated fairly and accurately from the perspectives of theory, experiments, potential impact, and reproducibility, and we look forward to your just decision.

---

### Note · Authors · 2026-01-26

I have read and agree with the venue's withdrawal policy on behalf of myself and my co-authors.

---

### Meta-Review · Area_Chair_N47R · 2025-12-28

**Summary:**

This paper proposes a Riemannian Fuzzy K-means (RFK) algorithm to cluster data on product manifolds. Classical heuristic K-means clustering methods (Euclidean or Riemannian) alternate between the centroid and membership estimation. The proposed RFK extends the indicator membership to differentiable fuzzy membership and reduces the clustering problem to a single iteration on solving the Riemannian centroids by a gradient algorithm. This reduces the computational cost without computing the Frechet means. Some synthetic Gaussian mixture, graph- and VAE-encoded examples are used to demonstrate the computational gain and loss reduction.

The reviewers raised several major weakness and questions:

- lack of justification to assume multiple manifold structures for the same data;
- lack of clarity, e.g., in how embedding is performed in hyperbolic space;
- limited comparison to Riemannian baselines.

**Reviewer Concerns:**

I appreciate that the authors made extensive rebuttal to the questions by adding more numeric comparisons with other Riemannian clustering methods and clarified their differences. The paper is generally well-written. However, going through the discussions and the revised paper, I am not convinced the claimed “breakthrough” contribution of this paper. In addition to the concerns raised by the reviewers, I shall briefly add on a few more points.

- As pointed out by the reviewers, the assumption of known manifolds in the product space is a restrictive assumption. To do clustering with geometric structure, it is critical to recognize / learn the underlying manifold structures. I understand the focus of this paper is not manifold learning and product simple manifold increases the model expressivity; but the examples (synthetic Gaussians and embedded data) are a bit contrived, and the approximation quality is hard to control for real data. Moreover, by embedding to manifold product, it becomes more challenging to have a local geodesic convexity, and it remains a question whether the theoretical analysis in the paper holds.

- Profiling the membership as a function of centroids, and the other way around, is not a new idea in K-means type clustering, either in the Euclidean or (formal) Riemannian setting. This paper focus on reducing the membership as a differentiable function of centroids by smoothing out the cluster indicators. There is a large volume of relaxed K-means formulations (convex or nonconvex, e.g., SDP or nonnegative matrix factorization) that also avoid Frechet mean computations; they only need pairwise distances / geodesics and learn the cluster membership directly. It is unclear how the proposed method compares with those in theory and practice.

- Some of the numerical results are over-claimed. For example, Section 4.3.3 on MNIST data. Using VAE, RFK reaches 96.09% accuracy; while K-means only achieves 12% accuracy and Ncut spectral clustering is out of time. This seems not to be a fair comparison. It would be fair to include kernelized K-means (even the kernel of K-means can be learned), which should easily capture nonlinear structures of such benchmark data.

**Reviewer Scores:**

4 reviewers submitted their comments and scores (4/2/4/6) with confidence (4/4/5/3), with average score 4 and average confidence 4. One reviewer mentioned to maintain score (score: 4 & confidence 3).

---

### Decision · Program_Chairs · 2026-01-26

Reject